# Robust Bayesian Optimisation with Unbounded Corruptions

**Abdelhamid Ezzerg** [1]   **Ilija Bogunovic** [2]   **Jeremias Knoblauch** [1]

## Abstract

Bayesian Optimization is critically vulnerable to extreme outliers. Existing provably robust methods typically assume a bounded cumulative corruption budget, which makes them defenseless against even a single corruption of sufficient magnitude. To address this, we introduce a new adversary whose budget is only bounded in the *frequency* of corruptions, not in their magnitude. We then derive RCGP-UCB, an algorithm coupling the famous upper confidence bound (UCB) approach with a Robust Conjugate Gaussian Process (RCGP). We present stable and adaptive versions of RCGP-UCB, and prove that they achieve sublinear regret in the presence of up to $O(T^{1/4})$ and $O(T^{1/7})$ corruptions with possibly infinite magnitude. This robustness comes at near zero cost: without outliers, RCGP-UCB's regret bounds match those of the standard GP-UCB algorithm.

## 1. Introduction

Bayesian Optimization (BO) is a well-established and powerful framework for the sample-efficient optimization of expensive, black-box functions (Shahriari et al., 2016; Frazier, 2018; Garnett, 2023; Brochu et al., 2010). It has achieved significant success across a plethora of domains, including material science (Khatamsaz et al., 2023), drug discovery (Colliandre & Muller, 2024; Gessner et al., 2024; Guan & Fu, 2022), robotics (Calandra et al., 2014), machine learning hyperparameter tuning (Lindauer et al., 2022; Nguyen, 2019), and automated Machine Learning (AutoML) (Shen et al., 2024). Most surrogate functions in BO rely on Gaussian Process (GP) models (Rasmussen & Williams, 2005) to approximate the objective function and guide the search for the optimum, famously exemplified by the GP-UCB algorithm (Srinivas et al., 2010).

[1]University College London [2]University of Basel. Correspondence to: Abdelhamid Ezzerg <abdel-hamid.ezzerg.24@ucl.ac.uk>.

*Proceedings of the 43$^{rd}$ International Conference on Machine Learning*, Seoul, South Korea. PMLR 306, 2026. Copyright 2026 by the author(s).

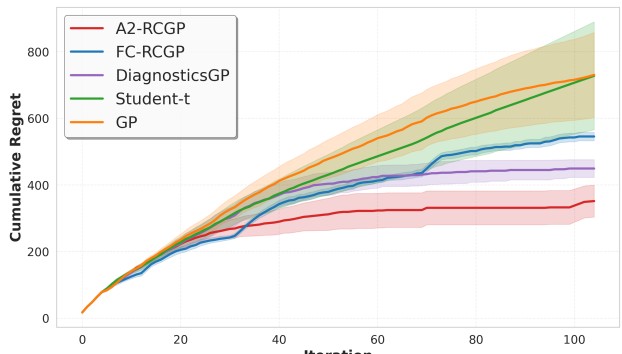

*Figure 1.* Cumulative regret for different BO algorithms on a corrupted Forrester example. Approaches based on standard GPs or Student-t processes perform worse than RCGP-based methods. DiagnosticsGP performs well on this example, but does not satisfy any regret bounds.

The theoretical guarantees and practical success of standard BO, however, often rely on idealized assumptions about the nature of the objective function or its observations, such as assuming Gaussian noise (Srinivas et al., 2010). In real-world experimental environments, however, these assumptions are frequently violated. Experimental setups can fail, petri dishes may be exposed to contamination, sensors malfunction, and high-throughput processes can yield unreliable data (Wakabayashi et al., 2022; Iwazaki et al., 2025). These scenarios generate extreme outliers that can skew the posteriors arbitrarily, and derail BO.

To address this, a significant body of work has focused on robust BO, and largely divides into three sub-streams of work. The first of these addresses robustness against GP model misspecification: the true underlying function may not perfectly align with the assumptions of the GP model, and may lie outside the assumed RKHS (Bogunovic & Krause, 2021; Wang et al., 2018). A second direction are BO algorithms whose optima remain stable even under input perturbations (Christianson & Gramacy, 2024; Ullah et al., 2021; Bogunovic et al., 2018). The last direction is geared towards dealing with observational noise, and is most closely related to what we shall do in the remainder. Here, the most common approach involves heavy-tailed models like Student-t processes (Shah et al., 2014; Jylänki et al., 2011). Other ideas rely on bias trimming techniques from robust regression (Chi et al., 2024) or min-max type

formulations guarding against the worst case (Tay et al., 2022). While often effective for moderate deviations, these methods tend to struggle with severe contamination, often sacrifice the analytical tractability of a conjugate GP update, and generally do not provide regret bounds.

Motivated by providing meaningful regret bounds for this third setting, several contributions have developed algorithms with guarantees under the additive corruption model (Bogunovic et al., 2018; Gupta et al., 2019). Originating in the stochastic bandit setting (Lykouris et al., 2018), this line of work assumes that while the adversary may inject a corruption of size $c_t$ at each time step, the cumulative corruption is bounded by some known constant $C$ so that $\sum_{t=1}^{T} |c_t| \leq C$. The resulting algorithms provide regret bounds which scale with the size of the total corruption budget $C$ (Kirschner & Krause, 2021; Bogunovic et al., 2020; 2022). Though useful if this corruption model is a good description of reality, assuming $C$ to be known and finite is brittle when dealing with outliers in many real-world settings. Indeed, a *single* catastrophic corruption is enough to exhaust the corruption budget the existing algorithms assume.

In this paper, we derive algorithms that can deal with outliers of arbitrary magnitude. To do so, we build on the literature on generalised Bayesian methods (Bissiri et al., 2016; Knoblauch et al., 2019), which replace Bayes updates with suitable generalisations (see also Dellaporta et al., 2022; Wild et al., 2023; McLatchie et al., 2025; Shen et al., 2025). While this framework has been recently applied to BO to handle model misspecification via tempered posteriors (Li & Luo, 2026), we instead leverage its capacity to handle outliers with possibly infinite magnitude (e.g. Ghosh & Basu, 2016; Knoblauch et al., 2018; Schmon et al., 2020; Duran-Martin et al., 2024; Frazier et al., 2025). For GPs, this has led to the development of Robust Conjugate Gaussian Processes (RCGPs) (Altamirano et al., 2024; Laplante et al., 2025), which *not only* retain robustness to outliers *but also* the conjugacy and associated closed forms of GPs. Herein, we propose using RCGP with a UCB-based acquisition function. This yields two RCGP-UCB algorithms which are resistant against up to $\mathcal{O}(T^{1/4})$ corruptions of infinite magnitude while retaining sublinear regret bounds. To the best of our knowledge, this makes them the first BO algorithms with meaningful regret bounds in the presence of possibly infinite magnitude corruptions. Further, this comes at zero cost to performance in the corruption-free setting: here, RCGP-UCB achieves the same theoretical bounds and empirical performance as standard GP-UCB. This is in stark contrast to previous heuristic approaches that, despite often performing well empirically, lacked clear formal guarantees. One such method is the algorithm proposed by Martinez-Cantin et al. (2018), which we will refer to as DiagnosticsGP.

**Contributions.** We contribute on several levels: first, we introduce a new adversary whose corruptions are limited in the number of data points, but of possibly infinite magnitude. Second, we modify the basic RCGP implementation to ensure that the behaviour of RCGP-UCB matches that of GP-UCB in the absence of outliers. Third, we leverage our developments to introduce the FC-RCGP-UCB and A2-RCGP-UCB algorithms. We prove that they achieve sublinear regret in the presence of up to $O(T^{1/4})$ and $O(T^{1/7})$ unbounded corruptions, respectively. Notably, our experiments demonstrate effective robustness against a larger $O(T^{1/3})$ budget, suggesting that these theoretical rates are conservative and that the algorithms can handle higher corruption frequencies in practice. To our knowledge, they are the first BO algorithms with provable sublinear regret in the presence of infinite-magnitude corruptions without incurring any cost in the well-specified setting.

## 2. Background

We operate under the standard assumptions established in Srinivas et al. (2010). We defer their full statement to Appendix B, but note that they accommodate any of the following three settings: (1) the objective function $f$ is sampled from a Gaussian Process (GP) prior over a finite domain; (2) $f$ is sampled from a GP prior over a compact and convex domain; or (3) $f$ resides in a Reproducing Kernel Hilbert Space (RKHS) with a bounded norm ($\|f\|_k \leq B$). Irrespective of the exact setting, we will require the kernel function to be bounded, such that $\sup_{\boldsymbol{x} \in \mathcal{X}} k(\boldsymbol{x}, \boldsymbol{x}) \leq \kappa$.

### 2.1. Bayesian Optimization with GPs

BO sequentially optimises an unknown black-box function $f : \mathcal{X} \to \mathbb{R}$, where the domain $\mathcal{X} \subset \mathbb{R}^d$ is compact. This proceeds in rounds $t = 1, \ldots, T$, with an agent querying a point $\boldsymbol{x}_t \in \mathcal{X}$ for each round, and then observing a corresponding value $y_t$. The objective is to minimize the cumulative regret over $T$ rounds: $R_T = \sum_{t=1}^{T} (f(\boldsymbol{x}^*) - f(\boldsymbol{x}_t))$, where $\boldsymbol{x}^* = \arg\max_{\boldsymbol{x} \in \mathcal{X}} f(\boldsymbol{x})$ is the global optimum.

**GPs.** We model the objective $f$ using a Gaussian Process (GP) prior, denoted $f \sim \mathcal{GP}(m(\boldsymbol{x}), k(\boldsymbol{x}, \boldsymbol{x}'))$ with zero mean, $m(\boldsymbol{x}) = 0$. Observations are assumed to be generated as $y_t = f(\boldsymbol{x}_t) + \epsilon_t$, where $\epsilon_t \sim \mathcal{N}(0, \sigma_{\text{noise}}^2)$ denotes independent Gaussian noise. Here, $\boldsymbol{x}_t = \{\boldsymbol{x}_1, \ldots, \boldsymbol{x}_t\}$ denotes the set of query points and $Y_t = \{y_1, \ldots, y_t\}$ the corresponding observations. Conditional on the data set $\mathcal{D}_t = (\boldsymbol{x}_t, Y_t)$, the GP posterior remains Gaussian, and is fully characterized by the posterior mean $\mu_t(\boldsymbol{x})$ and variance $\sigma_t^2(\boldsymbol{x})$ given by

$$\mu_t(\boldsymbol{x}) = \boldsymbol{k}_t(\boldsymbol{x})^T (\boldsymbol{K}_t + \sigma_{\text{noise}}^2 \boldsymbol{I})^{-1} \boldsymbol{y}_t,$$
$$\sigma_t^2(\boldsymbol{x}) = k(\boldsymbol{x}, \boldsymbol{x}) - \boldsymbol{k}_t(\boldsymbol{x})^T (\boldsymbol{K}_t + \sigma_{\text{noise}}^2 \boldsymbol{I})^{-1} \boldsymbol{k}_t(\boldsymbol{x}).$$

Here, $\boldsymbol{y}_t$ is the vector of observations, $\boldsymbol{K}_t$ denotes the kernel matrix evaluated on $\boldsymbol{x}_t$, and $\boldsymbol{k}_t(\boldsymbol{x})$ the vector of covariances between $x$ and $\boldsymbol{x}_t$.

**GP-UCB.** The popular GP-UCB algorithm (Srinivas et al., 2010) balances exploration and exploitation via optimism: it chooses the next $\boldsymbol{x}_t$ by maximising the Upper Confidence Bound (UCB):

$$\boldsymbol{x}_t = \arg\max_{\boldsymbol{x} \in \mathcal{X}} \mu_{t-1}(\boldsymbol{x}) + \sqrt{\beta_t'}\sigma_{t-1}(\boldsymbol{x}).$$

The parameter sequence $\beta_t'$ typically scales logarithmically with $t$, and is carefully chosen to endow the algorithm with a high-probability guarantee which ensures that the regret grows sublinearly as $\mathcal{O}(\sqrt{T\beta_T'\gamma_T})$, where $\gamma_T$ is the maximum information gain, which quantifies the maximum reduction in uncertainty about the true function $f$ after $T$ evaluations.

### 2.2. Frequency-Constrained Corruption

Existing adversarial BO algorithms guard against additive corruption so that

$$y_t = f(\boldsymbol{x}_t) + \epsilon_t + c_t \tag{1}$$

for a sequence of corruptions $c_t$ that are bounded in their total budget by some $C > 0$ so that $\sum_{t=1}^{T} |c_t| \leq C$. This constraint renders the BO algorithm's guarantees brittle: a single outlier of sufficient magnitude will derail it. This fragility is structural: a single corruption of significant magnitude can shift the posterior mean arbitrarily far from the true function. We provide a formal proof of this failure mode in Appendix I. To address this limitation, we introduce an adversarial framework inspired by the Huber contamination model (see Huber, 1981).

**Definition 2.1** (Frequency-Constrained Corruption). The adversary selects corruptions $c_t \in \mathbb{R}_\cup \{\infty\}$ after observing $\boldsymbol{x}_t$. The magnitudes $|c_t|$ are unbounded, but their *frequency* is bounded: the adversary can at most corrupt $T_\mathrm{c} = |\{t \in \{1, \ldots, T\} : c_t \neq 0\}|$ data points.

By permitting $|c_t| = \infty$, the adversary can force the algorithm to observe any arbitrary value $y_t \in \mathbb{R}$. This strengthens traditional corruption models, and effectively captures the kind of catastrophic but infrequent events that result in extreme outliers.

### 2.3. Robust Conjugate Gaussian Processes

To design a BO algorithm capable of dealing with corruptions of possibly infinite magnitude, we build on the Robust Conjugate Gaussian Process (RCGP) (Altamirano et al., 2024; Laplante et al., 2025). RCGPs retain closed forms of their posterior updates, and are provably robust against

outliers in a supervised learning setting. The key to this is a weight function $w$, which dampens the influence that anomalous observations have on posterior inferences. The RCGP posterior mean $\mu_t^\mathrm{R}(\boldsymbol{x})$ and variance $\sigma_t^\mathrm{R}(\boldsymbol{x})^2$ are given by

$$\mu_t^\mathrm{R}(\boldsymbol{x}) = \boldsymbol{k}_t(\boldsymbol{x})^T(\boldsymbol{K}_t + \sigma_\mathrm{noise}^2\boldsymbol{J}_w)^{-1}(\boldsymbol{y}_t - \boldsymbol{m}_w), \tag{2}$$
$$\sigma_t^\mathrm{R}(\boldsymbol{x})^2 = k(\boldsymbol{x},\boldsymbol{x}) - \boldsymbol{k}_t(\boldsymbol{x})^T(\boldsymbol{K}_t + \sigma_\mathrm{noise}^2\boldsymbol{J}_w)^{-1}\boldsymbol{k}_t(\boldsymbol{x}), \tag{3}$$

and differ from vanilla GPs through two terms dependent on the weights $w_i = w(\boldsymbol{x}_i, y_i)$:

$$\boldsymbol{J}_w = \mathrm{diag}\left(\frac{\sigma_\mathrm{noise}^2}{2w_i^2}\right), \quad \boldsymbol{m}_w = \left[\sigma_\mathrm{noise}^2\nabla_y \log(w_i^2)\right]_{i=1}^{t}.$$

As shown by Altamirano et al. (2024), the RCGP provides Huber robustness whenever the weight function decreases sufficiently fast so that $\sup_y |y| \cdot w(x,y)^2 < \infty$. This holds for the Inverse Multi-Quadric (IMQ) kernel (Javaran & Khaji, 2012), which is popular in the context of robustness (e.g. Matsubara et al., 2022; 2024). For a centering function $g : \mathcal{X} \to \mathcal{Y}$ and a scalar $c > 0$, it is given by:

$$w_\mathrm{IMQ}(x,y) \propto \left(1 + \frac{(y - g(\boldsymbol{x}))^2}{c^2}\right)^{-\frac{1}{2}}. \tag{4}$$

## 3. Weights and Zero-Cost Robustness

While we wish to design a robust BO algorithm, we also want a "best-of-both-worlds" performance guarantee: we want an algorithm that is not only resilient to outliers of possibly infinite magnitude, but also retains the same level of statistical efficiency as standard GP-UCB when there are no corruptions—a property we refer to as *zero-cost robustness* in the remainder. The key to achieving this lies in carefully controlling the IMQ weighting function in Eq. (4) to enforce a form of optimism about the data: otherwise, whenever the centering function $g$ deviates substantively from the true objective $f$, the IMQ function will tend to be pessimistic about the reliability of non-outliers with large residual values $|y_t - g(\boldsymbol{x}_t)|$, down-weight them, and lose statistical efficiency relative to a GP.

To enforce optimism and achieve zero-cost robustness, we adapt the key idea of Huber losses (Huber, 1992): by introducing a threshold $L$ that quantifies which residuals are deemed reasonable, we can treat any observation *exactly* as a standard GP would so long as its residual stays below the threshold value. This results in a plateau IMQ (P-IMQ) weight, see Figure 2.

**Definition 3.1** (P-IMQ Weight). For $W = \frac{\sigma_\mathrm{noise}}{\sqrt{2}}$, centering function $g$, and $L > 0$, the P-IMQ is

$$w_{L,g}(x,y) = \begin{cases} W & |y - g(\boldsymbol{x})| \leq L \\ W\left(1 + \frac{(|y - g(\boldsymbol{x})| - L)^2}{c^2}\right)^{-\frac{1}{2}} & \text{otherwise.} \end{cases}$$

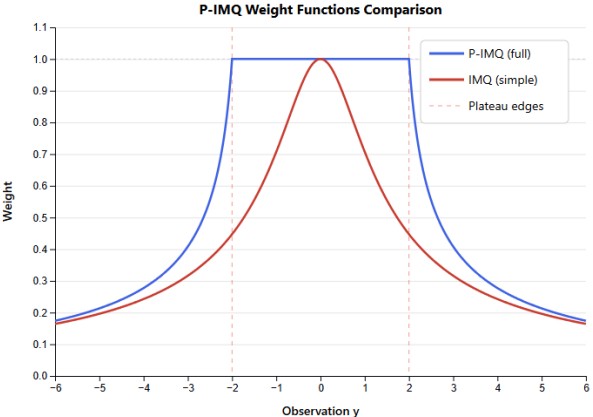

*Figure 2.* Comparison of the IMQ and P-IMQ weight functions. We use $W = 1$, $c = 1$, $g(\boldsymbol{x}) = 0$, and $L = 2$.

The P-IMQ weight function owes its name to its shape: when plotted, it gives rise to a symmetric plateau centered at $g(\boldsymbol{x})$ and of length $2L$. Figure 2 shows a comparison of the IMQ and P-IMQ weight functions. Once residuals fall outside of the plateau, we begin to suspect them of being outliers, and let the P-IMQ function monotonically decrease outside of the plateau region. Observations whose residuals fall within this plateau however are treated as fully trustworthy, and processed in the *exact* same way as a standard GP would. To see this, compare the GP and RCGP update equations and note that for all observations with residuals smaller than $L$, it holds that $\boldsymbol{J}_w = \text{diag}(\frac{\sigma_{\text{noise}}^2}{2W^2}) = \text{diag}(\frac{\sigma_{\text{noise}}^2}{2(\sigma_{\text{noise}}^2/2)}) = \boldsymbol{I}$ and $m_w = \sigma_{\text{noise}}^2 \nabla_y \log(w^2) = 0$. In fact, the condition that all residuals corresponding to uncorrupted observations lie within the plateau region with high probability is instrumental for establishing the zero-cost probability of RCGP-UBC, and we will often refer to it as the **plateau condition** in the remainder.

## 4. Algorithms and Analysis

To ensure the plateau condition holds, we have two choices: we can choose $L$ sufficiently large, or we can design a centering function $g$ that is a good approximation of $f$. While it is generally easier to fix the centering and just choose $L$ sufficiently large based on theoretical arguments, this could make the algorithm's guarantees vacuous as our regret bounds scale linearly in $L$ (see Appendix G). Alternatively, we could thus try and instead choose $g$ in an adaptive manner. In fact, in the supervised learning setting, Laplante et al. (2025) argue that it is preferable to take $g(\boldsymbol{x}) = \mu_{t-1}^R(\boldsymbol{x})$ as the posterior mean—rather than a fixed function. Unfortunately, adaptively setting $g$ introduces a potential vulnerability: the adversary can strategically inject corruptions to manipulate $g$, and cause a loss of efficiency by forcing genuinely uncorrupted observations to fall outside the plateau

and be erroneously down-weighted. This would be undesirable, and could destabilise the learning process.

In the remainder of this section, we first provide the acquisition function underlying RCGP-UCB, and then propose two algorithms that take different approaches to managing the inherent tension between algorithmic stability and adaptivity: first, we introduce the Fixed-Center RCGP-UCB algorithm, which uses theoretical bounds to set $L$ sufficiently large. This yields the usual sublinear regret guarantees so long as the number of outliers are at most $\mathcal{O}(T^{1/4})$. Next, we derive guarantees for the Anchor-Adapt RCGP-UCB algorithm, which dynamically adjusts the centering function $g$ but remains stable as long as the number of outliers are at most $\mathcal{O}(T^{1/7})$.

### 4.1. RCGP-UCB Acquisition Functions

Our acquisition function design is grounded in an analysis of the deviation between the robust posterior mean $\mu_{t-1}^R(\boldsymbol{x})$ computed using all observations and the idealized posterior mean $\mu_{\text{uc},t-1}(\boldsymbol{x})$ obtained from fitting a standard GP to the subset of all *uncorrupted* data points up to time point $t-1$. Whenever the plateau condition holds, a key result bounds this deviation (see Lemma D.3 in Appendix D), and shows that

$$|\mu_{t-1}^R(\boldsymbol{x}) - \mu_{\text{uc},t-1}(\boldsymbol{x})| \leq C_w \sqrt{T_c} \sigma_{\text{uc},t-1}(\boldsymbol{x});$$

with $C_w$ denoting a constant depending on the weight function $w$ (see Appendix G), $\sigma_{\text{uc},t-1}^2(\boldsymbol{x})$ the posterior variance based on all uncontaminated data points up to time point $t-1$, and $T_c$ as in Definition 2.1.

This deviation bound allows the construction of an inflated Upper Confidence Bound (UCB): adding the standard confidence interval for the idealized posterior $\mu_{uc,t-1}(\boldsymbol{x})$—which uses the multiplier $\sqrt{\beta_t'}$—with the deviation bound presented above, we derive a robust confidence interval around the true function $f(\boldsymbol{x})$ as

$$|f(\boldsymbol{x}) - \mu_{t-1}^R(\boldsymbol{x})| \leq \underbrace{(\sqrt{\beta_t'} + C_w \sqrt{T_c})}_{=:\sqrt{\beta_t},\text{ called 'Robust Multiplier'}} \sigma_{\text{uc},t-1}(\boldsymbol{x}).$$

A practical limitation of above confidence interval is its dependence on $\sigma_{\text{uc},t-1}(\boldsymbol{x})$ which is unobservable. To address this, we utilize the bound derived in Lemma D.6 (Appendix D), which relates the unobservable variance to the observable RCGP variance $\sigma_{t-1}^R(\boldsymbol{x})$ via a multiplicative factor $\Psi((t-1)_c) \triangleq \sqrt{1 + \frac{(t-1)_c \kappa}{\sigma_{noise}^2}(1 + \frac{(t-1)_c \kappa}{\sigma_{noise}^2})}$, where $(t-1)_c$ is the number of contaminated data points up to time $t-1$. Since $\Psi(.)$ is non-decreasing, using the total contamination budget $T_c$ yields a fully observable confidence bound:

$$|f(\boldsymbol{x}) - \mu_{t-1}^R(\boldsymbol{x})| \leq \sqrt{\beta_t}\Psi(T_c)\sigma_{t-1}^R(\boldsymbol{x})$$

**Algorithm 1** FC-RCGP-UCB

1: **Input:** $k$, $\kappa$, $B_f$, $T$, $T_c$, $\delta$
2: Compute $N_T(\delta/3)$
3: $L_T \leftarrow \sqrt{B_f \kappa} + N_T(\delta/3)$.
4: Compute $C_{w,T}$ using $L_T$
5: **for** $t = 1$ to $T$ **do**
6:     $\beta_t \leftarrow (\sqrt{\beta_t'(\delta/3)} + C_{w,T}\sqrt{T_c})^2$
7:     $\boldsymbol{x}_t \leftarrow \arg\max_{x \in \mathcal{X}} \mu_{t-1}^R(\boldsymbol{x}) + \sqrt{\beta_t}\Psi(T_c)\sigma_{t-1}^R(\boldsymbol{x})$
8:     Observe $y_t$
9:     RCGP-update$(g = 0, L_T, \boldsymbol{x}_t, y_t)$
10: **end for**

Consequently, our practical acquisition function maximizes

$$\mu_{t-1}^R(x) + \sqrt{\beta_t}\Psi(T_c)\sigma_{t-1}^R(x)$$

### 4.2. Algorithms

#### 4.2.1. FIXED-CENTER RCGP-UCB

The Fixed-Center-RCGP-UCB (FC-RCGP-UCB) fixes $g$ to be the zero prior mean. To ensure that the plateau condition holds, it thus has to employ a fixed but sufficiently large threshold value $L = L_T$, which is derived from known bounds on the function norm and the noise distribution as $L_T = \sqrt{B_f \kappa} + N_T(\delta)$, where $N_T(\delta)$ represents a high-probability bound on the noise magnitude, and $\sqrt{B_f \kappa}$ is a high-probability upper bound on $f$ (see Appendix B). We illustrate the resulting procedure in Algorithm 1, which outputs a sequence of queries that satisfies various theoretical guarantees with probability $1 - \delta$ for some user-specified confidence level $\delta > 0$. Here, the sub-routine RCGP-update denotes posterior mean and variance updates as in Eqs. (2) and (3). Note that the algorithm depends on having explicit knowledge of the adversary's corruption budget $T_c$, which is unknown in most practical settings. While this quantity can to be estimated, the ablation studies in Appendix H.3 show that ignoring the observability penalty term $\Psi(T_c)$ works well in practice.

#### 4.2.2. ANCHOR-ADAPT RCGP-UCB

Our second algorithm, Anchor-Adapt RCGP-UCB (A2-RCGP-UCB), addresses a limitation of the fixed-center design above. The robustness of FC-RCGP-UCB comes at an efficiency cost when its centering function is misspecified. The P-IMQ plateau treats an observation exactly as a standard GP would whenever its residual relative to the center $g$ stays within the half-width $L$, and FC-RCGP-UCB fixes $g(\boldsymbol{x}) \equiv 0$. When the problem is well specified this is harmless, but in general $f$ may live far from zero, and the experimenter must then choose $L$ without knowing how far. Misspecifying the width is costly in either direction. Set $L$ too small relative to where $f$ lives and many genuine observations fall *outside* the plateau, where they are themselves down-weighted, so the model extracts little signal from its data and convergence slows down. Set $L$ too wide and the plateau down-weights fewer corruptions, eroding robustness. A fixed center thus forces a delicate width calibration and, more fundamentally, trades robustness to model misspecification for adversarial robustness.

A natural remedy is to let the center track the data. Studying RCGP in spatio-temporal settings, Laplante et al. (2025) recommend centering the plateau at the previous robust posterior mean, $g(\boldsymbol{x}) = \mu_{t-1}^R(\boldsymbol{x})$, which largely removes the misspecification above. Under an adversary, however, this reintroduces fragility. With a fixed center, the adversary's only lever is the posterior mean and variance, whose perturbation the robust weighting is explicitly designed to absorb. Once the center is data-driven, the weight function itself depends on the corruptible data, so the adversary can instead attack the very mechanism that confers robustness: by shifting $\mu_{t-1}^R$, it moves the plateau and redefines which residuals count as reasonable, admitting corrupted points while pushing genuine observations outside, where they are down-weighted. The two extremes thus expose a fundamental tension: a fixed center is robust but suffers from misspecification, while a fully data-driven center adapts but is too sensitive to corruption. An effective method must balance the two intelligently.

The Anchor-Adapt RCGP-UCB (A2-RCGP-UCB) algorithm strikes this balance by running two RCGP models in parallel:

- an **anchor** model $M_{\text{⚓}}$, configured exactly as FC-RCGP-UCB with the fixed center $g_{\text{⚓}}(\boldsymbol{x}) = 0$ and a half-width $L_{\text{⚓},T}$. Because its center does not depend on the observations, the adversary cannot manipulate its weights, so its posterior mean $\mu_{\text{⚓},t-1}$ is a provably adversially-robust estimate of $f$;

- an **adaptive** model $M_{\text{⚙}}$, which centers its plateau at the anchor's estimate, $g_{\text{⚙}}(\boldsymbol{x}) = \mu_{\text{⚓},t-1}(\boldsymbol{x})$. Because this center tracks $f$, the adaptive model is far less misspecified than $g \equiv 0$ and recovers the efficiency FC-RCGP-UCB gives up. It supplies the posterior mean and variance that drive the acquisition rule.

Because the adaptive model is re-centered by the *robust* anchor rather than by its own corruptible fit, the adversary can reach the center only through the dampened anchor, which breaks the feedback loop. A2-RCGP-UCB thus occupies a principled middle ground: it adapts to misspecification while retaining provable robustness, at the cost of a weaker theoretical corruption budget tolerance. We summarise the full procedure in Algorithm 2.

---

**Algorithm 2** A2-RCGP-UCB

---

1: **Input:** $k, \kappa, B_f, T, T_c, \delta$.
2: Compute $N_T(\delta/3)$; $L_{\maltese,T} \leftarrow \sqrt{B_f \kappa} + N_T(\delta/3)$.
3: Compute $C_{w,\maltese,T}$ using $L_{\maltese,T}$
4: Compute $\beta_{\maltese,T}$ using $C_{w,\maltese,T}$
5: $L_{\sword,T} \leftarrow \sqrt{\beta_{\maltese,T}} \sqrt{\kappa} \Psi(T_c) + N_T(\delta/3)$.
6: Compute $C_{w,\sword,T}$ using $L_{\sword,T}$
7: **for** $t = 1$ to $T$ **do**
8: $\quad \beta_{\sword,t} \leftarrow (\sqrt{\beta'_t(\delta/3)} + C_{w,\sword,T}\sqrt{T_c})^2$
9: $\quad \boldsymbol{x}_t \leftarrow \arg\max_{x \in \mathcal{X}} \mu_{\sword,t-1}(\boldsymbol{x}) + \sqrt{\beta_{\sword,t}}\Psi(T_c)\sigma^{\mathrm{R}}_{t-1}(\boldsymbol{x})$
10: $\quad$ Observe $y_t$
11: $\quad$ RCGP-update$(M_{\maltese}, g_{\maltese} = 0, L_{\maltese,T}, \boldsymbol{x}_t, y_t)$
12: $\quad$ RCGP-update$(M_{\sword}, g_{\sword} = \mu_{\maltese,t-1}, L_{\sword,T}, \boldsymbol{x}_t, y_t)$
13: **end for**

---

### 4.3. Theoretical Guarantees

Throughout our results, we use $R_T = \mathcal{O}(h(T))$ to mean that $R_T$ grows in $T$ as $h(T)$, and $\tilde{\mathcal{O}}(h(T))$ to ignore polylogarithmic factors of $h(T)$. For example, $\mathcal{O}(T \log(T)) = \tilde{\mathcal{O}}(T)$. First, we prove the zero-cost robustness property for both methods: if there are no corruptions so that $T_c = 0$, we obtain the standard GP-UCB regret bound.

**Theorem 4.1** (Zero-Cost Robustness). *If $T_c = 0$, FC-RCGP-UCB and A2-RCGP-UCB achieve the same asymptotic regret as GP-UCB: with probability at least $1 - \delta$, their cumulative regret is bounded by*

$$R_T = \mathcal{O}(\sqrt{T\beta'_T \gamma_T}).$$

The proof is simple: if $T_c = 0$, the robust multipliers revert to those of GP-UCB, and the plateau condition is satisfied (see Appendix F).

Next, we study regret guarantees under frequency-constrained corruption as in Definition 2.1. The result for FC-RCGP-UCB is more direct, as the adversary cannot manipulate its weight function $w$ via $g$.

**Theorem 4.2** (Regret for FC-RCGP-UCB). *For frequency-constrained corruption and $\gamma_T$ the maximum information gain, the cumulative regret of FC-RCGP-UCB is bounded with probability at least $1 - \delta$ by*

$$R_T = \tilde{\mathcal{O}}\left(\Psi(T_c)\left(1 + \sqrt{T_c}\right)\sqrt{\beta'_T T(\gamma_T + T_c)}\right).$$

While FC-RCGP-UCB relies on a fixed centering function, A2-RCGP-UCB adapts it over time. Though this results in a weaker regret bound than the one obtained for FC-RCGP-UCB, our numerical experiments in later sections suggest that this is likely to be a limitation of the proof technique.

**Theorem 4.3** (Regret for A2-RCGP-UCB). *For frequency-constrained corruption and $\gamma_T$ the maximum information gain, the cumulative regret of A2-RCGP-UCB is bounded with probability at least $1 - \delta$ by*

$$R_T = \tilde{\mathcal{O}}\left(\left(1 + T_c\Psi(T_c)^2\right)\sqrt{\beta'_T T(\gamma_T + T_c)}\right).$$

The proofs of Theorems 4.2 and 4.3 can be found in Appendix D. We note that A2-RCGP-UCB achieves both adaptive centering and robustness in the presence of up to $\mathcal{O}(T^{1/7})$ corruptions with possibly infinite magnitude.

**Conditions for Sublinear Regret.** We analyze the corruption budgets $T_c = \mathcal{O}(T^\alpha)$ permissible for sublinear regret. Ideally, if the uncorrupted variance $\sigma^2_{uc,t}$ were directly observable (eliminating the penalty $\Psi(T_c)$), our algorithms would tolerate significantly higher corruption rates: FC-RCGP-UCB achieves sublinear regret for $\alpha < 1/2$ and A2-RCGP-UCB for $\alpha < 1/3$ using an RBF kernel. However, due to the need to upper-bound the unknown uncorrupted variance with the observable RCGP variance (Lemma D.6), our provable rates are more conservative: $\alpha < 1/4$ and $\alpha < 1/7$ respectively. Our experiments suggest these theoretical penalties are likely loose artifacts of the proof technique, as the algorithms empirically handle contamination budgets of $\mathcal{O}(T^{1/3})$. A full comparison is provided in Appendix E.

## 5. Experiments

We empirically evaluate the performance of FC-RCGP-UCB and A2-RCGP-UCB on several benchmarks. Our experiments are designed to validate the theoretical guarantees of zero-cost robustness (Theorem 4.1) and the algorithms' resilience against frequency-constrained, infinite-magnitude adversarial corruptions (Theorems 4.2 and 4.3).

### 5.1. Implementation

Our algorithms are implemented in Python, and use the BoTorch library (Balandat et al., 2020) for its BO infrastructure, and GPyTorch (Gardner et al., 2021) for Gaussian Process modeling. Our code integrates the RCGP model directly, and can be found `here`.

To optimize the kernel hyperparameters, we deviate from the standard approach of maximising marginal likelihood approach: as noted when RCGPs were first introduced (Altamirano et al., 2024), Gibbs posteriors generally do not admit valid marginal likelihoods, and maximising their generalised extensions of marginal likelihoods for hyperparameter selection is ill-posed. We address this by instead following the recommendations in Laplante et al. (2025), and optimise the hyperparameters—in particular the noise variance

and kernel hyperparameters—by minimising a weighted Leave-One-Out Cross-Validation (LOO-CV) objective.

While our theoretical analysis assumed prior knowledge of the number of corruptions $T_c$ to set the sequence of robust confidence multipliers $\{\beta_t\}_{t=1}^T$, the number $T_c$ is often unknown in practice. In practice, one can solve this problem either by bounding $T_c$ with a conservatively large number, or by estimating $T_c$ adaptively. In our experiments, we choose the latter option: both FC-RCGP-UCB and A2-RCGP-UCB estimate of $T_c$ as the count of observations outside the plateau corresponding to their P-IMQ weighting function. In all of our empirical results, this strategy resulted in good empirical performances.

### 5.2. Baselines

Throughout our experiments, we compare our new proposals against three established baselines. The first of these is **GP-UCB**—one of the most important BO algorithms originally derived in Srinivas et al. (2010). While this algorithm is brittle against outliers and corruption, it provides a useful benchmark for efficiency in the uncorrupted setting. The second comparator will be **Student-t Process UCB:** a modification of the GP-UCB algorithm which replaces the Gaussian process prior on the objective function with a Student-t process. Shah et al. (2014) derived closed-form formulas for the posterior mean and variance for this algorithm, and showed strong empirical using the Expected Improvement (EI) acquisition function. Lastly, we compare against a heuristic approach (**DiagnosticsGP**) proposed by Martinez-Cantin et al. (2018) which does not satisfy any known regret bounds, and attempts to discard outliers before fitting a standard GP on the remaining data points. One way of identifying outliers recommended by Martinez-Cantin et al. (2018) is to fit a GP with a Student-t likelihood, and to then discard observations whose likelihood values are very small. Notably, this process can be thought of as a limiting case of the RCGP with the P-IMQ weight function for $c \to 0$, and where $L$ is set dynamically—for example through a GP that is fitted with Student-t likelihoods. While Martinez-Cantin et al. (2018) recommended using a Laplace approximation to fit the GP with Student-t likelihoods, we found that variational approximations significantly improved stability; and implemented the algorithm this way instead.

### 5.3. Forrester Function Experiments

We begin by evaluating all algorithms on the Forrester function (Forrester et al., 2008), a common benchmark in BO. The objective function is observed with independent Gaussian noise of variance $\sigma_{\text{noise}}^2 = 1$. We conduct 10 independent experimental runs, each initialized with a unique random seed. Each run starts with five common, uncor-

rupted observations from a Sobol sequence which serve as a common starting point. Each algorithm then proceeds for 30 iterations in an uncorrupted setting and 100 iterations in a corrupted setting, defined by $T_c = \mathcal{O}(T^{1/3})$. Figures 1 and 3 present aggregated results for the corrupted and uncorrupted settings respectively. Solid lines represent the average cumulative regret across the different seeds while shaded areas represent one standard error away from the mean.

**Zero-Cost Robustness.** To validate Theorem 4.1, we first run the experiment without any corruptions ($T_c = 0$). As shown in Figure 3, both FC-RCGP-UCB and A2-RCGP-UCB achieve cumulative regret virtually identical to GP-UCB, and even seem to perform slightly better than Student-t Process UCB and the DiagnosticsGP approach. This is exactly in line with Theorem 4.1, and provides empirical illustration that the robustness mechanisms introduced by the RCGP with P-IMQ weights does not incur any cost in efficiency when corruptions are absent.

**Adversarial Corruption.** Next, we introduce a greedy, clairvoyant adversary that knows the true optimum $x^*$. The adversary's deterministic policy is based on the Euclidean distance to the optimum: if a queried point $x$ satisfies $\|x - x^*\| < 0.2$, it corrupts the observation to a fixed low value, whereas if $\|x - x^*\| > 0.5$, it corrupts it to a fixed high value. The adversary is greedy as it applies this policy to the earliest queried points that meet either condition until its corruption budget is exhausted. This strategy is potent because early interventions have a high potential to derail the BO algorithm before a robust surrogate model is formed. The results are presented in Figure 1. As expected, GP-UCB suffers the highest cumulative regret. The adaptive RCGP-based BO algorithm A2-RCGP-UCB demonstrates superior performance: it effectively ignores extreme outliers, and maintaining low cumulative regret throughout the optimization process. For individual runs with extremely large corruptions, we noticed that the Diagnosis-GP baseline could sometimes outperfom A2-RCGP-UCB. In fact, this is perhaps unsurprising: Diagnosis-GP effectively sets the weight to zero past a threshold and thus excludes extreme outliers completely. While this has the effect of introducing more robustness against extreme outliers than A2-RCGP-UCB, it also leads to a more pronounced loss of efficiency when an observation is mistakenly classified as an outlier. In fact, this is why A2-RCGP-UCB can outperform Diagnosis-GP in the uncorrupted setting depicted in Figure 3.

### 5.4. Hyperparameter Optimisation

Next, we evaluate our approach on a hyperparameter optimisation task for training a ResNet classifier (He et al., 2015) on the CIFAR-10 dataset (50,000 training samples, 10,000 validation samples). The objective is to maximize valida-

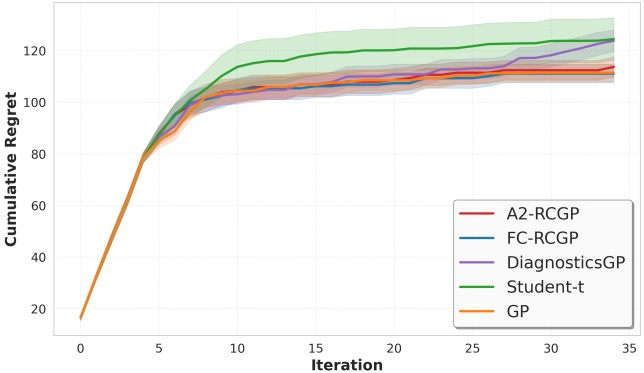

*Figure 3.* Zero-cost robustness validation: Cumulative regrets of the different BO algorithms in the uncorrupted Forrester experiment ($T_c = 0$).

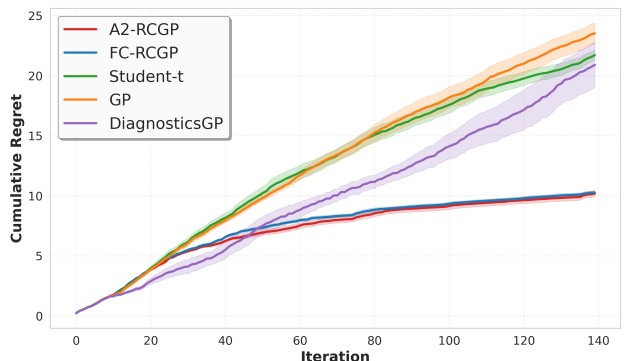

*Figure 4.* Cumulative regrets for different BO algorithms applied to hyperparameter optimization on CIFAR with simulated training crashes.

tion accuracy by optimizing the ResNet's learning rate and weight decay, subject to an adversary capable of crashing the machine during evaluation. These failures are mapped to a negative outlier value at $-2$, and induced with a frequency of $T_c = T^{1/3}$. Since the optimal point is unknown, we chose an adversary that greedily chooses to spend its corruption budget as swiftly as possible regardless of the queried set of hyperparameters. We define the regret as the difference in accuracy, as a number in the range $[0, 1]$, between the current ResNet configuration and the best obtained configuration across all models. We run all BO algorithms for 140 steps using the same random seed, then repeat the experiment 10 times across different seeds. Figure 4 shows the cumulative regrets of the different BO algorithms where solid lines represent the mean, and shaded regions represent the standard error.

As we can see, standard BO in this setting is often too conservative, and avoids optimal regions due to the risk of catastrophic failure. In contrast, as shown in Figure 4, outlier-robust methods such as RCGP-based approaches and Diagnosis-GP successfully navigate this challenging scenario. By ignoring crashed evaluation runs, they ignore the adversarial impact of the failure mechanism on the underlying performance signal. This allows for a more effective exploration of hyperparameter configurations, and demonstrates RCGP's capability for safe optimisation in environments susceptible to failure.

### 5.5. Lunar-Lander-3

Lastly, we assess performance in a high-dimensional setting using the Lunar-Lander-3 reinforcement learning environment (Towers et al., 2024). The task is to optimise a linear policy depending on 36 parameters, where the objective is given by the average cumulative reward over 4 episodes of 1000 steps each. This evaluation is subject to significant noise from the environment's inherent stochasticity. To fur-

ther challenge the algorithms' performances, an adversary also introduces additional spuriously low rewards of $-1000$ with a frequency of $T^{1/3}$. Similarly to Section 5.4, the optimum policy is unknown and we chose an adversary that injects corruption as fast as its budget allows it independently of the chosen policy. The experiment is then repeated 10 times with different random seeds. We plot the results in Figure 5, where we defined regret as the difference between the best-observed reward across all BO runs and that of the current policy. Solid lines represent the mean, and shaded regions represent the standard error.

Despite having run for 300 iterations, *none* of the BO algorithms have stabilised due to the problem's high dimension. As this behaviour is common in practice and for high-dimensional problems, it is worth understanding whether the asymptotic regret guarantees derived in our paper are practically meaningful before regret guarantees converge. On this example, our results suggest that they are: RCGP-based methods improve upon GP-UCB. More interestingly, other robust baselines significantly underperform—even relative to GP-UCB—which is due to their overly conservative exploration strategies being maladapted to high-dimensional spaces. For the Diagnostics-GP for instance, we hypothesise that many non-outlying data points will be treated as outliers and completely excluded from consideration due to the fact that $d$-dimensional Gaussians quickly concentrate onto a ring of radius $\sqrt{d}$ (Wainright, 2019).

### 5.6. Discussion

A2-RCGP-UCB and FC-RCGP-UCB achieved zero-cost robustness: their cumulative regrets matched GP-UCB in the absence of corruptions (Theorem 4.1). Moreover, across all three benchmarks, the RCGP-based algorithms achieved the lowest cumulative regret in the corrupted settings. Diagnostics-GP, which hard-thresholds outliers, can also recover from corruption, especially when its magni-

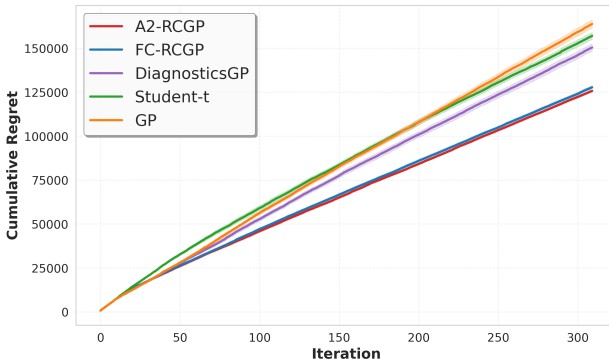

*Figure 5.* Cumulative regret of linear policy optimization on the 36D Lunar-Lander-3 environment.

tude is large; but this aggressiveness costs efficiency under low-magnitude corruptions and in the uncorrupted setting.

The ablations separate the roles of P-IMQ's two ingredients. Robustness is guaranteed by the weighting mechanism: our corrupted-regret analysis (Appendix D) shows any weight satisfying the Huber condition $\sup_y |y|\, w(\boldsymbol{x}, y)^2 < \infty$ (Altamirano et al., 2024) bounds a corruption's influence and will achieve the regret bounds in Theorems 4.2 and 4.3. The decay rate of the weight is nonetheless a genuine design choice that the experimenter should set from their knowledge of $f$: a fast-decaying tail sharply rejects residuals outside the expected range and is preferable when one is confident about where $f$ lives, whereas a slow-decaying tail is safer under uncertainty, as it is less likely to discard genuine observations. This is borne out in the uncorrupted setting, where IMQ, which has the slowest decay, performs best (Appendix H.2). However, it is also worth noting that input standarization seems to flatten the differences between the regrets achieved by the different decay rates making the choice less consequential. As expected, the plateau governs convergence dynamics: too tight a half-width $L$ down-weights genuine observations and slows convergence, while too wide an $L$ lets corruptions exert more influence, producing the U-shaped optimum in $L$ (Appendix H.1).

A final theme is insensitivity to hyperparameters. Although our analysis assumes a known budget $T_{\mathrm{c}}$, which is a standard assumption in the litteratyre (Kirschner & Krause, 2021; Bogunovic et al., 2020; 2022), and a known function-norm bound, neither is necessary in practice. Estimating $T_{\mathrm{c}}$ online works well, and the acquisition ablation (Appendix H.3) shows that using $T_{\mathrm{c}} = 0$ matches or improves on it; that a smaller multiplier than the theory prescribes never loses the optimum indicates the derived confidence intervals, and the regret bounds they entail, are loose and could be sharpened. A fixed heuristic ($\sqrt{B_f \kappa} = 1$) sufficed even for larger-magnitude functions, and input standardisation largely flattened the differences between weight-decay rates

(Appendix H.2). Consistently, our runs tolerated budgets scaling as $T^{1/3}$, matching the 'ideal' rates of Section 4.3 rather than the proven ones. This suggests that our proof technique could be further refined to obtain tighter regret bounds.

## 6. Conclusion

We addressed a fundamental limitation in the literature on robust BO by deriving an algorithm with sublinear regret in the presence of unbounded corruptions. Our approach is built on two pillars: on a conceptual level, we replaced a magnitude-constrained budget for the adversary with a frequency-constrained one. On an inferential level, we handle this adversary by considering methodology that goes beyond standard Bayesian approaches (Altamirano et al., 2024; Laplante et al., 2025). A key insight leveraged in our development is the inherent lack of robustness of Bayesian methods (see e.g. Bissiri et al., 2016; Knoblauch et al., 2019)—an issue that has recently generated a lot of attention, and has spawned post-Bayesian approaches that go beyond representing uncertainty via conditional probabilities. The RCGP that our method revolves around is but one example of this larger body of work. As we have shown, the intersection between these ideas and active learning is underexplored, and shows much promise: by going beyond Bayes' rule as an inferential device for updating beliefs that quantify uncertainty, we may be able to obtain guarantees that would be hard to achieve if we were using belief updates based on conditional probabilities. The current paper is an example of this, and obtained the first algorithms that provably achieve sublinear regret in the presence of up to $\mathcal{O}(T^{1/4})$ unbounded corruptions without sacrificing efficiency in the absence of corruptions.

## Acknowledgments

Abdelhamid Ezzerg was supported by the Engineering and Physical Sciences Research Council Doctoral Training Partnership (EPSRC DTP) EP/W524335/1. Ilija Bogunovic was supported by the EPSRC New Investigator Award EP/X03917X/1; the EPSRC EP/S021566/1; and Google Research Scholar award. Jeremias Knoblauch was supported through EPSRC via EP/W005859/1.

## Impact Statement

This paper presents work whose goal is to advance the field of Machine Learning. There are many potential societal consequences of our work, none which we feel must be specifically highlighted here.

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

# A. Notation

This section summarizes the mathematical notation used throughout the appendices.

**Optimization Setting and Function Properties**

- $\mathcal{X}$: The decision domain, (e.g. a compact subset of $\mathbb{R}^d$). $d$ is the dimension of the domain.

- $f(\boldsymbol{x})$: The true underlying objective function.

- $\boldsymbol{x}^*$: The optimizer, $\boldsymbol{x}^* = \arg\max_{\boldsymbol{x} \in \mathcal{X}} f(\boldsymbol{x})$.

- $t, T$: Current time step and the total time horizon, respectively.

- $R_T$: Cumulative regret, $R_T = \sum_{t=1}^{T} (f(\boldsymbol{x}^*) - f(\boldsymbol{x}_t))$.

- $k(\boldsymbol{x}, \boldsymbol{x}')$: The kernel (covariance) function.

- $\mathcal{H}_k(\mathcal{X})$: The Reproducing Kernel Hilbert Space (RKHS) associated with $k$.

- $\mathcal{O}(\cdot), \tilde{\mathcal{O}}(\cdot)$: Standard Big-O notation. $\tilde{\mathcal{O}}(\cdot)$ hides polylogarithmic factors in $T$. For example, $\mathcal{O}(T \log(T)) = \tilde{\mathcal{O}}(T)$.

**Observations and Corruption Model**

- $\boldsymbol{x}_t$: The point (a vector) selected by the algorithm at time $t$.

- $\epsilon_t$: Benign observation noise at time $t$ (assumed sub-Gaussian).

- $\sigma_{\text{noise}}^2$: The noise variance or sub-Gaussian parameter.

- $c_t$: Adversarial corruption at time $t$. The magnitude can be arbitrarily large.

- $y_t$: The scalar observation, $y_t = f(\boldsymbol{x}_t) + \epsilon_t + c_t$.

- $T_{\text{c}}$: The total budget on the number of corruptions, $T_{\text{c}} = |\{t : c_t \neq 0\}|$.

- $\alpha$: The scaling exponent for corruptions, assuming $T_{\text{c}} = \mathcal{O}(T^\alpha)$.

**Data Subsets and Matrices**

- $\mathcal{D}_t = \{(\boldsymbol{x}_i, y_i)\}_{i=1}^t$: The full dataset up to time $t$.

- $\mathcal{D}_{\text{uc},t}, \mathcal{D}_{\text{c},t}$: The subsets of uncorrupted ($c_i = 0$) and corrupted ($c_i \neq 0$) observations, respectively.

- $T_{\text{c}}$: The count of corrupted points up to time $t$, $T_{\text{c}} = |\mathcal{D}_{\text{c},t}|$.

- $\boldsymbol{X}_t, \boldsymbol{y}_t$: The matrix of inputs and vector of observations up to time $t$. Each row of $\boldsymbol{X}_t$ is a vector $\boldsymbol{x}_i$ and each element of $\boldsymbol{y}_t$ is an observation $y_i$.

- $\boldsymbol{X}_{\text{uc},t}, \boldsymbol{y}_{\text{uc},t}$ (and $\boldsymbol{X}_{\text{c},t}, \boldsymbol{y}_{\text{c},t}$): Inputs and observations restricted to the respective subsets.

- $\boldsymbol{K}_t$: The Gram matrix $[k(\boldsymbol{x}_i, \boldsymbol{x}_j)]_{i,j=1}^t$.

- $\boldsymbol{k}_t(\boldsymbol{x})$: The kernel vector $[k(\boldsymbol{x}, \boldsymbol{x}_1), \ldots, k(\boldsymbol{x}, \boldsymbol{x}_t)]^T$.

- $\boldsymbol{K}_{uc,uc}, \boldsymbol{K}_{c,c}, \boldsymbol{K}_{uc,c}$: Partitions of the Gram matrix corresponding to the respective subsets.

**Posterior Distributions**

- $m(\boldsymbol{x})$: The GP prior mean (often assumed $m(\boldsymbol{x}) = 0$). $\boldsymbol{m}_t$ is the vector of prior means at $\boldsymbol{X}_t$.

- $\mu_{\text{uc},t}(\boldsymbol{x}), \sigma_{\text{uc},t}(\boldsymbol{x})$: The **standard GP posterior** mean and standard deviation conditioned only on the uncorrupted data $\mathcal{D}_{\text{uc},t}$.

- $\mu_t^{\text{R}}(\boldsymbol{x}), \sigma_t^{\text{R}}(\boldsymbol{x})$: The **Robust Conjugate Gaussian Process (RCGP) posterior** mean and standard deviation conditioned on the full dataset $\mathcal{D}_t$.

- $\text{cov}_{\mathcal{D}}(\boldsymbol{X}_A, \boldsymbol{X}_B)$: The posterior covariance operator conditioned on a dataset $\mathcal{D} = \{(\boldsymbol{X}_D, \boldsymbol{y}_D)\}$. It is defined as $k(\boldsymbol{X}_A, \boldsymbol{X}_B) - \boldsymbol{k}(\boldsymbol{X}_A, \boldsymbol{X}_D)^T (\boldsymbol{K}_{D,D} + \sigma_{\text{noise}}^2 \boldsymbol{I})^{-1} \boldsymbol{k}(\boldsymbol{X}_D, \boldsymbol{X}_B)$. Depending on the arguments, it can return a scalar, vector, or matrix.

  - $\text{cov}_{\mathcal{D}}(\boldsymbol{x}, \boldsymbol{x}')$: The posterior covariance between two points $\boldsymbol{x}$ and $\boldsymbol{x}'$.
  - $\text{cov}_{\mathcal{D}}(\boldsymbol{X}, \boldsymbol{x})$: The posterior covariance between a set of points $\boldsymbol{X}$ and a single point $\boldsymbol{x}$, returning a vector $[\text{cov}_{\mathcal{D}}(\boldsymbol{x}', \boldsymbol{x})]_{\boldsymbol{x}' \in \boldsymbol{X}}$.
  - $\text{cov}_{\mathcal{D}}(\boldsymbol{X}_A, \boldsymbol{X}_B)$: The full posterior covariance matrix between two sets of points $\boldsymbol{X}_A$ and $\boldsymbol{X}_B$, returning a matrix $[\text{cov}_{\mathcal{D}}(\boldsymbol{x}', \boldsymbol{x}'')]_{\boldsymbol{x}' \in \boldsymbol{X}_A, \boldsymbol{x}'' \in \boldsymbol{X}_B}$.
  - We primarily use the covariance operator conditioned on the uncorrupted set, denoted $\text{cov}_{\mathcal{D}_{\text{uc},t}}$.

- Similarly we define $\text{cov}_{\mathcal{D}}^{\text{R}}(\boldsymbol{X}_A, \boldsymbol{X}_B) = k(\boldsymbol{X}_A, \boldsymbol{X}_B) - \boldsymbol{k}(\boldsymbol{X}_A, \boldsymbol{X}_D)^T (\boldsymbol{K}_{D,D} + \sigma_{\text{noise}}^2 \boldsymbol{J}_w)^{-1} \boldsymbol{k}(\boldsymbol{X}_D, \boldsymbol{X}_B)$ as the posterior covariance operator for RCGP.

- $\boldsymbol{S}$: The Schur complement used in the deviation analysis (Lemma D.2). $\boldsymbol{S} = \text{cov}_{\mathcal{D}_{\text{uc},t}}(\boldsymbol{X}_{\text{c},t}, \boldsymbol{X}_{\text{c},t}) + \sigma_{\text{noise}}^2 \boldsymbol{J}_{w,\text{c},t}$.

**RCGP Framework and P-IMQ**

- $w(\boldsymbol{x}, y)$: The RCGP weighting function. We use the P-IMQ: Plateau-Inverse Multi-Quadric weighting function defined in Definition 3.1.

- $g(\boldsymbol{x})$: The centering function for P-IMQ.

- $L$ (or $L_T, L_t(\boldsymbol{x})$): The half-width of the plateau for P-IMQ.

- $W$: The maximum weight of the P-IMQ, set to $\sigma_{\text{noise}}/\sqrt{2}$.

- $\boldsymbol{J}_w$: The diagonal weighting modification matrix, $\boldsymbol{J}_w = \text{diag}(\sigma_{\text{noise}}^2/(2w_i^2))$.

- $\boldsymbol{m}_w$: The mean modification vector (gradient correction), $\boldsymbol{m}_w = \boldsymbol{m}_t + \sigma_{\text{noise}}^2 \nabla_{\boldsymbol{y}} \log(\boldsymbol{w}^2)$.

- $\boldsymbol{A}_t$: The regularized kernel matrix in RCGP, $\boldsymbol{A}_t = \boldsymbol{K}_t + \sigma_{\text{noise}}^2 \boldsymbol{J}_w$.

- The **Plateau Condition**. The event that all uncorrupted observations fall within the plateau, ensuring $\boldsymbol{J}_w$ restricted to the uncorrupted indices is the identity matrix.

- In the derivation of Lemma D.2, we will define the following constants:

  - $C_1 \triangleq \sup_{\boldsymbol{x} \in \mathcal{X}, y \in \mathbb{R}} |w(\boldsymbol{x}, y)(y - m_w(\boldsymbol{x}, y) - \mu_{uc}(\boldsymbol{x}))|$. In the presented algorithms, the weighting function will change over time, as such, we will denote $C_{1,t}$ the value $C_{1,t} \triangleq \sup_{\boldsymbol{x} \in \mathcal{X}, y \in \mathbb{R}} |w_t(\boldsymbol{x}, y)(y - m_{w,t}(\boldsymbol{x}, y) - \mu_{\text{uc},t}(\boldsymbol{x}))|$.
  - $C_w \triangleq \frac{\sqrt{2}C_1}{\sigma_{\text{noise}}^2}$.

**Regret Analysis Parameters and Constants**

- $\delta$: The failure probability. The bounds derived in Theorems 4.1, 4.2, and 4.3 hold with probability at least $1 - \delta$.

- $\beta'_t(\delta')$: The standard UCB confidence parameter sequence.

- $\beta_t(\delta')$: The **robust confidence parameter**. To maintain consistency with the GP-UCB literature where the multiplier is $\sqrt{\beta'_t}$, we define our robust multiplier as $\sqrt{\beta_t}$. It is defined as:

$$\sqrt{\beta_t(\delta')} \triangleq \sqrt{\beta'_t(\delta')} + C_{w,t-1}\sqrt{T_{c,t-1}} \tag{5}$$

  where $C_{w,t-1}$ is the maximum deviation constant and $T_{c,t-1}$ is the number of corruptions up to time $t-1$. This implies that $\beta_t(\delta') = \left(\sqrt{\beta'_t(\delta')} + C_{w,t-1}\sqrt{T_{c,t-1}}\right)^2$.

- $N_T(\delta'')$: The high-probability bound on the noise magnitude over $T$ rounds. $N_T(\delta'')$ is slected such that the event $E_{\text{noise}}(\delta'') = \{\forall t \leq T : |\epsilon_t| \leq N_T(\delta'')\}$ holds with probability at least $1 - \delta''$.

- $\gamma_T$: The maximum information gain after $T$ rounds.

- $d_t(\boldsymbol{x})$: The posterior deviation due to corruptions up to time $t$: $d_t(\boldsymbol{x}) = \mu_t^{\text{R}}(\boldsymbol{x}) - \mu_{\text{uc},t}^{\text{R}}(\boldsymbol{x})$.

- $\Delta_t(\boldsymbol{x})$: The absolute deviation of the uncorrupted posterior mean at time $t$ from the centering function, $\Delta_t(\boldsymbol{x}) = |g(\boldsymbol{x}) - \mu_{\text{uc},t}^{\text{R}}(\boldsymbol{x})|$.

- We use $\succeq$ to denote the Loewner order, i.e., $A \succeq B$ means $A - B$ is positive semi-definite.

**Hierarchical Model (A2-AW)**

- $M_{\text{⚓}}$: The Anchor model (stable configuration). **For ease of notation within the proof, we will denote the anchor model by $M_{\text{A}}$.**

- $M_{\text{⚡}}$: The Acquisition model (adaptive configuration). **For ease of notation within the proof, we will denote the acquisition model by $M_{\text{R}}$.**

- $\mu_{A,t}^{\text{R}}, \sigma_{A,t}^{\text{R}}$ (and $\mu_{R,t}^{\text{R}}, \sigma_{R,t}^{\text{R}}$): Posterior means and standard deviations of the respective models.

- $L_{A,T}$: Fixed plateau width for $M_{\text{A}}$.

- $L_{R,t}(\boldsymbol{x})$: Adaptive plateau width for $M_{\text{R}}$.

- $\beta_{A,t}, \beta_{R,t}$: Robust confidence parameters for the Anchor and Acquisition models, respectively. The confidence multipliers used in the UCB bounds are $\sqrt{\beta_{A,t}}$ and $\sqrt{\beta_{R,t}}$.

**High-Probability Events**

- $E_{\text{noise}}(\delta')$: $\{|\epsilon_t| \leq N_T(\delta') \quad \forall t \leq T, \boldsymbol{x}_t \in \mathcal{X}\}$. This is the event that the noise is bounded by $N_T(\delta')$ for all time steps $t \leq T$.

- $E_{\text{conf}}(\delta')$: $\{|f(\boldsymbol{x}) - \mu_{\text{uc},t}(\boldsymbol{x})| \leq \sqrt{\beta'_t(\delta')}\sigma_{\text{uc},t}(\boldsymbol{x}) \quad \forall t \leq T, \boldsymbol{x}\}$. Please note the use of $\beta'_t(\delta')$, the confidence parameter for the standard GP-UCB algorithm, instead of $\beta_t(\delta')$.

- $E_{\text{Plateau}}(\delta')$: Joint high-probability events $E_{\text{noise}}(\delta'/2) \cap E_{\text{conf}}(\delta'/2)$.

## B. Assumptions

We make the following assumptions:

**Assumption B.1** (GP-UCB Assumptions)**.** We make the same assumptions as in Srinivas et al. (2010). We distinguish three mutually exclusive cases corresponding to Theorems 1, 2, and 3 in Srinivas et al. (2010), respectively:

- **Case 1: the domain $\mathcal{X}$ is finite**. We assume that $f$ is a sample from a GP with mean zero and covariance $k(\boldsymbol{x}, \boldsymbol{x}')$. The noise is assumed i.i.d. $\epsilon_t \sim \mathcal{N}(0, \sigma_{\text{noise}}^2)$.

- **Case 2: the domain $\mathcal{X}$ is compact and convex**. We assume that $f$ is a sample from a GP with mean zero and covariance $k(\boldsymbol{x}, \boldsymbol{x}')$. We further assume that the kernel $k(\boldsymbol{x}, \boldsymbol{x}')$ satisfies the high probability bound on the derivatives of GP sample paths $f$: for some constants $a, b > 0$,

$$\mathbb{P}\left\{\sup_{x \in \mathcal{X}} |\partial f / \partial x_j| > L\right\} \leq a e^{-(L/b)^2} \text{ for } j = 1, \ldots, d.$$

  The noise is assumed i.i.d. $\epsilon_t \sim \mathcal{N}(0, \sigma_{\text{noise}}^2)$.

- **Case 3: Arbitrary functions in RKHS**. We assume that $f$ lies in the RKHS $\mathcal{H}_k(\mathcal{X})$ corresponding to the kernel $k(\boldsymbol{x}, \boldsymbol{x}')$. We further assume that $\|f\|_k^2 \leq B_f$. We also assume that the noise $\epsilon_t$ is such that $\mathbb{E}[\epsilon_t | \text{history}] = 0$ and $|\epsilon_t| \leq \sigma_{\text{noise}}$ almost surely.

**In all three cases, the prior mean is assumed to be zero, i.e., $m(\boldsymbol{x}) = 0$.**

**Assumption B.2** (Kernel Boundedness). The kernel is bounded, i.e., $|k(\boldsymbol{x}, \boldsymbol{x}')| \leq \kappa$ for all $\boldsymbol{x}, \boldsymbol{x}' \in \mathcal{X}$.

Srinivas et al. (2010) also assumed kernel boundedness in their analysis. We highlighted it separately because we directly use this assumption in our analysis.

**Assumption B.3** (Objective Function Boundedness). The function $f$ is bounded by $B$ with a probability of at least $1 - \delta$: $\Pr\left(\sup_{\boldsymbol{x} \in \mathcal{X}} |f(\boldsymbol{x})| \leq B\right) \geq 1 - \delta$. We denote this high-probability event by $E_f(\delta)$.

A stronger version of Assumption B.3, where $f$ is bounded by $B$ in an absolute sense and not as a probability event, was implicitly assumed in the following cases discussed by Srinivas et al. (2010):

- **Case 2: the domain $\mathcal{X}$ is compact and convex**: The function $f$ is bounded as a continuous function over a compact domain.

- **Case 3: Arbitrary functions in RKHS**: The function $f$ is assumed to have a bounded norm in the RKHS $\mathcal{H}_k(\mathcal{X})$ ($\|f\|_k^2 \leq B_f$) according to Assumption B.1 Case 3. In this case, $f$ is bounded by $\sqrt{B_f \kappa}$ (where $\kappa$ is the kernel bound in Assumption B.2).

Case 1 remain the only case where Srinivas et al. (2010) did not assume objective function boundedness.
**In order to keep a conistent notation across all three cases, we will denote $\sqrt{B_f} = \frac{B}{\sqrt{\kappa}}$ for Case 1 and Case 2.**

## C. Proofs Structure

The core of the regret analysis in Appendix D is to **bound the deviation** between the robust posterior, which is influenced by corrupted data, and an idealised uncorrupted posterior. The proof proceeds in several key stages. First, we establish a high-probability **Plateau** event, which demonstrates that the P-IMQ function's plateau is wide enough to contain all uncorrupted observations. When this event holds, the RCGP model behaves identically to a standard GP if conditioned solely on the clean data. Second, in Section D.3, we derive an explicit **Deviation Bound**. Lemma D.2 provides a formula for the posterior mean deviation, and Lemma D.3 shows this is bounded by a term proportional to the square root of the number of corruptions ($\sqrt{T_c}$) and the idealized uncorrupted posterior standard deviation ($\sigma_{\text{uc},t}(\boldsymbol{x})$). This bound is then used in Lemma D.4 to construct an **Enlarged Confidence Interval** for the RCGP posterior. Finally, this robust confidence interval is plugged into a standard UCB-style **Regret Analysis** to derive the final regret bounds. For the A2-RCGP-UCB algorithm, this process is tiered: a stable Anchor model is first analyzed to provide guarantees for the more adaptive Acquisition model.

Section F proves regret bounds in the absence of corruption as a direct result of the general analysis in Appendix D when $T_c = 0$.

Lastly, Section G analyses the constant $C_1 = \sup_{\boldsymbol{x} \in \mathcal{X}, y \in \mathbb{R}} |w(\boldsymbol{x}, y)(y - m_w(\boldsymbol{x}, y) - \mu_{uc}(\boldsymbol{x}))|$ as a function of the P-IMQ plateau's width. Analysis of this constant is crucial as it directly arises in the deviation bound of Lemma D.3. Since the plateau width changes over time, this step is required to understand the asymptotic behavior of $C_{1,T}$ as a function of $T$.

The analysis of the constant $C_1$ was deferred to a separate section because it is specific to the P-IMQ function. The regret analysis would be valid for other other weighting functions as long as they: 1. have a plateau with value $W = \frac{\sigma_{\text{noise}}}{\sqrt{2}}$. 2. Are strictly positive and bounded from above. 3. Have a continuous gradient $\nabla_y w(y)$.

## D. Robust Regret Analysis (Adversarial Corruption)

This appendix provides a detailed regret analysis for two robust Bayesian Optimization algorithms based on the RCGP model: Fixed-Center RCGP-UCB (FC-RCGP-UCB) and the Anchor-Adapt RCGP-UCB (A2-RCGP-UCB). We establish rigorous regret bounds under a powerful adversarial corruption model.

### D.1. Plateau Event

We start by briefly explaining the plateau event and why it is crucial for the analysis. From the definition of the P-IMQ function in Definition 3.1, we have that the P-IMQ function has a plateau around the center $g(\boldsymbol{x})$ with width $L(\boldsymbol{x})$. Within the plateau, the P-IMQ function is equal to $W = \frac{\sigma_{\text{noise}}}{\sqrt{2}}$ and its gradient relative to the observation $y$ is zero. Therefore, if all observations are within the plateau of the P-IMQ we have:

$$\boldsymbol{J}_w = \text{diag}\left(\frac{\sigma_{\text{noise}}^2}{2.w_i^2}\right) = \text{diag}\left(\frac{\sigma_{\text{noise}}^2}{2.W^2}\right) = \boldsymbol{I}$$

and,

$$m_w = \sigma_{\text{noise}}^2[\nabla_y \log(w(\boldsymbol{x}_i, y_i)^2)]_i = 0$$

As a consequence, the RCGP posterior defined in Equations 2 and 3 becomes exactly the same as the standard GP posterior. In our analysis, we will construct the plateau width $L(x)$ of the P-IMQ function such that, with high probability, all uncorrupted observations are within the plateau region. We will refer to this event as $E_{\text{Plateau}}$ and we will also refer to it as the **plateau** condition. When the plateau event holds, the RCGP and standard GP posterior distributions are exactly the same when conditioned on uncorrupted observations. We write this as:

$$\mu_{\text{uc},t}^{\text{R}}(\boldsymbol{x}) = \mu_{\text{uc},t}(\boldsymbol{x})$$

$$\sigma_{\text{uc},t}^{\text{R}}(\boldsymbol{x}) = \sigma_{\text{uc},t}(\boldsymbol{x})$$

### D.2. High-Probability Events

For the regret analysis, we will define the following high-probability events:

- $E_{\text{noise}}(\delta')$: $\{|\epsilon_t| \leq N_T(\delta') \quad \forall t \leq T, \boldsymbol{x}_t \in \mathcal{X}\}$. This is the event that the noise is bounded by $N_T(\delta')$ for all time steps $t \leq T$. $N_T(\delta')$ is constructed such that the event $E_{\text{noise}}(\delta')$ holds with probability at least $1 - \delta'$. $N_T(\delta')$ is constructed depending on the each case of the domain considered in Assumption B.1:

  - **Case 1: the domain $\mathcal{X}$ is finite.** $N_T(\delta') = \sigma_{\text{noise}}\sqrt{2\log(T/\delta')}$.
  - **Case 2: the domain $\mathcal{X}$ is compact and convex.** $N_T(\delta') = \sigma_{\text{noise}}\sqrt{2\log(T/\delta')}$.
  - **Case 3: arbitrary functions in RKHS.** $N_T(\delta') = \sigma_{\text{noise}}$ (the noise is assumed bounded by $\sigma_{\text{noise}}$ almost surely in this B.1 Case 3).

- $E_{\text{conf}}(\delta')$: $\{|f(\boldsymbol{x}) - \mu_{\text{uc},t}(\boldsymbol{x})| \leq \sqrt{\beta_t'(\delta')}\sigma_{\text{uc},t}(\boldsymbol{x}) \quad \forall t \leq T, \boldsymbol{x}\}$. Please note the use of $\beta_t'(\delta')$, the confidence parameter for the stnadard GP-UCB algorithm, instead of $\beta_t(\delta')$. Srinivas et al. (2010) proved that, under Assumptions B.1 and B.2, the event $\{|f(\boldsymbol{x}) - \mu_{\text{uc},t}(\boldsymbol{x})| \leq \sqrt{\beta_{t_{\text{uc}}}'(\delta')}\sigma_{\text{uc},t}(\boldsymbol{x}) \quad \forall t \leq T, \boldsymbol{x}\}$ holds with a probability at least $1 - \delta'$. Since $\beta_t'$ is non-decreasing in $t$, the event $E_{\text{conf}}(\delta')$ holds with probability at least $1 - \delta'$ under the same assumptions.

  $\beta_t'(\delta')$ is constructed in Srinivas et al. (2010). We distinguish three cases similar to the ones in Assumption B.1:

  - **Case 1: the domain $\mathcal{X}$ is finite:** $\beta_t'(\delta') = 2\log(|\mathcal{X}|t^2\pi^2/(6\delta'))$.
  - **Case 2: the domain $\mathcal{X}$ is compact and convex:**

    $$\beta_t'(\delta') = 2\log(t^2 2\pi^2/(3\delta') + 2d\log(t^2 dbr\sqrt{\log(4da/\delta')})$$

    Where the constants $a, b, r$ are defined in the Theorem 2 of the GP-UCB paper (Srinivas et al., 2010).

- **Case 3: arbitrary functions in RKHS**: $\beta_t'(\delta') = 2B_f + 300\gamma_t \log^3(t/\delta')$.

- $E_f(\delta')$: $\{\sup_{\boldsymbol{x}\in\mathcal{X}} |f(\boldsymbol{x})| \leq B\}$. The event that the objective function is bounded by $B$. In Assumption B.3, we assume that $E_f(\delta')$ holds with probability at least $1 - \delta'$.

- $E_{\text{Plateau}}(\delta')$: Joint high-probability events $E_{\text{noise}}(\delta'/3) \cap E_{\text{conf}}(\delta'/3) \cap E_f(\delta'/3)$. The event holds with probability at least $1 - \delta'$. The derivation of Theorems 4.1, 4.2, and 4.3 hinges on the uncorrupted observations falling within the Plateau-IMQ plateau region.

### D.3. Technical Lemmas:

#### D.3.1. BOUNDING THE POSTERIOR DEVIATION

We define the deviation between the RCGP posterior and the uncorrupted standard posterior at time $t$ as $d_t(\boldsymbol{x}) = \mu_t^{\text{R}}(\boldsymbol{x}) - \mu_{\text{uc},t}(\boldsymbol{x})$. The following lemmas derive bounds on this deviation, provided the high probability event $E_{\text{Plateau}}(\delta)$ holds.

**Lemma D.1.** *Let's assume that Assumptions B.1 B.2, and B.3 hold. When the high probability events $E_{conf}(\delta')$ and $E_f(\delta')$ hold, the uncorrupted standard posterior mean is bounded by a constant $M_t = \mathcal{O}(\sqrt{\beta_t'})$ for all $t \leq T$ and $\boldsymbol{x} \in \mathcal{X}$.*

*Proof.* For all $t \leq T$ and $\boldsymbol{x} \in \mathcal{X}$, we have:

$$
\begin{aligned}
|\mu_{\text{uc},t}(\boldsymbol{x})| &\leq |f(\boldsymbol{x}) - \mu_{\text{uc},t}(\boldsymbol{x})| + |f(\boldsymbol{x})| && \text{(Triangle inequality)} \\
&\leq \sqrt{\beta_t'(\delta')}\sigma_{\text{uc},t}(\boldsymbol{x}) + \sqrt{B_f}\kappa && \text{(Events } E_{\text{conf}}(\delta') \text{ and } E_f(\delta') \text{ hold)} \\
&\leq \sqrt{\kappa}(\sqrt{\beta_t'(\delta')} + \sqrt{B_f}) && \text{(Assumption B.2)}.
\end{aligned}
$$

Therefore, we have $M_t = \mathcal{O}(\sqrt{\beta_t'})$ for all $t \leq T$ and $\boldsymbol{x} \in \mathcal{X}$. $\qquad\square$

**Lemma D.2.** *When the plateau condition holds, the deviation $d_t(\boldsymbol{x})$ is given by:*

$$
d_t(\boldsymbol{x}) = \text{cov}_{\mathcal{D}_{uc,t}}(\boldsymbol{X}_{c,t}, \boldsymbol{x})^T \boldsymbol{S}^{-1}(\boldsymbol{y}_{c,t} - \boldsymbol{m}_{w,c,t} - \mu_{uc,t}(\boldsymbol{X}_{c,t})),
$$

*where $\boldsymbol{S} = \text{cov}_{\mathcal{D}_{uc,t}}(\boldsymbol{X}_{c,t}, \boldsymbol{X}_{c,t}) + \sigma_{noise}^2 \boldsymbol{J}_{w,c,t}$ is the Schur complement of the matrix $\boldsymbol{A}_t$ with respect to the uncorrupted subset.*

*Proof.* Let $\boldsymbol{A}_t = \boldsymbol{K}_t + \sigma_{\text{noise}}^2 \boldsymbol{J}_w$ be the regularized kernel matrix for $\mathcal{D}_t$. We partition the matrices according to the uncorrupted (uc) and corrupted (c) sets.

$$
\boldsymbol{A}_t = \begin{pmatrix} \boldsymbol{A}_{\text{uc,uc}} & \boldsymbol{K}_{\text{uc,c}} \\ \boldsymbol{K}_{\text{c,uc}} & \boldsymbol{A}_{\text{c,c}} \end{pmatrix}.
$$

We assume that the plateau condition holds. In this case, the RCGP parameters for the uncorrupted subset are $\boldsymbol{J}_{w,\text{uc}} = \boldsymbol{I}$ and the gradient correction is zero ($m_w(\boldsymbol{x}) = m(\boldsymbol{x}) = 0$). Thus, $\boldsymbol{A}_{\text{uc,uc}} = \boldsymbol{K}_{\text{uc,uc}} + \sigma_{\text{noise}}^2 \boldsymbol{I}$.

Let's define the coefficient vector $\boldsymbol{a}_t$ for the RCGP posterior mean as $\boldsymbol{a}_t = \boldsymbol{A}_t^{-1}(\boldsymbol{y}_t - \boldsymbol{m}_{w,t})$, so that $\mu_t^{\text{R}}(\boldsymbol{x}) = \boldsymbol{k}_t(\boldsymbol{x})^T \boldsymbol{a}_t$. Similarly, let $\boldsymbol{a}_{\text{uc,t}} = \boldsymbol{A}_{\text{uc,uc}}^{-1} \boldsymbol{y}_{\text{uc}}$ be the coefficient vector for the uncorrupted posterior mean $\mu_{\text{uc},t}(\boldsymbol{x}) = \boldsymbol{k}_{\text{uc},t}(\boldsymbol{x})^T \boldsymbol{a}_{\text{uc,t}}$. Since $\boldsymbol{m}_{w,\text{uc},t} = \boldsymbol{0}$, we have $\boldsymbol{m}_{w,t} = (\boldsymbol{0}^T, \boldsymbol{m}_{w,c,t}^T)^T$.

We augment $\boldsymbol{a}_{\text{uc,t}}$ with zeros: $\tilde{\boldsymbol{a}}_{\text{uc,t}} = (\boldsymbol{a}_{\text{uc,t}}^T, 0)^T$. Let $\Delta \boldsymbol{a}_t = \boldsymbol{a}_t - \tilde{\boldsymbol{a}}_{\text{uc,t}}$. We solve for $\Delta \boldsymbol{a}_t$ using $\boldsymbol{A}_t \Delta \boldsymbol{a}_t = (\boldsymbol{y}_t - \boldsymbol{m}_{w,t}) - \boldsymbol{A}_t \tilde{\boldsymbol{a}}_{\text{uc,t}}$. We compute the second term on the right-hand side. Since $\boldsymbol{A}_{\text{uc,uc}} \boldsymbol{a}_{\text{uc,t}} = \boldsymbol{y}_{\text{uc}}$ and $\boldsymbol{K}_{\text{c,uc}} \boldsymbol{a}_{\text{uc,t}} = \mu_{\text{uc},t}(\boldsymbol{X}_c)$:

$$
\boldsymbol{A}_t \tilde{\boldsymbol{a}}_{\text{uc,t}} = \begin{pmatrix} \boldsymbol{A}_{\text{uc,uc}} \boldsymbol{a}_{\text{uc,t}} \\ \boldsymbol{K}_{\text{c,uc}} \boldsymbol{a}_{\text{uc,t}} \end{pmatrix} = \begin{pmatrix} \boldsymbol{y}_{\text{uc}} \\ \mu_{\text{uc},t}(\boldsymbol{X}_{\text{c,t}}) \end{pmatrix}.
$$

Now we compute $\boldsymbol{A}_t \Delta \boldsymbol{a}_t$:

$$
\boldsymbol{A}_t \Delta \boldsymbol{a}_t = \begin{pmatrix} \boldsymbol{y}_{\text{uc}} \\ \boldsymbol{y}_{\text{c},t} - \boldsymbol{m}_{w,c,t} \end{pmatrix} - \begin{pmatrix} \boldsymbol{y}_{\text{uc}} \\ \mu_{\text{uc},t}(\boldsymbol{X}_{\text{c,t}}) \end{pmatrix} = \begin{pmatrix} \boldsymbol{0} \\ \boldsymbol{y}_{\text{c},t} - \boldsymbol{m}_{w,c,t} - \mu_{\text{uc},t}(\boldsymbol{X}_{\text{c,t}}) \end{pmatrix} = \boldsymbol{b}.
$$

We solve the system $A_t \Delta a_t = b$ for $\Delta a_t = (\Delta a_{\text{uc}}^T, \Delta a_c^T)^T$ using blockwise substitution. The block matrix equation expands to:

$$A_{\text{uc,uc}} \Delta a_{\text{uc}} + K_{\text{uc,c}} \Delta a_c = 0 \tag{6}$$

$$K_{\text{c,uc}} \Delta a_{\text{uc}} + A_{\text{c,c}} \Delta a_c = y_{\text{c},t} - m_{w,c,t} - \mu_{\text{uc},t}(X_{\text{c},t}) \tag{7}$$

From Equation 6, we get

$$\Delta a_{\text{uc}} = -A_{\text{uc,uc}}^{-1} K_{\text{uc,c}} \Delta a_c \tag{8}$$

Substituting this into Equation 7 gives:

$$(A_{\text{c,c}} - K_{\text{c,uc}} A_{\text{uc,uc}}^{-1} K_{\text{uc,c}}) \Delta a_c = y_{\text{c},t} - m_{w,c,t} - \mu_{\text{uc},t}(X_{\text{c},t}).$$

The term in the parenthesis is the Schur complement of $A_t$ with respect to the block $A_{\text{uc,uc}}$. As given in the lemma statement, we denote this by $S$. Thus, we can solve for $\Delta a_c$:

$$\Delta a_c = S^{-1}(y_{\text{c},t} - m_{w,c,t} - \mu_{\text{uc},t}(X_{\text{c},t})).$$

The deviation is $d_t(x) = k_t(x)^T \Delta a_t = k_{\text{uc}}(x)^T \Delta a_{\text{uc}} + k_c(x)^T \Delta a_c$. Substituting the from Equation 8:

$$d_t(x) = k_{\text{uc}}(x)^T(-A_{\text{uc,uc}}^{-1} K_{\text{uc,c}} \Delta a_c) + k_c(x)^T \Delta a_c$$
$$= (k_c(x)^T - k_{\text{uc}}(x)^T A_{\text{uc,uc}}^{-1} K_{\text{uc,c}}) \Delta a_c.$$

The term in the left parenthesis is precisely the transpose of the posterior covariance vector, $\text{cov}_{\mathcal{D}_{\text{uc},t}}(X_{\text{c},t}, x)$. Substituting this and the expression for $\Delta a_c$ into the equation for the deviation gives the final result:

$$d_t(x) = \text{cov}_{\mathcal{D}_{\text{uc},t}}(x, X_{\text{c},t})^T S^{-1}(y_{\text{c},t} - m_{w,c,t} - \mu_{\text{uc},t}(X_{\text{c},t})).$$

$\square$

**Lemma D.3.** *Let's assume that Assumption B.2 holds. Let's also assume that the plateau condition holds. Then, the deviation between the robust posterior and the uncorrupted standard posterior is bounded by:*

$$|d_t(x)| \le C_{w,t} \cdot \sqrt{T_c} \cdot \sigma_{uc,t}(x). \tag{9}$$

*Where $C_{w,t} = \frac{\sqrt{2} C_{1,t}}{\sigma_{noise}^2}$ and $C_{1,t} = \sup_{x \in \mathcal{X}, y \in \mathbb{R}} |w_t(x, y)(y - m_w(x, y) - \mu_{uc,t}(x))|$*

*Proof.* From Lemma D.2, the deviation is given by $d_t(x) = \text{cov}_{\mathcal{D}_{\text{uc},t}}(X_{\text{c},t}, x)^T S^{-1}(y_{\text{c},t} - m_{w,c,t} - \mu_{\text{uc},t}(X_{\text{c},t}))$. Let $u(x) = \text{cov}_{\mathcal{D}_{\text{uc},t}}(X_{\text{c},t}, x)$ and $g_t = y_{\text{c},t} - m_{w,c,t} - \mu_{\text{uc},t}(X_{\text{c},t})$. The deviation can be expressed as the inner product:

$$d_t(x) = u(x)^T S^{-1} g_t = (S^{-1/2} u(x))^T (S^{-1/2} g_t).$$

Applying the Cauchy-Schwarz inequality to the squared magnitude of the deviation yields:

$$|d_t(x)|^2 = \left| (S^{-1/2} u(x))^T (S^{-1/2} g_t) \right|^2 \le \left\| S^{-1/2} u(x) \right\|_2^2 \left\| S^{-1/2} g_t \right\|_2^2.$$

We proceed by bounding each term on the right-hand side separately.

**Bound on $\left\| S^{-1/2} u(x) \right\|_2^2$**

Let $C_0 = \text{cov}_{\mathcal{D}_{\text{uc},t}}(X_{\text{c},t}, X_{\text{c},t})$. From the definition $S = C_0 + \sigma_{\text{noise}}^2 J_{w,c,t}$ and the fact that $\sigma_{\text{noise}}^2 J_{w,c,t}$ is a positive semi-definite matrix, we have the matrix inequality $S \succeq C_0$ in the Loewner order. This implies $S^{-1} \preceq C_0^{-1}$. Which in turn allows us to bound the first term as follows:

$$\left\| S^{-1/2} u(x) \right\|_2^2 = u(x)^T S^{-1} u(x) \le u(x)^T C_0^{-1} u(x).$$

We now prove that this final term is bounded by the variance $\sigma_{\text{uc},t}^2(\boldsymbol{x})$. This inequality, $\boldsymbol{u}(\boldsymbol{x})^T \boldsymbol{C}_0^{-1} \boldsymbol{u}(\boldsymbol{x}) \leq \sigma_{\text{uc},t}^2(\boldsymbol{x})$, is a direct consequence of the fact that the posterior covariance operator $\text{cov}_{\mathcal{D}_{\text{uc},t}}$ is, by definition, a valid positive semi-definite kernel. Any Gram matrix generated by a valid kernel must be positive semi-definite. Let us construct such a matrix by evaluating the operator $\text{cov}_{\mathcal{D}_{\text{uc},t}}$ on the set of points $\{\boldsymbol{x}\} \cup \boldsymbol{X}_{\text{c},t}$:

$$\begin{pmatrix} \text{cov}_{\mathcal{D}_{\text{uc},t}}(\boldsymbol{x}, \boldsymbol{x}) & \text{cov}_{\mathcal{D}_{\text{uc},t}}(\boldsymbol{x}, \boldsymbol{X}_{\text{c},t}) \\ \text{cov}_{\mathcal{D}_{\text{uc},t}}(\boldsymbol{X}_{\text{c},t}, \boldsymbol{x}) & \text{cov}_{\mathcal{D}_{\text{uc},t}}(\boldsymbol{X}_{\text{c},t}, \boldsymbol{X}_{\text{c},t}) \end{pmatrix} = \begin{pmatrix} \sigma_{\text{uc},t}^2(\boldsymbol{x}) & \boldsymbol{u}(\boldsymbol{x})^T \\ \boldsymbol{u}(\boldsymbol{x}) & \boldsymbol{C}_0 \end{pmatrix}.$$

Since this block matrix must be positive semi-definite, its Schur complement with respect to the block $\boldsymbol{C}_0$ must also be positive semi-definite. The Schur complement is given by:

$$\sigma_{\text{uc},t}^2(\boldsymbol{x}) - \boldsymbol{u}(\boldsymbol{x})^T \boldsymbol{C}_0^{-1} \boldsymbol{u}(\boldsymbol{x}).$$

As this complement is a scalar, for it to be positive semi-definite, it must simply be non-negative. This yields the inequality $\sigma_{\text{uc},t}^2(\boldsymbol{x}) - \boldsymbol{u}(\boldsymbol{x})^T \boldsymbol{C}_0^{-1} \boldsymbol{u}(\boldsymbol{x}) \geq 0$, which confirms the bound. Combining these results gives the final bound for the first term:

$$\left\| \boldsymbol{S}^{-1/2} \boldsymbol{u}(\boldsymbol{x}) \right\|_2^2 \leq \sigma_{\text{uc},t}^2(\boldsymbol{x}).$$

**Bound on** $\left\| \boldsymbol{S}^{-1/2} \boldsymbol{g}_t \right\|_2^2$

Since the covariance matrix $\boldsymbol{C}_0 = \text{cov}_{\mathcal{D}_{\text{uc},t}}(\boldsymbol{X}_{\text{c},t}, \boldsymbol{X}_{\text{c},t})$ is positive semi-definite, we have the matrix inequality $\boldsymbol{S} \succeq \sigma_{\text{noise}}^2 \boldsymbol{J}_{w,\text{c},t}$, which implies $\boldsymbol{S}^{-1} \preceq (\sigma_{\text{noise}}^2 \boldsymbol{J}_{w,\text{c},t})^{-1}$. This yields the following bound:

$$\left\| \boldsymbol{S}^{-1/2} \boldsymbol{g}_t \right\|_2^2 = (\boldsymbol{g}_t)^T \boldsymbol{S}^{-1} \boldsymbol{g}_t \leq (\boldsymbol{g}_t)^T (\sigma_{\text{noise}}^2 \boldsymbol{J}_{w,\text{c},t})^{-1} \boldsymbol{g}_t.$$

The matrix $\boldsymbol{J}_{w,\text{c},t}$ is diagonal with entries $\frac{\sigma_{\text{noise}}^2}{2w_i^2}$ for $i \in \mathcal{D}_{\text{c},t}$. Substituting this definition, we expand the right-hand side:

$$(\boldsymbol{g}_t)^T (\sigma_{\text{noise}}^2 \boldsymbol{J}_{w,\text{c},t})^{-1} \boldsymbol{g}_t = \sum_{i \in \mathcal{D}_{\text{c},t}} (g_{t,i})^2 \frac{2w_i^2}{\sigma_{\text{noise}}^4} = \frac{2}{\sigma_{\text{noise}}^4} \sum_{i \in \mathcal{D}_{\text{c},t}} (w_i g_{t,i})^2.$$

By the definition of the influence constant $C_{1,t}$, we have $|w_i g_{t,i}| = |w_i(y_i - m_{w,i} - \mu_{\text{uc},t}(\boldsymbol{x}_i))| \leq C_{1,t}$ for each $i \in \mathcal{D}_{\text{c},t}$. As the set $\mathcal{D}_{\text{c},t}$ contains at most $T_\text{c}$ points, we can bound the sum:

$$\left\| \boldsymbol{S}^{-1/2} \boldsymbol{g}_t \right\|_2^2 \leq \frac{2}{\sigma_{\text{noise}}^4} \sum_{i \in \mathcal{D}_{\text{c},t}} C_{1,t}^2 = \frac{2 T_\text{c} C_{1,t}^2}{\sigma_{\text{noise}}^4}.$$

**Combining the Bounds**

Finally, we substitute the bounds for both terms back into the Cauchy-Schwarz inequality:

$$|d_t(\boldsymbol{x})|^2 \leq \sigma_{\text{uc},t}^2(\boldsymbol{x}) \cdot \frac{2 T_\text{c} C_{1,t}^2}{\sigma_{\text{noise}}^4}.$$

Taking the square root of both sides and rearranging terms gives:

$$|d_t(\boldsymbol{x})| \leq \frac{\sqrt{2} C_{1,t}}{\sigma_{\text{noise}}^2} \cdot \sqrt{T_\text{c}} \cdot \sigma_{\text{uc},t}(\boldsymbol{x}).$$

Substituting the definition $C_{w,t} = \frac{\sqrt{2} C_{1,t}}{\sigma_{\text{noise}}^2}$ completes the proof. $\qquad \square$

**Lemma D.4.** *Let's assume that Assumptions B.1 and B.2 hold. Let's also assume that the plateau condition and the event $E_{conf}(\delta')$ hold. Then, for any $t \geq 1$, the RCGP posterior computed after step $t - 1$ satisfies:*

$$|f(\boldsymbol{x}) - \mu_{t-1}^{R}(\boldsymbol{x})| \leq \sqrt{\beta_t(\delta')}\sigma_{uc,t-1}(\boldsymbol{x}),$$

*where the robust confidence parameter is defined as $\beta_t(\delta') = \left(\sqrt{\beta_t'(\delta')} + C_{w,t-1}\sqrt{(t-1)_c}\right)^2$, with $(t-1)_c$ being the number of corruptions up to step $t - 1$.*

*Proof.* We use the triangle inequality:

$$
\begin{aligned}
|f(\boldsymbol{x}) - \mu_{t-1}^{R}(\boldsymbol{x})| &= |(f(\boldsymbol{x}) - \mu_{\text{uc},t-1}(\boldsymbol{x})) + (\mu_{\text{uc},t-1}(\boldsymbol{x}) - \mu_{t-1}^{R}(\boldsymbol{x}))| \\
&\leq |f(\boldsymbol{x}) - \mu_{\text{uc},t-1}(\boldsymbol{x})| + |d_{t-1}(\boldsymbol{x})|.
\end{aligned}
$$

Conditional on $E_{\text{conf}}(\delta')$, the standard GP confidence interval holds for the uncorrupted posterior. For a posterior based on $(t-1)_{\text{uc}}$ points, the bound uses the parameter $\beta_t'$. Thus, the first term is bounded by $\sqrt{\beta_{t_{\text{uc}}}'(\delta')}\sigma_{\text{uc},t-1}(\boldsymbol{x})$, which is, in turn, bounded by $\sqrt{\beta_t'(\delta')}\sigma_{\text{uc},t-1}(\boldsymbol{x})$ as $\beta_t'$ is non-decreasing in $t$. By Lemma D.3, the second term, $|d_{t-1}(\boldsymbol{x}))|$, is bounded by $C_{w,t-1}\sqrt{(t-1)_c}\sigma_{\text{uc},t-1}(\boldsymbol{x})$. Combining these bounds gives the desired result:

$$
\begin{aligned}
|f(\boldsymbol{x}) - \mu_{t-1}^{R}(\boldsymbol{x})| &\leq \sqrt{\beta_t'(\delta')}\sigma_{\text{uc},t-1}(\boldsymbol{x}) + C_{w,t-1}\sqrt{(t-1)_c}\sigma_{\text{uc},t-1}(\boldsymbol{x}) \\
&= \left(\sqrt{\beta_t'(\delta')} + C_{w,t-1}\sqrt{(t-1)_c}\right)\sigma_{\text{uc},t-1}(\boldsymbol{x}) \\
&= \sqrt{\beta_t(\delta')}\sigma_{\text{uc},t-1}(\boldsymbol{x}).
\end{aligned}
$$

$\square$

### D.3.2. OBSERVABLE BOUNDS OF THE POSTERIOR DEVIATION

Lemma D.4 provides a confidence interval encompassing the objective function $f$. However, it depends on the quantity $\sigma_{\text{uc},t-1}(x)$ which is a theoretical and not observable in practice. In this section, we provide bounds on $\sigma_{\text{uc},t-1}(x)$ using the observable quantity $\sigma_{t-1}^{R}(x)$. Lemma D.5 provides a relationship between the uncorrupted posterior variance $\sigma_{\text{uc},t-1}(x)$ and the observable RCGP posterior variance $\sigma_{t-1}^{R}(x)$. Lemma D.6 uses this relationship to provide an observable upper bound on $\sigma_{\text{uc},t-1}(x)$.

**Lemma D.5.** *Let $\boldsymbol{u}(\boldsymbol{x}) = \text{cov}_{\mathcal{D}_t}^{R}(\boldsymbol{x}, \boldsymbol{X}_{c,t})$ be the robust posterior covariance between a query point $\boldsymbol{x}$ and the set of corrupted observations $\boldsymbol{X}_{c,t}$. Let $\boldsymbol{C}_0 = \text{cov}_{\mathcal{D}_{uc,t}}(\boldsymbol{X}_{c,t}, \boldsymbol{X}_{c,t})$ be the covariance of the corrupted inputs conditioned on the uncorrupted dataset $\mathcal{D}_{uc,t}$, and let $\boldsymbol{\Lambda}_c = \sigma_{noise}^2 \boldsymbol{J}_{w,c,t}$ be the diagonal noise matrix derived from the weights of the corrupted points.*

*The uncorrupted posterior variance $\sigma_{uc,t}^2(\boldsymbol{x})$ is related to the RCGP posterior variance $\sigma_t^R(\boldsymbol{x})^2$ by:*

$$\sigma_{uc,t}^2(\boldsymbol{x}) = \sigma_t^R(\boldsymbol{x})^2 + \boldsymbol{u}(\boldsymbol{x})^T \left(\boldsymbol{\Lambda}_c^{-1} + \boldsymbol{\Lambda}_c^{-1}\boldsymbol{C}_0\boldsymbol{\Lambda}_c^{-1}\right)\boldsymbol{u}(\boldsymbol{x}) \tag{10}$$

*Proof.* For ease of notation, we will denote $\boldsymbol{u}(\boldsymbol{x}) = \boldsymbol{u}$. Dropping the argument $\boldsymbol{x}$ when it is unambiguous.

**Step 1: The Variance Update**
We consider the standard Gaussian Process update where the RCGP posterior is obtained by adding the set of corrupted observations $\boldsymbol{X}_{\text{c},t}$ (with effective noise variance $\boldsymbol{\Lambda}_c$) to the uncorrupted model. Let $\boldsymbol{w} = \text{cov}_{\mathcal{D}_{\text{uc},t}}(\boldsymbol{x}, \boldsymbol{X}_{\text{c},t})$ denote the covariance vector between $\boldsymbol{x}$ and $\boldsymbol{X}_{\text{c},t}$ conditioned on the uncorrupted data. This variance vector is also a function of $\boldsymbol{x}$ which was dropped in the notation.

The variance update is given by:

$$\sigma_t^R(\boldsymbol{x})^2 = \sigma_{\text{uc},t}^2(\boldsymbol{x}) - \boldsymbol{w}^T \boldsymbol{S}^{-1}\boldsymbol{w}$$

where $\boldsymbol{S} = \boldsymbol{C}_0 + \boldsymbol{\Lambda}_c$ is the Schur complement. Rearranging to isolate the uncorrupted variance:

$$\sigma_{\text{uc},t}^2(\boldsymbol{x}) = \sigma_t^R(\boldsymbol{x})^2 + \boldsymbol{w}^T \boldsymbol{S}^{-1}\boldsymbol{w} \tag{11}$$

**Step 2: The Covariance Vector Update**

The update rule for the posterior covariance vector between $x$ and the new observations $X_{c,t}$ is:

$$u^T = w^T - w^T S^{-1} C_0$$

Factoring out $w^T$:

$$u^T = w^T \left[ I - S^{-1} C_0 \right]$$

Substituting $S = C_0 + \Lambda_c$:

$$u^T = w^T \left[ I - (C_0 + \Lambda_c)^{-1} C_0 \right]$$

Using the matrix identity $I - (A + B)^{-1} A = (A + B)^{-1} B$, we obtain:

$$u^T = w^T (C_0 + \Lambda_c)^{-1} \Lambda_c = w^T S^{-1} \Lambda_c$$

Transposing (and using the symmetry of $S$) allows us to express $w$ in terms of the observable covariance $u$:

$$w = S \Lambda_c^{-1} u \tag{12}$$

**Step 3: Deriving final expression**

We substitute Equation 12 into the update term from Equation 11:

$$
\begin{aligned}
w^T S^{-1} w &= \left( u^T \Lambda_c^{-1} S \right) S^{-1} \left( S \Lambda_c^{-1} u \right) \\
&= u^T \Lambda_c^{-1} S \Lambda_c^{-1} u && \text{(Since } SS^{-1} = I\text{)} \\
&= u^T \Lambda_c^{-1} (C_0 + \Lambda_c) \Lambda_c^{-1} u && \text{(Substitute } S = C_0 + \Lambda_c\text{)} \\
&= u^T \left( \Lambda_c^{-1} C_0 \Lambda_c^{-1} + \Lambda_c^{-1} \right) u
\end{aligned}
$$

Substituting this back into Equation 11 completes the proof. $\qquad \square$

**Lemma D.6.** *Assuming Assumption B.2 holds, the uncorrupted posterior standard deviation $\sigma_{uc,t}(x)$ is bounded by the observable RCGP posterior standard deviation $\sigma_t^R(x)$ as:*

$$\sigma_{uc,t}(x) \le \sigma_t^R(x) \cdot \sqrt{1 + \frac{t_c \kappa}{\sigma_{noise}^2} \left( 1 + \frac{t_c \kappa}{\sigma_{noise}^2} \right)}$$

*Proof.* Using the result of the previous lemma (Equation 10), we bound the quadratic form involving $u$. Recall that $\Lambda_c = \sigma_{noise}^2 J_{w,c,t}$. Since the weights $w_i$ are strictly positive and upper bounded by $W = \frac{\sigma_{noise}}{\sqrt{2}}$, $\Lambda_c$ is positive definite and satisfies $\Lambda_c^{-1} \preceq \sigma_{noise}^{-2} I$. Furthermore, since $C_0$ is a covariance matrix, its maximum eigenvalue is bounded by its trace: $\lambda_{\max}(C_0) \le \text{tr}(C_0) \le t_c \kappa$, where $t_c = |\mathcal{D}_{c,t}|$ is the number of corrupted points and $\kappa$ is the kernel bound.

We bound the term $u^T \left( \Lambda_c^{-1} + \Lambda_c^{-1} C_0 \Lambda_c^{-1} \right) u$:

$$
\begin{aligned}
u^T \left( \Lambda_c^{-1} + \Lambda_c^{-1} C_0 \Lambda_c^{-1} \right) u &\le \lambda_{\max} \left( \Lambda_c^{-1} + \Lambda_c^{-1} C_0 \Lambda_c^{-1} \right) \|u\|_2^2 \\
&\le \left( \frac{1}{\sigma_{noise}^2} + \frac{t_c \kappa}{\sigma_{noise}^4} \right) \|u\|_2^2 \\
&= \left( \frac{1}{\sigma_{noise}^2} + \frac{t_c \kappa}{\sigma_{noise}^4} \right) \sum_{i \in \mathcal{D}_{c,t}} \text{cov}_{\mathcal{D}_t}^R(x, x_i)^2 && \text{(Definition of } u\text{)} \\
&\le \left( \frac{1}{\sigma_{noise}^2} + \frac{t_c \kappa}{\sigma_{noise}^4} \right) \sum_{i \in \mathcal{D}_{c,t}} \sigma_t^R(x)^2 \sigma_t^R(x_i)^2 && \text{(Cauchy-Schwarz on Covariance)}
\end{aligned}
$$

Using the property that $\sigma_t^R(x_i)^2 \le \kappa$ (prior variance bound):

$$u^T \left( \Lambda_c^{-1} + \Lambda_c^{-1} C_0 \Lambda_c^{-1} \right) u \le \sigma_t^R(x)^2 \left( \frac{1}{\sigma_{noise}^2} + \frac{t_c \kappa}{\sigma_{noise}^4} \right) t_c \kappa$$

Substituting this back into Equation 10:

$$\sigma_{\text{uc},t}^2(\boldsymbol{x}) \leq \sigma_t^{\text{R}}(\boldsymbol{x})^2 \left[ 1 + \frac{t_{\text{c}}\kappa}{\sigma_{\text{noise}}^2} + \left( \frac{t_{\text{c}}\kappa}{\sigma_{\text{noise}}^2} \right)^2 \right]$$

Taking the square root yields the statement of the lemma. $\qquad\square$

**Lemma D.7.** *Let's assume that Assumptions B.1 and B.2 hold and the event $E_{conf}$ is realized. Let's also assume that the high-probability plateau condition holds. Then, for any $t \geq 1$, the RCGP posterior computed after step $t-1$ satisfies:*

$$|f(\boldsymbol{x}) - \mu_{t-1}^R(\boldsymbol{x})| \leq \sqrt{\beta_t(\delta')}\Psi((t-1)_c)\sigma_{t-1}^R(\boldsymbol{x}),$$

*where $\Psi((t-1)_c) = \sqrt{1 + \frac{(t-1)_c\kappa}{\sigma_{noise}^2}\left(1 + \frac{(t-1)_c\kappa}{\sigma_{noise}^2}\right)}$, and $(t-1)_c$ is the number of corruptions up to step $t-1$.*

*Proof.* This is a straightforward application of the bound in Lemma D.6 to the confidence interval of Lemma D.4 $\qquad\square$

### D.4. Analysis of FC-RCGP-UCB (Algorithm 1)

#### D.4.1. ALGORITHM CONFIGURATION AND EVENT

FC-RCGP-UCB uses a probability budget of $\delta$. We define the joint event $E_{\text{Plateau}} = E_{\text{noise}}(\delta/3) \cap E_{\text{conf}}(\delta/3) \cap E_f(\delta/3)$. By a union bound, $\mathbb{P}(E_{\text{Plateau}}) \geq 1 - \delta$. The analysis is conditional on $E_{\text{Plateau}}$.

- Center: $g(\boldsymbol{x}) = 0$.
- Fixed plateau width: $L_T = \sqrt{B_f\kappa} + N_T(\delta/3)$.

#### D.4.2. REGRET ANALYSIS

The technical lemmas from Section D.3 rely on the plateau condition. We first prove that this condition holds for FC-RCGP-UCB when the events $E_{\text{noise}}(\delta/3)$ and $E_f(\delta/3)$ are realised.

**Lemma D.8.** *Let's assume Assumptions B.1 and B.3 hold. If the events $E_{noise}(\delta/3)$ and $E_f(\delta/3)$ are realised, then the plateau condition holds for FC-RCGP-UCB for all $t \leq T$.*

*Proof.* Consider an uncorrupted observation $y_t = f(\boldsymbol{x}_t) + \epsilon_t$. The distance to the center $g(\boldsymbol{x}) = 0$ is $|y_t|$.

$$|y_t| \leq |f(\boldsymbol{x}_t)| + |\epsilon_t|.$$

By Assumption B.3 and the event $E_f(\delta/3)$, $|f(\boldsymbol{x}_t)| \leq \sqrt{B_f\kappa}$. By Assumption B.1 and the event $E_{\text{noise}}(\delta/3)$, $|\epsilon_t| \leq N_T(\delta/3)$.

$$|y_t| \leq \sqrt{B_f\kappa} + N_T(\delta/3) = L_T.$$

As such, the uncorrupted observations are within the plateau satisfying the plateau condition. $\qquad\square$

**Lemma D.9.** *Let's assume that Assumptions B.1, B.2, and B.3 hold. Further, let's assume that the event $E_f(\delta/3)$ is realised. Then, the deviation constant $C_{w,T}$ for FC-RCGP-UCB scales as follows: $C_{w,T} = \tilde{\mathcal{O}}\left(\sqrt{\beta_T'}\right)$.*

*Proof.* The deviation constant $C_{w,T} = \frac{\sqrt{2}C_{1,T}}{\sigma_{\text{noise}}^2}$ scales linearly with $C_{1,T}$. Lemma G.1 shows that $C_{1,T}$ scales linearly with the width $L_T$ and the center deviation $\Delta(x)$.

- Width Scaling: $L_T = \sqrt{B_f\kappa} + \mathcal{O}(\sqrt{\log(T/\delta)}) = \mathcal{O}(\sqrt{\log(T/\delta)}) = \tilde{\mathcal{O}}(1)$.
- Center Deviation: $\Delta(\boldsymbol{x}) = |g(\boldsymbol{x}) - \mu_{\text{uc}}(\boldsymbol{x})| = |0 - \mu_{\text{uc}}(\boldsymbol{x})|$. By Lemma D.1, this is bounded by a constant $M_T = \tilde{\mathcal{O}}\left(\sqrt{\beta_T'}\right)$.

Therefore, $C_{w,T} = \mathcal{O}(L_T + M_T) = \tilde{\mathcal{O}}\left(\sqrt{\beta_T'}\right)$. $\qquad\square$

We now have all the ingredients to bound the cumulative regret of FC-RCGP-UCB.

**Theorem D.10** (Regret Bound for FC-RCGP-UCB). *Under Assumptions B.1, B.2, and B.3, with probability at least $1 - \delta$, the cumulative regret of FC-RCGP-UCB is bounded by:*

$$R_T = \tilde{\mathcal{O}} \left( \Psi(T_c) \left( 1 + \sqrt{T_c} \right) \sqrt{\beta_T' T(\gamma_T + T_c)} \right).$$

*Proof.* We analyze the regret conditional on the high-probability event $E_{\text{Plateau}} = E_{\text{noise}}(\delta/3) \cap E_{\text{conf}}(\delta/3) \cap E_f(\delta/3)$.

*1. Validation of Prerequisites:* Since $E_{\text{noise}}(\delta/3)$ and $E_f(\delta/3)$ hold, Lemma D.8 validates the plateau condition, allowing the use of the Technical Lemmas.

*2. Confidence Multiplier Scaling:* The robust confidence multiplier at step $t$ is $\sqrt{\beta_t} = (\sqrt{\beta_t'(\delta/3)} + C_{w,T}\sqrt{T_c})$. The sequence $\sqrt{\beta_t}$ is non-decreasing in $t$. The scaling of the multiplier is determined by its final value:

$$\sqrt{\beta_T} = \tilde{\mathcal{O}} \left( \sqrt{\beta_T'} \right) + \tilde{\mathcal{O}} \left( \sqrt{\beta_T'} \right) \sqrt{T_c} = \tilde{\mathcal{O}} \left( \sqrt{\beta_T'}(1 + \sqrt{T_c}) \right).$$

Where we injected $C_{w,T} = \tilde{\mathcal{O}} \left( \sqrt{\beta_T'} \right)$ from Lemma D.9

*3. Regret Decomposition:* The instantaneous regret is $r_t = f(x^*) - f(\boldsymbol{x}_t)$. We bound this by exploiting the confidence intervals and the UCB selection rule. The confidence intervals apply because the event $E_{\text{Plateau}}$ (which implies $E_{\text{conf}}(\delta/3)$) holds. Which allows using the enlarged condidence bound from Lemma D.7. The UCB/LCB confidence bound is as follows:

$$|f(\boldsymbol{x}) - \mu_{t-1}^{\text{R}}(\boldsymbol{x})| \leq \sqrt{\beta_t}\Psi((t-1)_c)\sigma_{t-1}^{\text{R}}(\boldsymbol{x}) \tag{13}$$

We also remind that $\boldsymbol{x}_t$ is the point selected by the UCB algorithm, as such:

$$\boldsymbol{x}_t = \underset{\boldsymbol{x}}{\text{argmax}}\, \mu_{t-1}^{\text{R}}(\boldsymbol{x}) + \sqrt{\beta_t}\Psi((t-1)_c)\sigma_{t-1}^{\text{R}}(\boldsymbol{x}) \tag{14}$$

The instantaneous regret is then bounded as follows:

$$r_t = f(\boldsymbol{x}^*) - f(\boldsymbol{x}_t) \leq \left( \mu_{t-1}^{\text{R}}(\boldsymbol{x}^*) + \sqrt{\beta_t}\Psi((t-1)_c)\sigma_{t-1}^{\text{R}}(\boldsymbol{x}^*) \right) - f(\boldsymbol{x}_t)$$
$$\text{(Step 1: Apply the UCB bound 13 on } f(\boldsymbol{x}^*)\text{)}$$
$$\leq \left( \mu_{t-1}^{\text{R}}(\boldsymbol{x}_t) + \sqrt{\beta_t}\Psi((t-1)_c)\sigma_{t-1}^{\text{R}}(\boldsymbol{x}_t) \right) - f(\boldsymbol{x}_t)$$
$$\text{(Step 2: Apply the UCB selection property 14)}$$
$$= (\mu_{t-1}^{\text{R}}(\boldsymbol{x}_t) - f(\boldsymbol{x}_t)) + \sqrt{\beta_t}\Psi((t-1)_c)\sigma_{t-1}^{\text{R}}(\boldsymbol{x}_t)$$
$$\text{(Step 3: Rearrange terms)}$$
$$\leq \sqrt{\beta_t}\Psi((t-1)_c)\sigma_{t-1}^{\text{R}}(\boldsymbol{x}_t) + \sqrt{\beta_t}\Psi((t-1)_c)\sigma_{t-1}^{\text{R}}(\boldsymbol{x}_t)$$
$$\text{(Step 4: Apply the LCB on } f(\boldsymbol{x}_t) \text{ from Eq. 13)}$$
$$= 2\sqrt{\beta_t}\Psi((t-1)_c)\sigma_{t-1}^{\text{R}}(\boldsymbol{x}_t).$$

To bound the cumulative regret, we sum the instantaneous regrets. Since the sequences $\sqrt{\beta_t}$ and $\Psi((t-1)_c)$ are non-decreasing, we can bound each $\sqrt{\beta_t}$ and $\Psi((t-1)_c)$ by their respective terminal values $\sqrt{\beta_T}$ and $\Psi((T-1)_c)$:

$$R_T = \sum_{t=1}^{T} r_t \leq \sum_{t=1}^{T} 2\sqrt{\beta_t}\Psi((t-1)_c)\sigma_{t-1}^{\text{R}}(\boldsymbol{x}_t) \leq 2\sqrt{\beta_T}\Psi((T-1)_c)\sum_{t=1}^{T}\sigma_{t-1}^{\text{R}}(\boldsymbol{x}_t).$$

Applying the Cauchy-Schwarz inequality then gives:

$$R_T \leq 2\sqrt{\beta_T}\Psi((T-1)_c)\sqrt{T\sum_{t=1}^{T}\sigma_{t-1}^{\text{R}}{}^2(\boldsymbol{x}_t)}.$$

*4. Bounding Sum of Variances:* We start by bounding $\sigma_{t-1}^{R}{}^{2}(\boldsymbol{x})$ by $\sigma_{\text{uc},t-1}^{2}(\boldsymbol{x})$. In fact, $\sigma_{t-1}^{R}{}^{2}(\boldsymbol{x})$ is the posterior variance after conditioning on all of the observed data up to step $t-1$ while $\sigma_{\text{uc},t-1}^{2}(\boldsymbol{x})$ is the posterior variance after conditioning the subset of data containing uncorrupted data only. As such we write:

$$\sum_{t=1}^{T} \sigma_{t-1}^{R}{}^{2}(\boldsymbol{x}_t) \leq \sum_{t=1}^{T} \sigma_{\text{uc},t-1}^{2}(\boldsymbol{x}_t)$$

The sum of variances is partitioned into contributions from uncorrupted and corrupted points:

$$\sum_{t=1}^{T} \sigma_{\text{uc},t-1}^{2}(\boldsymbol{x}_t) = \sum_{t\in\mathcal{D}_{\text{uc}}} \sigma_{\text{uc},t-1}^{2}(\boldsymbol{x}_t) + \sum_{t\in\mathcal{D}_{\text{c}}} \sigma_{\text{uc},t-1}^{2}(\boldsymbol{x}_t)$$

The first term exactly correspond to the sum of variances observed by standard GP if it only encountered uncorrupted data over a horizon of $T_{\text{uc}}$ steps. The term can be bounded by the maximum information gain $\gamma_{T_{\text{uc}}}$ which is bounded by $\gamma_T$. The second term can be bounded by $T_{\text{c}}\kappa$ since it any individual variance in the sum can be bounded by $\kappa$. This sum is bounded by $\tilde{\mathcal{O}}\left(\gamma_T + T_{\text{c}}\right)$.

*5. Final Bound:* Substituting the scaling of $\sqrt{\beta_T} = \tilde{\mathcal{O}}\left(\sqrt{\beta_T'}(1 + \sqrt{T_{\text{c}}})\right)$ and the bound on the sum of variances yields the general regret bound stated in the theorem:

$$R_T = \tilde{\mathcal{O}}\left(\Psi(T_{\text{c}})\left(1 + \sqrt{T_{\text{c}}}\right)\sqrt{\beta_T' T(\gamma_T + T_{\text{c}})}\right).$$

$\square$

### D.4.3. CONDITIONS FOR SUB-LINEAR REGRET

We briefly discuss when the regret obtained in Theorem D.10 is sub-linear. We distinguish the following cases:

- For kernels with slow information growth,such as the RBF kernel: $\gamma_T = \tilde{\mathcal{O}}\left(1\right)$, $\beta_T' = \tilde{\mathcal{O}}\left(1\right)$ for all three cases of the domain $\mathcal{X}$ discussed in Assumption B.1. The regret is then $R_T = \tilde{\mathcal{O}}\left(T_{\text{c}}^2\sqrt{T}\right)$ which is sublinear for $T_c = \mathcal{O}(T^\alpha)$ with $0 < \alpha < 1/4$.

- For kernels with fast information growth, such as the Matérn kernel: $\gamma_T = \tilde{\mathcal{O}}\left(T^\eta\right)$. We need to consider the three cases of the domain $\mathcal{X}$ discussed in Assumption B.1:

    - **Cases 1 and 2:** $\beta_T' = \tilde{\mathcal{O}}\left(1\right)$. In this scenario, the regret is sublinear for $T_c = \mathcal{O}(T^\alpha)$ with $0 < \alpha < \min(\frac{1}{4}, \frac{1-\eta}{3})$.
    - **Cases 3:** $\beta_T' = \tilde{\mathcal{O}}\left(\gamma_T\right)$. In this scenario, the regret is sublinear for $T_c = \mathcal{O}(T^\alpha)$ with $0 < \alpha < \min(\frac{1-\eta}{4}, \frac{1-2\eta}{3})$.

### D.5. Analysis of A2-RCGP-UCB (Algorithm 2)

The FC-RCGP-UCB algorithm enjoys sub-linear regret even against a strong adversary which can corrupt up to $T^{1/4}$ observations with infinite magnitude. However, it might be too conservative in practice given the rigid nature of the Plateau-IMQ's fixed center. The A2-RCGP-UCB algorithm trades some of the theoretical allowed corruption budget for more adaptability by introducing two models: a stable Anchor model ($M_A$) and an adaptive Acquisition model ($M_R$).

### D.5.1. ALGORITHM CONFIGURATION AND EVENTS

**Model $M_A$ (Anchor) Configuration (Identical to FC)**

- Center: $g_A(\boldsymbol{x}) = 0$. Width: $L_{A,T} = \sqrt{B_f\kappa} + N_T(\delta/3)$.

**Model $M_R$ (Acquisition) Configuration (Adaptive)**

- Center: $g_{R,t}(\boldsymbol{x}) = \mu_{A,t-1}(\boldsymbol{x})$.

- Width (Adaptive): $L_{R,t}(\boldsymbol{x}) = \sqrt{\beta_{A,t}} \Psi((t-1)_c) \sigma_{A,uc,t-1}^R(\boldsymbol{x}) + N_T(\delta/3)$.

  **Note:** In theory, we could set $L_{R,t}(\boldsymbol{x}) = \sqrt{\beta_{A,t}} \sigma_{A,uc,t-1}^R(\boldsymbol{x}) + N_T(\delta/3)$ which would yield tighter plateau regions and thus better allowed corruption budgets $T_c$. However, $\sigma_{A,uc,t-1}(\boldsymbol{x})$ is not observable, and we need to use the pessimistic, but observable, bound from Lemma D.6.

### D.5.2. STAGE 1: ANALYSIS OF THE ANCHOR MODEL ($M_A$)

**Lemma D.11.** *Let's assume that Assumptions B.1, B.2, and B.3 hold. Conditional on the event $E_{Plateau}$:*

1. *The plateau condition holds for $M_A$.*

2. *The deviation constant $C_{w,A,T}$ scales as $\tilde{\mathcal{O}}\left(\sqrt{\beta_T'}\right)$.*

3. *Let the robust confidence parameter for the Anchor model be $\beta_{A,t}$. The Anchor Bound holds for any $t \geq 1$: $|f(\boldsymbol{x}) - \mu_{A,t-1}^R(\boldsymbol{x})| \leq \sqrt{\beta_{A,t}} \Psi((t-1)_c) \sigma_{A,uc,t-1}^R(\boldsymbol{x})$.*

4. *The scaling of the robust confidence multiplier is $\sqrt{\beta_{A,t}} = \tilde{\mathcal{O}}\left(\sqrt{\beta_T'}\left(1 + \sqrt{T_c}\right)\right)$.*

*Proof.* $M_A$ is configured identically to FC-RCGP-UCB. The proof follows directly from the analysis of FC-RCGP-UCB.
1. The plateau condition holds for $M_A$ by Lemma D.8 applied to $M_A$.
2. $C_{w,A,T} = \tilde{\mathcal{O}}\left(\sqrt{\beta_T'}\right)$ holds by Lemma D.9 applied to $M_A$.
3. The Anchor Bound holds by applying Lemma D.7 to $M_A$.
4. The robust confidence multiplier is $\sqrt{\beta_{A,t}} = (\sqrt{\beta_t'(\delta/3)} + C_{w,A,T}\sqrt{T_c})$. Substituting the scaling of $C_{w,A,T}$ yields $\sqrt{\beta_{A,t}} = \tilde{\mathcal{O}}\left(\sqrt{\beta_T'}\left(1 + \sqrt{T_c}\right)\right)$. $\qquad\square$

### D.5.3. STAGE 2: ANALYSIS OF THE ADAPTIVE MODEL ($M_R$)

We use the guarantees from the stable Anchor model $M_A$ to validate the adaptive Acquisition model $M_R$.

**Lemma D.12.** *Conditional on the event $E_{Plateau}$, the plateau condition holds for $M_R$ using the theoretically defined adaptive width $L_{R,t}(x)$.*

*Proof.* We must show that for an uncorrupted observation $y_t = f(\boldsymbol{x}_t) + \epsilon_t$, the distance to the center $|y_t - g_{R,t}(\boldsymbol{x}_t)|$ is less than the theoretical adaptive width $L_{R,t}(\boldsymbol{x}_t)$. The center is $g_{R,t}(\boldsymbol{x}_t) = \mu_{A,t-1}(\boldsymbol{x}_t)$.

We bound the distance:

$$|y_t - g_{R,t}(\boldsymbol{x}_t)| \leq |f(\boldsymbol{x}_t) - \mu_{A,t-1}(\boldsymbol{x}_t)| + |\epsilon_t|.$$

Conditional on $E_{Plateau}$, the noise is bounded by $N_T(\delta/3)$, and by the Anchor Bound (Lemma D.11, item 3), the first term is bounded by $\sqrt{\beta_{A,t}} \Psi((t-1)_c) \sigma_{A,uc,t-1}^R(\boldsymbol{x}_t)$. Thus:

$$|y_t - g_{R,t}(\boldsymbol{x}_t)| \leq \sqrt{\beta_{A,t}} \Psi((t-1)_c) \sigma_{A,uc,t-1}^R(\boldsymbol{x}_t) + N_T(\delta/3).$$

The theoretical adaptive width is $L_{R,t}(\boldsymbol{x}_t) = \sqrt{\beta_{A,t}} \Psi((t-1)_c) \sigma_{A,uc,t-1}^R(\boldsymbol{x}_t) + N_T(\delta/3)$. Therefore, the distance to the center is bounded by the width, and the observation is within the plateau. This proves $J_{uc,R} = I$. $\qquad\square$

**Lemma D.13.** *Under Assumptions B.1, B.2, and B.3, the deviation constant $C_{w,R,T}$ for $M_R$ scales as $\tilde{\mathcal{O}}\left(\Psi(T_c)\sqrt{\beta_T' T_c}\right)$.*

*Proof.* The deviation constant $C_{w,R,T}$ scales with $C_{1,T}$, which depends on the maximum plateau width and the maximum center deviation over the domain and time horizon by Lemma G.1.

**1. Maximum Width Scaling:**

$$\sup_{x,t} L_{R,t}(\boldsymbol{x}) = \sup_{x,t}(\sqrt{\beta_{A,t}} \Psi((t-1)_c) \sigma_{A,uc,t-1}^R(\boldsymbol{x}) + N_T(\delta/3)).$$

The variance is always bounded by the prior maximum $\sqrt{\kappa}$. Moreover, the sequences $\Psi((t-1)_c)$ and $\sqrt{\beta_{A,t}}$ are increasing:

$$\sup_{x,t} L_{R,t}(\boldsymbol{x}) \leq \Psi((T-1)_c)\sqrt{\beta_{A,T}\kappa} + N_T(\delta/3).$$

By Lemma D.11 (item 4), $\sqrt{\beta_{A,T}} = \tilde{\mathcal{O}}\left(\sqrt{\beta'_T}\left(1 + \sqrt{T_c}\right)\right)$. Therefore, the maximum width scales as $\tilde{\mathcal{O}}\left(\Psi(T_c)\sqrt{\beta'_T}\left(1 + \sqrt{T_c}\right)\right)$.

**2. Maximum Center Deviation Scaling:**

$$\Delta_R(\boldsymbol{x}) = |g_{R,t}(\boldsymbol{x}) - \mu_{\text{uc,t-1}}(\boldsymbol{x})| = |\mu_{A,t-1}(\boldsymbol{x}) - \mu_{\text{uc,t-1}}(\boldsymbol{x})|.$$

This is the deviation of the Anchor model, $d_{A,t-1}(x)$. We bound this using Lemma D.3 applied to $M_A$:

$$\sup_{x,t} \Delta_R(\boldsymbol{x}) \le C_{w,A,T}\sqrt{T_c}\sup_x \sigma_{\text{uc},t-1}(\boldsymbol{x}) \le C_{w,A,T}\sqrt{T_c}\sqrt{\kappa}.$$

By Lemma D.11 (item 2), $C_{w,A,T} = \tilde{\mathcal{O}}(\sqrt{\beta'_T})$. Therefore, the maximum center deviation scales as $\tilde{\mathcal{O}}(\sqrt{\beta'_T})$.

**3. Conclusion:** The maximum width scaling is the dominant term. Therefore, the deviation constant $C_{w,R,T} = \tilde{\mathcal{O}}\left(\Psi(T_c)\sqrt{\beta'_T}\left(1 + \sqrt{T_c}\right)\right)$ by Lemma G.1. $\qquad\square$

**Lemma D.14.** *Conditional on $E_{Plateau}$. Let the robust confidence parameter for the Acquisition model be $\beta_{R,t}$. For any $t \ge 1$, the confidence bound for $M_R$ holds: $|f(\boldsymbol{x}) - \mu_{R,t-1}(\boldsymbol{x})| \le \sqrt{\beta_{R,t}}\Psi((t-1)_c)\sigma^R_{R,t-1}(\boldsymbol{x})$. The scaling of the robust confidence multiplier is $\sqrt{\beta_{R,t}} = \tilde{\mathcal{O}}\left(\sqrt{\beta'_T}\left(1 + T_c\Psi(T_c)\right)\right)$.*

*Proof.* Conditional on $E_{\text{Plateau}}$, Lemma D.12 shows that the plateau condition holds for $M_R$. This validates the use of Lemma D.7 for $M_R$. The robust confidence multiplier is $\sqrt{\beta_{R,t}} = (\sqrt{\beta'_t(\delta/3)} + C_{w,R,T}\sqrt{T_c})$. We substitute the scaling of $C_{w,R,T}$ from Lemma D.13:

$$\sqrt{\beta_{R,t}} = \tilde{\mathcal{O}}(\sqrt{\beta'_T}) + \left(\tilde{\mathcal{O}}\left(\Psi(T_c)\sqrt{\beta'_T}\left(1 + \sqrt{T_c}\right) \cdot \sqrt{T_c}\right)\right) = \tilde{\mathcal{O}}\left(\sqrt{\beta'_T}\left(1 + T_c\Psi(T_c)\right)\right).$$

$\qquad\square$

### D.5.4. Stage 3: Regret Analysis

**Theorem D.15** (Regret Bound for A2-RCGP-UCB)**.** *Under Assumptions B.1, B.2, and B.3, with probability at least $1 - \delta$, the cumulative regret of A2-RCGP-UCB is bounded by:*

$$R_T = \tilde{\mathcal{O}}\left(\left(1 + T_c\Psi(T_c)^2\right)\sqrt{\beta'_T T(\gamma_T + T_c)}\right).$$

*Proof.* We analyze the regret conditional on the high-probability event $E_{\text{Plateau}}$. The analysis follows the structure of Theorem D.10, using the acquisition model $M_R$. The cumulative regret is bounded by:

$$R_T = \tilde{\mathcal{O}}(\sqrt{\beta_{R,T}}\Psi((T-1)_c)\sqrt{T(\gamma_T + T_c)}).$$

We substitute the scaling of the multiplier $\sqrt{\beta_{R,T}}$ from Lemma D.14, $\sqrt{\beta_{R,T}} = \tilde{\mathcal{O}}\left(\sqrt{\beta'_T}\left(1 + T_c\Psi(T_c)\right)\right)$ and use $\Psi(T_c) = \tilde{\mathcal{O}}\left(1 + T_c\right)$ and $\Psi(T_c) \ge 1$ to obtain:

$$R_T = \tilde{\mathcal{O}}\left(\left(1 + T_c\Psi(T_c)^2\right)\sqrt{\beta'_T T(\gamma_T + T_c)}\right).$$

$\qquad\square$

### D.5.5. Conditions for sub-linear regret

We briefly discuss when the regret obtained in Theorem D.15 is sub-linear. We distinguish the following cases:

- For kernels with slow information growth, such as the RBF kernel: $\gamma_T = \tilde{\mathcal{O}}(1)$, $\beta'_T = \tilde{\mathcal{O}}(1)$ for all three cases of the domain $\mathcal{X}$ discussed in Assumption B.1. The regret is then $R_T = \tilde{\mathcal{O}}\left(T_c^{7/2}\sqrt{T}\right)$ which is sublinear for $T_c = \mathcal{O}(T^\alpha)$ with $0 < \alpha < 1/7$.

- For kernels with fast information growth, such as the Matérn kernel: $\gamma_T = \tilde{\mathcal{O}}(T^\eta)$. We need to consider the three cases of the domain $\mathcal{X}$ discussed in Assumption B.1:

  - **Cases 1 and 2:** $\beta'_T = \tilde{\mathcal{O}}(1)$. In this scenario, the regret is sublinear for $T_c = \mathcal{O}(T^\alpha)$ with $0 < \alpha < \min(\frac{1}{7}, \frac{1-\eta}{6})$.
  - **Cases 3:** $\beta'_T = \tilde{\mathcal{O}}(\gamma_T)$. In this scenario, the regret is sublinear for $T_c = \mathcal{O}(T^\alpha)$ with $0 < \alpha < \min(\frac{1-\eta}{7}, \frac{1-2\eta}{6})$.

# E. Summary of Conditions for Sublinear Regrets

We briefly summarize the conditions on the corruption frequency $\alpha$ for the two RCGP-UCB algorithms to achieve sublinear regret. We distinguish scenarios where a a kernel with slow information gain is used such as the RBF where $\gamma_T = \tilde{\mathcal{O}}(1)$ and kernels with fast information gain such as the Matérn kernel where $\gamma_T = \tilde{\mathcal{O}}(T^\eta)$. We also need to distinguish between the three cases of the domain considered in Assumption B.1. We refer to Table 1 for a summary of the conditions.

*Table 1.* Comparison of permissible corruption budgets $T_c = O(T^\alpha)$ for sublinear regret. The "Current" column reflects the proven bounds including the observability penalty $\Psi(T_c)$. The "Ideal" column reflects the improved bounds if a tighter approximation of the uncorrupted variance $\sigma^2_{uc,t}$ was available.

| Kernel's Information Gain Setting | FC-RCGP-UCB | | A2-RCGP-UCB | |
| --- | --- | --- | --- | --- |
| | **Current** | **Ideal** | **Current** | **Ideal** |
| **Slow Information Gain** ($\gamma_T \approx \tilde{O}(1)$) | $\alpha < \frac{1}{4}$ | $\alpha < \frac{1}{2}$ | $\alpha < \frac{1}{7}$ | $\alpha < \frac{1}{3}$ |
| **Fast Information Gain** ($\gamma_T \approx O(T^\eta)$) | | | | |
| Cases 1 & 2 | $\min\left(\frac{1}{4}, \frac{1-\eta}{3}\right)$ | $\min\left(\frac{1}{2}, 1-\eta\right)$ | $\min\left(\frac{1}{7}, \frac{1-\eta}{6}\right)$ | $\min\left(\frac{1}{3}, \frac{1-\eta}{2}\right)$ |
| Case 3 | $\min\left(\frac{1-\eta}{4}, \frac{1-2\eta}{3}\right)$ | $\min\left(\frac{1-\eta}{2}, 1-2\eta\right)$ | $\min\left(\frac{1-\eta}{7}, \frac{1-2\eta}{6}\right)$ | $\min\left(\frac{1-\eta}{3}, \frac{1-2\eta}{2}\right)$ |

# F. Regret Analysis in the Well-Specified Case (Zero-Cost Robustness)

In this section, we analyze the performance of the FC-RCGP-UCB and A2-RCGP-UCB algorithms in the absence of adversarial corruptions ($T_c = 0$). We demonstrate that both algorithms achieve the same regret bounds as standard GP-UCB (Srinivas et al., 2010), thus providing robustness without sacrificing efficiency.

The analysis relies on the general framework established in Section D. We show that the well-specified results are direct corollaries of the general theorems when $T_c = 0$.

## F.1. Analysis of FC-RCGP-UCB (Well-Specified)

**Theorem F.1** (Regret Bound for FC-RCGP-UCB in the uncorrupted setting)**.** *Under Assumptions B.1, B.2, and B.3, in the uncorrupted setting ($T_c = 0$), the cumulative regret of FC-RCGP-UCB is bounded with probability at least $1 - \delta$ by:*

$$R_T = \mathcal{O}(\sqrt{T\beta'_T(\delta/3)\gamma_T}).$$

*This matches the standard GP-UCB regret bound.*

*Proof.* The proof follows directly from the analysis of the general case (Theorem D.10) by setting $T_c = 0$.

**1. Verification of Conditions:** The analysis in Section D relies on the high-probability event $E_{\text{Plateau}}$ (w.p. $1 - \delta$). Under this event, Lemma D.8 established that the plateau condition holds for any $T_c \geq 0$. This remains valid when $T_c = 0$ and serves as the prerequisite for the regret analysis.

**2. Regret Bound Derivation:** The general regret bound derived in Theorem D.10 is:

$$R_T = \tilde{\mathcal{O}}\left(\Psi(T_c)\left(1 + \sqrt{T_c}\right)\sqrt{\beta'_T T(\gamma_T + T_c)}\right)$$

Substituting $T_c = 0$ and using $\Psi(0) = 1$:

$$R_T = \mathcal{O}\left(\sqrt{T\beta'_T(\delta/3)\gamma_T}\right).$$

$\square$

## F.2. Analysis of A2-RCGP-UCB (Well-Specified)

**Theorem F.2** (Regret Bound for A2-RCGP-UCB in the uncorrupted setting). *Under Assumptions B.1, B.2, and B.3, in the uncorrupted setting ($T_c = 0$), the cumulative regret of A2-RCGP-UCB is bounded with probability at least $1 - \delta$ by:*

$$R_T = \mathcal{O}(\sqrt{T\beta'_T(\delta/3)\gamma_T}).$$

*This matches the standard GP-UCB regret bound.*

*Proof.* The proof follows from the general analysis (Theorem D.15) by setting $T_c = 0$.

**1. Verification of Conditions:** The analysis relies on the high-probability event $E_{\text{Plateau}}$ (w.p. $1 - \delta$). Under this event, Lemmas D.11 and D.12 established that the plateau condition holds for both $M_A$ and $M_R$ for any $T_c \geq 0$.

**2. Regret Bound Derivation:** The general regret bound derived in Theorem D.15 is:

$$R_T = \tilde{\mathcal{O}}\left(\left(1 + T_c\Psi(T_c)^2\right)\sqrt{\beta'_T T(\gamma_T + T_c)}\right).$$

Substituting $T_c = 0$ and using $\Psi(0) = 1$:

$$R_T = \tilde{\mathcal{O}}\left(\sqrt{\beta'_T T(\gamma_T)}\right).$$

$\square$

## F.3. Discussion: Algorithmic Behavior and Adaptivity

When $T_c = 0$ and the high-probability events hold, both algorithms not only recover the standard regret bounds but also behave identically to standard GP-UCB. Since the plateau condition holds (as proven in the general analysis), the RCGP updates are mathematically equivalent to standard GP updates.

It is worth noting that A2-RCGP-UCB retains a practical advantage over FC-RCGP-UCB even in this setting. FC-RCGP-UCB uses a fixed center $g(x) = 0$. If the prior is misspecified (e.g., the true function is far from zero), learning can be slow. A2-RCGP-UCB, however, adapts its center $g_{R,t}(x) = \mu_{A,t-1}(x)$ towards the true function, potentially leading to better empirical performance (smaller constants in the regret).

## G. Bounded Influence of the P-IMQ Function

The regret analysis relies on bounding the deviation between the robust posterior and the idealized uncorrupted posterior (Lemma D.3). The deviation bound is given in Lemma D.3 as $C_w\sqrt{T_c}\sigma_{uc,t}(x)$ where $C_w = \frac{\sqrt{2}C_1}{\sigma^2}$. To understand, how the $C_w$ depends on the timestep and on the P-IMQ properties, we need to study the term $C_1$:

$$C_1 = \sup_{x\in\mathcal{X},y\in\mathbb{R}} |w(x,y)(y - m_w(x,y) - \mu_{uc}(x))|$$

where $m_w(x,y) = \sigma_{\text{noise}}^2 \nabla_y \log(w^2(x,y))$ is the gradient correction term.

Recall the P-IMQ function with width $L$, center $g(x)$, and shape parameter $c$:

$$w(x,y) = \begin{cases} W_{max} & \text{if } |y - g(x)| \leq L \\ W_{max}\left(1 + \frac{(|y-g(x)|-L)^2}{c^2}\right)^{-\frac{1}{2}} & \text{if } |y - g(x)| > L \end{cases}$$

where $W_{max} = \sigma_{\text{noise}}/\sqrt{2}$.

**Lemma G.1** (Linear Scaling of P-IMQ's $C_1$). *Let $\Delta(x) = |g(\boldsymbol{x}) - \mu_{uc}(x)|$ be the deviation between the P-IMQ center and the uncorrupted posterior mean. The constant $C_1$ is bounded by:*

$$C_1 \leq W_{max}\left(\sqrt{L^2 + c^2} + \sup_x \Delta(x) + C_m\right)$$

*where $C_m = \frac{4\sigma_{noise}^2}{3\sqrt{3}c}$ is a constant independent of $L$ and $\Delta(x)$. Consequently, $C_1 = \mathcal{O}(L + \sup_x \Delta(x))$.*

*Proof.* We analyze the function $I(y) = |w(y)(y - m_w(y) - \mu_{uc})|$. (We omit the dependence on $x$ where clear for brevity). Let $r = |y - g(\boldsymbol{x})|$ be the residual magnitude and $s = \text{sign}(y - g(\boldsymbol{x}))$.

First, we calculate the gradient correction term $m_w(y)$. We use the property that $m_w(y) = 2\sigma_{noise}^2 \frac{1}{w(y)} \nabla_y w(y)$.

**Case 1:** $r \leq L$ **(Inside the plateau).** $w(y) = W_{max}$ (constant). Thus, $\nabla_y w(y) = 0$, and $m_w(y) = 0$.

**Case 2:** $r > L$ **(Outside the plateau).** We calculate the gradient $\nabla_y w(y)$. Note that $\nabla_y r = s$.

$$\nabla_y w = \frac{dw}{dr}\nabla_y r$$
$$= W_{max}\left(-\frac{1}{2}\right)\left(1 + \frac{(r-L)^2}{c^2}\right)^{-\frac{3}{2}}\left(\frac{2(r-L)}{c^2}\right)s$$
$$= -W_{max}\frac{(r-L)s}{c^2}\left(1 + \frac{(r-L)^2}{c^2}\right)^{-\frac{3}{2}}.$$

Now we calculate $m_w(y)$:

$$m_w(y) = 2\sigma_{noise}^2 \frac{1}{w(y)}\nabla_y w(y)$$
$$= 2\sigma_{noise}^2 \frac{(1 + \frac{(r-L)^2}{c^2})^{\frac{1}{2}}}{W_{max}} \cdot \left(-W_{max}\frac{(r-L)s}{c^2}\left(1 + \frac{(r-L)^2}{c^2}\right)^{-\frac{3}{2}}\right)$$
$$= -2\sigma_{noise}^2 \frac{(r-L)s}{c^2}\left(1 + \frac{(r-L)^2}{c^2}\right)^{-1}.$$

Now we analyze the influence function $I(y)$. We decompose the term inside the parenthesis to separate the residual from the center bias:

$$y - m_w - \mu_{uc} = (y - g(\boldsymbol{x})) + (g(\boldsymbol{x}) - \mu_{uc}) - m_w(y)$$
$$= rs + (g(\boldsymbol{x}) - \mu_{uc}) - m_w(y).$$

We bound the influence function using the triangle inequality:

$$I(y) \leq \underbrace{|w(y)rs|}_{T_1} + \underbrace{|w(y)(g(\boldsymbol{x}) - \mu_{uc})|}_{T_2} + \underbrace{|w(y)m_w(y)|}_{T_3}.$$

We bound the supremum of each term separately.

**Term 1:** $T_1 = |w(y)r|$. If $r \leq L$, $T_1 = W_{max}r \leq W_{max}L$. If $r > L$. Let $z = r - L > 0$.

$$T_1(z) = W_{max}\left(1 + \frac{z^2}{c^2}\right)^{-\frac{1}{2}}(z + L).$$

To find the maximum, we calculate the derivative w.r.t $z$:

$$T_1'(z) = W_{max}\left[\left(1 + \frac{z^2}{c^2}\right)^{-\frac{1}{2}} - (z + L)\frac{z}{c^2}\left(1 + \frac{z^2}{c^2}\right)^{-\frac{3}{2}}\right].$$

Setting $T_1'(z) = 0$ implies $(1 + z^2/c^2) - (z^2 + zL)/c^2 = 0$, which simplifies to $1 - zL/c^2 = 0$. The maximum occurs at $z^* = c^2/L$. The maximum value is:

$$T_1(z^*) = W_{max} \left(1 + \frac{c^2}{L^2}\right)^{-\frac{1}{2}} \left(\frac{c^2}{L} + L\right)$$

$$= W_{max} \frac{L}{\sqrt{L^2 + c^2}} \frac{c^2 + L^2}{L} = W_{max} \sqrt{L^2 + c^2}.$$

**Term 2:** $T_2 = |w(y)(g(\boldsymbol{x}) - \mu_{uc})|$. Since $w(y) \leq W_{max}$ for all $y$, we have:

$$\sup T_2 \leq W_{max} \sup_x |g(\boldsymbol{x}) - \mu_{uc}(x)| = W_{max} \sup_x \Delta(x).$$

**Term 3:** $T_3 = |w(y)m_w(y)|$. If $r \leq L$, $T_3 = 0$. If $r > L$, substituting the expressions for $w(y)$ and $m_w(y)$ (using $z = r - L$):

$$|T_3(z)| = \left| W_{max} \left(1 + \frac{z^2}{c^2}\right)^{-\frac{1}{2}} \cdot \left(-2\sigma_{\text{noise}}^2 \frac{zs}{c^2} \left(1 + \frac{z^2}{c^2}\right)^{-1}\right) \right|$$

$$= W_{max} \frac{2\sigma_{\text{noise}}^2}{c^2} \cdot z \left(1 + \frac{z^2}{c^2}\right)^{-\frac{3}{2}}.$$

Let $h(z) = z(1 + z^2/c^2)^{-3/2}$. The derivative $h'(z) = 0$ when $1 - 2z^2/c^2 = 0$, so the maximum occurs at $z^* = c/\sqrt{2}$. The maximum value of $h(z)$ is $(c/\sqrt{2})(1 + 1/2)^{-3/2} = (c/\sqrt{2})(3/2)^{-3/2} = 2c/(3\sqrt{3})$.

$$\sup T_3 \leq W_{max} \frac{2\sigma_{\text{noise}}^2}{c^2} \frac{2c}{3\sqrt{3}} = W_{max} \frac{4\sigma_{\text{noise}}^2}{3\sqrt{3}c} = W_{max} C_m.$$

This term $C_m$ is constant with respect to $L$ and $\Delta(x)$.

**Combining the bounds:**

$$C_1 = \sup_{x,y} I(y) \leq W_{max} \left(\sqrt{L^2 + c^2} + \sup_x \Delta(x) + C_m\right).$$

This confirms that $C_1$ scales linearly with $L$ and $\sup_x \Delta(x)$, i.e., $C_1 = \mathcal{O}(L + \sup_x \Delta(x))$. $\qquad\square$

*Remark* G.2 (Implications for RCGP-UCB bounds). Since $C_w = \frac{\sqrt{2}C_1}{\sigma_{\text{noise}}^2}$, it inherits the same asymptotic behavior as $C_1$.

For FC-RCGP-UCB (and $M_A$ in A2-RCGP-UCB), $g(\boldsymbol{x}) = 0$. $\Delta(x) = |\mu_{uc}(x)|$, which we prove in Lemma D.1 to be $\mathcal{O}(log(T^2))$. The width $L_T$ is logarithmic in T ($\mathcal{O}(log(T))$). Thus $C_{w,T} = \mathcal{O}(log(T^2)) = \tilde{\mathcal{O}}(1)$.

For $M_R$ in A2-RCGP-UCB, both the width $L_{R,T}$ and the maximum center deviation $\sup_x \Delta_R(x)$ scale as $\mathcal{O}(\sqrt{T_c})$ (see proof of Lemma D.13). Thus $C_{w,R,T} = \mathcal{O}(\sqrt{T_c})$.

# H. Ablation Studies

All ablation experiments are run on the Forrester benchmark with the same adversarial corruption setup used in the Section 5.3. The adversary applies corruption such that points close to the optimum observe a value of $-v$ while points far from the optimum observe a value of $v$. Each configuration uses 10 random seeds and reports cumulative regret as mean $\pm$ standard deviation after the final iteration. Botorch's default input standardisation is applied throughout. It is also worth noting that the weighting function is applied to observations after Botorch's default input standardisation. We have observed similar trends when running the same ablation experiments without input standardisation, but the standard deviation of the cumulative regret was higher. Input standarisation helped improve the numerical stability across all runs which helped all seeded runs to converge. Unless stated otherwise, final simple regret reaches 0 for every RCGP-based configuration: all RCGP variants successfully locate the optimum under the given settings but the efficiency at which they achieve this varies.

## H.1. Plateau Width

We sweep the plateau half-width $L$ for FC-RCGP-UCB under both magnitude-20 corruption and uncorrupted observations, and the pair $(L_R, L_A)$ for A2-RCGP-UCB under magnitude-20 corruption. All other components are fixed at the main-paper defaults. The uncorrupted runs use 55 iterations rather than 105, as the optimum is recovered faster without corruption.

*Table 2.* FC-RCGP-UCB cumulative regret as a function of plateau half-width $L$.

| $L$ | Corrupted ($v = 20$, 105 iter.) | Uncorrupted (55 iter.) |
|---|---|---|
| 0.00 (IMQ) | $499.64 \pm 156.60$ | $137.28 \pm 42.10$ |
| 0.50 | $373.75 \pm 95.12$ | $126.47 \pm 34.44$ |
| 0.75 | $322.56 \pm 126.07$ | $128.13 \pm 37.72$ |
| 1.00 | $\mathbf{257.39 \pm 146.99}$ | $130.90 \pm 39.08$ |
| 1.50 | $330.55 \pm 141.06$ | $138.28 \pm 52.22$ |
| 2.00 | $419.74 \pm 345.38$ | $\mathbf{116.65 \pm 33.66}$ |

*Table 3.* A2-RCGP-UCB cumulative regret on a grid of plateau half-widths, corruption magnitude 20, 105 iterations.

| $L_R \setminus L_A$ | 0.5 | 1.0 | 1.5 |
|---|---|---|---|
| 0.5 | $196.84 \pm 66.30$ | $227.14 \pm 47.44$ | $250.11 \pm 77.95$ |
| 1.0 | $269.36 \pm 228.55$ | $287.22 \pm 190.38$ | $261.78 \pm 74.89$ |
| 1.5 | $\mathbf{185.84 \pm 26.01}$ | $313.53 \pm 211.88$ | $273.00 \pm 86.95$ |

A plateau half width of $L = 0$ corresponds to the original IMQ weight function without a plateau.

Under corruption, cumulative regret is U-shaped in $L$, with a minimum at $L = 1$ and higher regret at both the tightest ($L = 0$) and widest ($L = 2$) settings. This is consistent with the intended mechanism: a too-tight plateau down-weights legitimate observations, while a too-wide plateau lets large corruptions pass through unweighted. As a reminder, all the variants rached the optimum eventually, the plateau half-width effect is on the efficiency of convergence.

Without corruption the cumulative regret varies little across the sweep and the widest tested value is nominally lowest, consistent with there being no outliers to penalise.

For A2-RCGP-UCB, the configuration $(L_R, L_A) = (1.5, 0.5)$ minimises both the mean and the standard deviation.

## H.2. Weight Function

We replace the IMQ tail of P-IMQ with three alternatives that differ only in decay rate outside the plateau: Cauchy (slow), Matérn-$3/2$ (fast), and RBF (Gaussian, fastest). Plateau widths and all other components are unchanged. All experiments use 105 iterations and corruption magnitude 20 in the corrupted setting.

*Table 4.* Weight-function ablation, cumulative regret. Tail decay speed increases left-to-right.

| Algorithm | Setting | P-IMQ | P-Cauchy | P-Matérn-$3/2$ | P-RBF |
|---|---|---|---|---|---|
| FC-RCGP | Uncorrupted | $\mathbf{146.17 \pm 35.27}$ | $227.89 \pm 14.01$ | $280.28 \pm 18.76$ | $291.30 \pm 11.78$ |
| FC-RCGP | Corrupted | $322.56 \pm 126.07$ | $378.94 \pm 157.71$ | $\mathbf{299.09 \pm 137.77}$ | $318.14 \pm 195.58$ |
| A2-RCGP | Uncorrupted | $\mathbf{134.73 \pm 42.16}$ | $139.78 \pm 52.79$ | $141.83 \pm 53.95$ | $145.11 \pm 52.06$ |
| A2-RCGP | Corrupted | $273.00 \pm 86.95$ | $\mathbf{263.40 \pm 118.01}$ | $266.20 \pm 135.07$ | $287.54 \pm 129.89$ |

In uncorrupted settings the gentlest tail (IMQ) gives the lowest cumulative regret and the fastest-decaying tail (RBF) the highest, for both algorithms; for FC-RCGP the standard-deviation bands of IMQ, Cauchy, and the faster-decaying tails do not overlap, though Matérn and RBF are themselves indistinguishable. Under corruption all four weights for both algorithms fall within one standard deviation of each other and cannot be separated. However, it is likely that the input standarisation is partly responsible for the flattened weights performance curves.

### H.3. Dependence on $T_c$

We compare two ways of setting the robust confidence multiplier $\sqrt{\beta_t}$. The version matching our regret analysis uses the full $\sqrt{\beta_t} = \sqrt{\beta_t'} + C_w\sqrt{T_c}$, with $T_c$ estimated online as in our implementation. The simpler alternative sets $T_c = 0$, recovering $\sqrt{\beta_t} = \sqrt{\beta_t'}$ and removing any dependence on knowing the corruption budget. The corruption magnitude is 20 and runs are 105 iterations. We run the comparison both with BoTorch's input standardisation, enabled by default (Table 5), and with it disabled (Table 6).

*Table 5.* Acquisition-function ablation under input standardisation, cumulative regret. Final simple regret is at the numerical floor for both variants on every seed.

| Algorithm | $T_c = 0$ | $T_c$ estimated (default) |
|---|---|---|
| FC-RCGP | $\mathbf{257.39 \pm 146.99}$ | $509.19 \pm 179.31$ |
| A2-RCGP | $\mathbf{185.84 \pm 26.01}$ | $251.41 \pm 46.90$ |

*Table 6.* Acquisition-function ablation with input standardisation disabled, cumulative regret (mean over seeds). The two variants are near-identical.

| Algorithm | $T_c$ estimated (default) | $T_c = 0$ |
|---|---|---|
| FC-RCGP | 431 | 438 |
| A2-RCGP | 384 | 384 |

With standardisation enabled, dropping the inflation term ($T_c = 0$) attains markedly lower cumulative regret than the analysed, budget-aware default, while still locating the optimum on every seed; with standardisation disabled the two are near-identical (Table 6).

A possible explanation as to why $T_c = 0$ performs better in practice is that the RCGP models are already conservative in how they incorporate observations. Inflating the confidence intervals even more adds an unnecessary level of caution reducing observations' data efficiency.

Two points follow. First, that removing the $C_w\sqrt{T_c}$ inflation *improves* performance without ever losing the optimum indicates that the confidence intervals underlying our analysis are loose; a tighter proof technique could plausibly shrink them and bring the regret bounds closer to the behaviour observed here. Second, setting $T_c = 0$ eliminates any need to know the adversary's budget $T_c$ in advance—a quantity typically unavailable in practice—yet performs at least as well as the budget-aware default. This robustness to an unknown budget is what leads us to recommend using $T_c = 0$ in our other experiments.

### H.4. Corruption Magnitude

We vary the magnitude $v$ of the injected corruptions over $\{10, 20, 30\}$, with all other parameters—including the corruption budget $T_c$—fixed at the same settings as in Section 5.3. A level-$v$ adversary returns $-v$ for queries near the optimum and $+v$ for queries far from it. Runs are 105 iterations.

*Table 7.* Cumulative regret as a function of corruption magnitude $v$. Best per row in bold.

| $v$ | A2-RCGP | FC-RCGP | DiagnosticGP | Student-t | GP |
|---|---|---|---|---|---|
| 10 | $\mathbf{184.03 \pm 82.40}$ | $240.05 \pm 74.26$ | $471.60 \pm 329.40$ | $554.81 \pm 447.56$ | $583.78 \pm 478.46$ |
| 20 | $\mathbf{185.84 \pm 26.01}$ | $257.39 \pm 146.99$ | $449.13 \pm 84.03$ | $727.45 \pm 512.58$ | $730.31 \pm 403.22$ |
| 30 | $\mathbf{220.25 \pm 20.70}$ | $309.74 \pm 230.92$ | $367.51 \pm 76.77$ | $924.24 \pm 461.48$ | $956.02 \pm 477.81$ |

A2-RCGP-UCB has the lowest cumulative regret at every corruption magnitude, with FC-RCGP-UCB second. The cumulative regret of both RCGP-based methods changes little as the magnitude increases. It is worth nothing that the lower corruption value $v = 10$ corresponds to within-plateau attacks in this experiments' setting. This indicate that the algorithms are robust both to small and large corruptions, albeit the robustness to within-plateau attacks stems from the assumption on the adversary's budget. The two GP-based baselines instead grow with magnitude. Student-t-GP fails to locate the optimum

from $v = 20$ onward (final simple regret $1.48$), and GP-UCB's final simple regret rises to $0.22$ at $v = 30$, whereas the RCGP-based methods and DiagnosticGP simple regrets remain $0$.

## I. Brittleness of Standard GPs

We provide an example to show that a single unbounded corrutpion can derail the entire BO process over the entirety of the $T$ steps.

We assume a dataset $\mathcal{D}_{t-1}$ of clean observations. At step $t$, the adversary corrupts the observation at $\boldsymbol{x}_\mathrm{c}$ such that $y_c = f(\boldsymbol{x}_\mathrm{c}) + \epsilon + c$. We analyze the posterior mean of the standard GP, denoted $\mu_t^{GP}(\boldsymbol{x})$.

Using the standard block matrix inversion lemma (equivalent to Lemma 2 in our submission with $J_w = I$), we can express the standard GP posterior mean $\mu_t^{GP}(\boldsymbol{x})$ as a function of the "clean" posterior mean $\mu_{uc,t}(\boldsymbol{x})$ (conditioned only on $\mathcal{D}_{t-1}$) plus a correction term induced by the new observation $y_c$.

$$\mu_t^{GP}(\boldsymbol{x}) = \mu_{uc,t}(\boldsymbol{x}) + \frac{\mathrm{cov}_{\mathcal{D}_{uc}}(\boldsymbol{x}, \boldsymbol{x}_\mathrm{c})}{\sigma_{uc,t}^2(\boldsymbol{x}_\mathrm{c}) + \sigma_{noise}^2} \left(y_c - \mu_{uc,t}(\boldsymbol{x}_\mathrm{c})\right)$$

Where: $\mu_{uc,t}(\boldsymbol{x})$ and $\sigma_{uc,t}^2(\boldsymbol{x})$ are the posterior mean and variance conditioned on the clean history $\mathcal{D}_{t-1}$. $\mathrm{cov}_{\mathcal{D}_{uc}}(\boldsymbol{x}, \boldsymbol{x}_\mathrm{c})$ is the posterior covariance between the query point $\boldsymbol{x}$ and the corrupted location $\boldsymbol{x}_\mathrm{c}$ given $\mathcal{D}_{t-1}$. $y_c - \mu_{uc,t}(\boldsymbol{x}_\mathrm{c})$ is the predictive residual. In this expression, we have used that the Schur complement $S = \sigma_{uc,t}^2(\boldsymbol{x}_\mathrm{c}) + \sigma_{noise}^2$ is a scalar and easily invertible.

Let us define $W_t(\boldsymbol{x}, \boldsymbol{x}_\mathrm{c})$ as:

$$W_t(\boldsymbol{x}, \boldsymbol{x}_\mathrm{c}) \triangleq \frac{\mathrm{cov}_{\mathcal{D}_{uc}}(\boldsymbol{x}, \boldsymbol{x}_\mathrm{c})}{\sigma_{uc,t}^2(\boldsymbol{x}_\mathrm{c}) + \sigma_{noise}^2}$$

Crucially, for a standard GP, $W_t$ depends only on the input locations $\{\boldsymbol{x}_1, \ldots, \boldsymbol{x}_\mathrm{c}\}$ and the kernel hyperparameters. It is independent of the observation values $y$. Substituting $y_c = f(\boldsymbol{x}_\mathrm{c}) + \epsilon + c$:

$$\mu_t^{GP}(\boldsymbol{x}) = \underbrace{\mu_{uc,t}(\boldsymbol{x}) + W_t(\boldsymbol{x}, \boldsymbol{x}_\mathrm{c})(f(\boldsymbol{x}_\mathrm{c}) + \epsilon - \mu_{uc,t}(\boldsymbol{x}_\mathrm{c}))}_{\text{Term independent of } c} + \underbrace{W_t(\boldsymbol{x}, \boldsymbol{x}_\mathrm{c}) \cdot c}_{\text{Linear corruption term}}$$

To incur linear regret, the adversary aims to force the algorithm to select a suboptimal point $\boldsymbol{x}_\mathrm{bad}$ (where $f(\boldsymbol{x}_\mathrm{bad}) < f(\boldsymbol{x}^*) - \Delta$) over the true optimum $\boldsymbol{x}^*$. This occurs if the Upper Confidence Bound (UCB) of $\boldsymbol{x}_\mathrm{bad}$ exceeds the UCB of $\boldsymbol{x}^*$. Ideally, the adversary wants to raise $\mu_t^{GP}(\boldsymbol{x}_\mathrm{bad})$ arbitrarily high. The condition to force $\mu_t^{GP}(\boldsymbol{x}_\mathrm{bad})$ to exceed any arbitrary bound $B$ (e.g., $B = \max_{\boldsymbol{x}} f(\boldsymbol{x}) + \mathrm{Confidence}(\boldsymbol{x}^*)$) is:

$$\mu_t^{GP}(\boldsymbol{x}_\mathrm{bad}) = \mu_{uc,t}(\boldsymbol{x}_\mathrm{bad}) + W_t(\boldsymbol{x}_\mathrm{bad}, \boldsymbol{x}_\mathrm{c})(f(\boldsymbol{x}_\mathrm{c}) + \epsilon - \mu_{uc,t}(\boldsymbol{x}_\mathrm{c})) + W_t(\boldsymbol{x}_\mathrm{bad}, \boldsymbol{x}_\mathrm{c}) \cdot c > B$$

Solving for $c$:

$$c > \frac{B - \mu_{uc,t}(\boldsymbol{x}_\mathrm{bad})}{W_t(\boldsymbol{x}_\mathrm{bad}, \boldsymbol{x}_\mathrm{c})} - (f(\boldsymbol{x}_\mathrm{c}) + \epsilon - \mu_{uc,t}(\boldsymbol{x}_\mathrm{c}))$$

It is possible to argue that potentially $W_t(\boldsymbol{x}_\mathrm{bad}, \boldsymbol{x}_\mathrm{c}) \to 0$ due to the "screening" effect of data or variance reduction. However, even if $W_t \approx O(T^{-k})$, the adversary can set $c \approx O(T^{k+1})$ to successfully corrupt the GP.

## J. Practical Considerations and Experimental Details

This section outlines the practical implementation details of our proposed algorithms and the configuration of our empirical evaluation.

### J.1. Practical Considerations

Several practical considerations are crucial for the stable and effective deployment of the RCGP-UCB algorithms.

**Data Standardization**    Our implementation leverages the MIT-licenced BoTorch (Balandat et al., 2020) and GPytorch (Gardner et al., 2021) Python libraries, which, by default, standardise the input data to have zero mean and unit variance. We observed that this practice significantly enhances numerical stability. Furthermore, this normalization mitigates the risk of severe prior mean misspecification. As a consequence, with the data centered around zero, the performance of the Fixed-Center (FC-RCGP) algorithm tends to be empirically similar to that of the Anchor-Adapt (A2-RCGP) algorithm in our experiments, as the fixed zero-mean center becomes a more reasonable assumption.

**P-IMQ Hyperparameter Selection**    The data normalization also facilitates the selection of the P-IMQ hyperparameters: the plateau half-width $L$ and the shape parameter $c$. With observations scaled to a standard range, one can use reliable manual estimates, such as $c = 1$ and $L = 1.96$ (corresponding to the 95% confidence interval of a standard normal distribution). Alternatively, we found that a data-driven heuristic approach performed well: setting $c = 1$ and defining $L$ as the 95% quantile of the absolute deviation of the observed data from the median.

**Proxy for the Uncorrupted Posterior Variance $\sigma_{\mathbf{uc}}(\boldsymbol{x})$**    The theoretical acquisition function relies on the posterior standard deviation of an idealized GP conditioned only on uncorrupted data, $\sigma_{uc}(x)$, which is inaccessible in practice. A valid approach involves attempting to identify and prune outliers before computing the variance. This can be done by filtering away observations outside of the P-IMQ plateau region. However, we opt to use the RCGP posterior standard deviation, $\sigma^R(\boldsymbol{x})$, directly as a proxy. This choice is justified because the RCGP model is inherently more cautious than a standard GP, and its posterior variance is guaranteed to be at least as large as that of a GP fit on a subset of the data. Using $\sigma^R(x)$ avoids introducing an additional, potentially aggressive, pruning step that could erroneously discard valid data points and excessively increase the algorithm's exploratory caution.

**Estimation of the Corruption Count $T_c$**    The robust confidence multiplier, $\sqrt{\beta_t}$, depends on the total number of corruptions, $T_c$. Since $T_c$ is typically unknown, it must be estimated. In our experiments, we estimate $T_c$ at each step as the number of observations whose residuals fall outside the plateau of the P-IMQ weighting function. Other thresholds based on the weight values could also be employed. Interestingly, we also observed strong empirical performance when simply setting $T_c = 0$ in the computation of $\beta_t$. This can be attributed to the fact that the RCGP model itself provides a baseline level of cautiousness, leading to slightly more exploration than standard GP-UCB even without inflating the confidence bounds. In practice, the user can choose between an adaptive estimate of $T_c$ and setting $T_c = 0$, depending on their desired level of exploration and prior beliefs about the corruption frequency.

### J.2. System Specifications

All experiments were conducted on a Docker container running within Windows Subsystem for Linux 2 (WSL2). The system specifications are as follows:

**Hardware Configuration:**

- **CPU:** Intel Core i9-13980HX (13th Generation), 16 physical cores, 32 logical processors

- **Memory:** 32 GB RAM

- **GPU:** NVIDIA GeForce RTX 4090 with 16 GB VRAM

- **Storage:** 1 TB available space

**Software Environment:**

- **Operating System:** Ubuntu 24.04.2 LTS (Noble Numbat)

- **Container Platform:** Docker running on WSL2

- **Python Version:** 3.12.3

- **CUDA Version:** 12.9

- **NVIDIA Driver:** 576.52

- **Kernel:** Linux 6.6.87.2-microsoft-standard-WSL2

**Container Details:**

- **Architecture:** x86_64

- **Memory Allocation:** Full access to 32 GB RAM

- **GPU Access:** Full access to RTX 4090 with CUDA support

The provided code comes with a config file allowing for easy setup of the docker container and dependency management using the uv Python's dependency manager.

### J.3. Running the experiments

To reproduce the experiments, use the command `uv run python <filename>`. Where `<filename>` depends on the experiment:

- **Forrester experiment:** `experiments/synthetic/forrester_adversarial.py`
  - This experiment is quite fast and should run within 40 minutes for 20 random seeds.

- **CIFAR hyperparameters optimization:** `experiments/hpt_cifar/test_hpt_cifar_lr_wd.py`
  - This script will take around 8 hours to execute.

- **Lunar lander:** `experiments/lunar_lander/lunar_lander.py`
  - This experiment will take between 8 to 12 hours to run.

Due to the cubic time complexity of matrix inversion ,which happens at every BO step, the main driver of the time needed to reach is script is the number of BO iterations. The number of seeds only drives the time spent linearly.

### J.4. Experiments' Details

To ensure clarity and reproducibility, our experimental configurations are defined by a set of common parameters and experiment-specific settings. This section first outlines the parameters shared across all experiments and then details the unique configurations for the Forrester, hyperparameters optimization on CIFAR-10, and Lunar Lander tasks. All configurations are derived from the scripts located in the experiment directory.

#### J.4.1. COMMON CONFIGURATION PARAMETERS

The following parameters are common to all the experiments' scripts.

- **N_ITERATIONS**: The number of Bayesian optimization steps performed. This number excludes the initial uncorrupted data points.

- **N_INITIAL**: The number of initial points evaluated to seed the surrogate model before the optimization loop begins. These points are chosen via a quasi-random sequence and are not subject to corruption.

- **SEED**: The random seed used to initialize the experiment, ensuring the reproducibility of initial points and other stochastic processes. When **N_SEEDS**, SEED is the base seed, and the actual seed used for each experiment is SEED + i where $i$ is the index of the experiment.

- **STANDARDIZE**: A boolean flag that, when set to True, standardizes the observed outcomes (Y values) to have a zero mean and unit variance. This practice is the default behavior for GPyTorch and BoTorch models. We noticed that outcome standarisation greatly helped with numerical stability.

- **FIT_HYPERPARAMETERS**: A boolean flag to enable or disable the fitting of surrogate model hyperparameters (e.g., likelihood noise and kernel's lengthscale and outputscale) during the optimization loop.

### J.4.2. FORRESTER FUNCTION

This experiment evaluates the algorithms on the 1D Forrester function under a targeted adversarial attack.

The script is located at `experiments/synthetic/forrester_adversarial.py`.

- **Key Configuration**:
    - `N_ITERATIONS`: 30
    - `N_INITIAL`: 5
    - `USE_ROBUST_HEURISTICS`: `True`. Enables data-driven heuristics for setting the P-IMQ 'plateau_width' and shape parameter 'c'.

- **Adversarial Model**: An `AdversarialCorruptor` is used with a fixed budget. It attempts to mislead the optimizer by reporting false values based on the query's proximity to the true optimum.
    - `CORRUPTION_TYPE`: `'time_budget'`. Define the frequency budget of the adversary as $T^\alpha$ where $\alpha$ is given by `TIME_BUDGET_ALPHA`.
    - `near_threshold`: 0.1. Queries within this distance of the optimum are corrupted.
    - `far_threshold`: 0.4. Queries beyond this distance from the optimum are corrupted.
    - `high_value`: 25.0. The false value reported for points far from the optimum.
    - `low_value`: -10.0. The false value reported for points near the optimum.

### J.4.3. CIFAR-10 HYPERPARAMETER OPTIMIZATION

This experiment involves a 2D hyperparameter optimization task for a ResNet classifier on the CIFAR-10 dataset, focusing on learning rate and weight decay.

The script is located at `experiments/hpt_cifar/hpt_cifar_lr_wd.py`.

- **Objective and Search Space**: The goal is to maximize validation accuracy by tuning the learning rate (in $[10^{-5}, 10^{-1}]$) and weight decay (in $[10^{-6}, 10^{-2}]$) on a log scale.

- **Key Configuration**:
    - `N_ITERATIONS`: 140
    - `N_INITIAL`: 10
    - `MAX_EPOCHS`: 4. Each hyperparameter evaluation involves training the network for 4 epochs.

- **Corruption Model**: Simulates training failures by reporting a low objective value.
    - `CORRUPTION_TYPE`: `"time_budget"`. The frequency of corruption is proportional to $T^{1/3}$, where $T$ is the number of iterations.
    - `TIME_BUDGET_ALPHA`: 1/3. The exponent for the time-based corruption budget.
    - `CRASH_VALUE`: -2.0. The fixed low accuracy value reported when a simulated failure occurs.

### J.4.4. LUNAR LANDER POLICY OPTIMIZATION

This experiment tackles a high-dimensional (36D) policy optimization task for the `LunarLander-v3` environment. The script is located at `experiments/lunar_lander/lunar_lander.py`.

- **Objective and Search Space**: The objective is to maximize cumulative reward by optimizing a 36-parameter linear policy. Each parameter is bounded within $[-2.0, 2.0]$.

- **Key Configuration**:
    - `N_ITERATIONS`: 300
    - `N_INITIAL`: 10

- N_EPISODES: 4. Each policy evaluation is the average reward over 4 episodes to reduce noise in the objective function.

- **Corruption Model**: An adversary introduces spuriously low reward values to create deceptive optima and mislead the optimizer.

  - CORRUPTION_TYPE: 'time_budget'. The corruption frequency scales with $T^{1/3}$.
  - TIME_BUDGET_ALPHA: 1/3. The exponent for the time-based corruption budget.
  - CORRUPTION_VALUE: -1000.0. The deceptive, low reward value reported during a corruption event.

