# OpenReview forum: "Robust Bayesian Optimisation with Unbounded Corruptions"
_ICML.cc/2026/Conference — ICML 2026 regular_

### Official Review · Reviewer_fhHM · 2026-02-21

**Soundness:** 3
**Presentation:** 3
**Significance:** 3
**Originality:** 3
**Overall Recommendation:** 4
**Confidence:** 4

**Summary:**

This paper studies robust Bayesian optimisation in the presence of extreme outliers. The authors argue that existing theoretically grounded robust Bayesian optimisation methods typically assume a bounded corruption magnitude, which makes them fragile when rare but catastrophic failures occur. The work introduces a new adversarial model where corruptions are limited in frequency rather than magnitude.

The paper proposes a Bayesian optimisation algorithm that combines a robust conjugate Gaussian process model with an upper confidence bound acquisition rule. Two variants of the algorithm are presented, one using a fixed reference model and one using an adaptive reference model. Theoretical analysis shows sublinear regret guarantees under this new corruption model while matching the performance of standard Gaussian process optimisation in the absence of outliers. Experiments on synthetic benchmarks, hyperparameter tuning, and reinforcement learning demonstrate improved robustness compared to standard baselines.

**Compliance With Llm Reviewing Policy:**

Affirmed.

**Key Questions For Authors:**

1. The algorithm assumes knowledge or estimation of the corruption frequency. How sensitive is performance when this estimate is inaccurate?

2. Can the authors clarify how the proposed approach compares computationally with existing robust Bayesian optimisation methods in large-scale settings?

3. The adaptive variant appears empirically strong but theoretically weaker. Do the authors believe this gap is fundamental or an artifact of the analysis?

4. How sensitive are results to the choice of weighting function used in the robust Gaussian process model?

5. Could the authors elaborate on practical scenarios where infinite-magnitude corruptions realistically occur and how the proposed model captures them?

**Limitations:**

No. The impact statement is minimal and does not meaningfully discuss limitations or broader societal considerations. The authors could expand this section to clarify deployment risks and practical constraints.

**Strengths And Weaknesses:**

Soundness
Strengths
1. The problem formulation is clear and well motivated, especially the shift from magnitude-constrained corruption to frequency-constrained corruption.
2. The methodological development builds logically on recent work in generalized Bayesian inference and robust Gaussian processes.
3. Theoretical guarantees are provided and appear consistent with the stated assumptions.
4. Experimental results generally support the claim that the proposed method maintains performance when no corruptions are present while improving robustness under adversarial conditions.
Weaknesses
1. The theoretical guarantees rely on several strong assumptions, including prior knowledge or accurate estimation of the number of corrupted observations and bounds on the objective function.
2. Some parts of the analysis introduce conservative bounds that may not reflect realistic performance, which the authors themselves acknowledge.
3. The robustness mechanism is heavily tied to a specific weighting scheme, and the paper does not clearly discuss how sensitive performance is to this design choice.
Overall, the work appears technically solid within the scope of its assumptions.

Presentation
Strengths
1. The paper is generally well structured, progressing from motivation to methodology, analysis, and experiments.
2. The conceptual narrative around “zero-cost robustness” is clear and easy to follow.
3. The experiments are described in sufficient detail to understand the setup.
Weaknesses
1. The exposition becomes dense in the methodological sections, especially when discussing weighting functions and confidence bounds.
2. Some intuition behind the adaptive variant of the algorithm could be introduced earlier to help readers follow the technical development.
3. The broader positioning within machine learning could be improved, since parts of the paper read more like a statistical inference contribution than a general ML algorithmic advance.

Significance
Strengths
1. Robust optimisation under extreme outliers is an important problem in real-world applications such as robotics, hyperparameter tuning, and automated experimentation.
2. The idea of robustness without sacrificing performance in the clean-data setting is conceptually appealing.
3. The proposed adversarial model broadens how robustness can be studied in Bayesian optimisation.
Weaknesses
1. The contribution is somewhat specialized within the Bayesian optimisation community and may not strongly influence broader areas of machine learning.
2. The practical scalability to high-dimensional settings remains unclear, as some experiments suggest slow convergence.
Overall significance is moderate.

Originality
Strengths
1. Introducing a corruption model that limits frequency rather than magnitude is a meaningful conceptual shift.
2. The integration of robust conjugate Gaussian processes into a theoretically justified optimisation framework is novel.
3. The notion of achieving robustness without degrading performance in the clean setting is a compelling design principle.
Weaknesses
1. Many building blocks originate from existing literature, and the main novelty lies in combining and analysing them rather than introducing entirely new algorithmic mechanisms.

---

> ### Author Rebuttal · Authors · 2026-03-30
>
> We thank the reviewer for their constructive feedback. We address the questions first, then comment on the weaknesses.
>
> ## Questions
>
> ### Q1: Sensitivity to $T_ c$ estimation
>
> On the corrupted Forrester experiment (100 steps):
>
> | Method | Estimated $T_ c$ | $T_ c = 0$ |
> |--------|-----------------|-----------|
> | RCGP | 431 | 438 |
> | A2-RCGP | 384 | 384 |
>
> The algorithms are insensitive to $T_ c$: setting $T_ c = 0$ works just as well. Robustness comes from the RCGP model's weighting function, not the inflated confidence multiplier. The dependence on $T_ c$ in our bounds is an artifact of the proof technique.
>
> ### Q2: Computational comparison
>
> **FC-RCGP-UCB** has identical cost to GP-UCB: $O(t^3)$ per step with the same rank-1 update optimizations available.
>
> More broadly, because RCGP computations mirror standard GP algebra (differing only through the diagonal matrix $J_ w$), speed-ups commonly used with GPs, such as sparse variational GPs (SVGPs), are directly applicable.
>
> **A2-RCGP-UCB** maintains two RCGP models, so it is roughly twice as expensive. However, since its center is adaptive, it requires fitting every step and doesn't benefit from rank-1 update optimizations as standard GP and RCGP. In practical BO, hyperparameters are re-estimated at every step anyway, so the overhead is not prohibitive. A2-RCGP is designed for the sparse-data regime where prior misspecification matters most; once hyperparameters estimates stabilize, switching to FC-RCGP is recommended.
>
>
> ### Q3: Theory-practice gap for the adaptive variant
>
> This gap is partly an artifact of the analysis and partly a genuine price for adaptivity. On the analysis side, the cascading use of $\Psi(T_ c)$ and the requirement that uncorrupted observations fall within the constructed plateau inflate the constants — this is needed for our current proof technique but may not be fundamental. On the other hand, there is a real theoretical cost: the adaptive centering $g_ R = \mu_ {A,t-1}$ can be manipulated by the adversary, making A2-RCGP inherently more susceptible to corruption than the fixed-center variant. FC-RCGP is more stable theoretically, though it loses efficiency when its zero center is misspecified. We have run an ablation study with different corruption levels and A2RCGP had higher variance in its cumulative regret, this suggests that the benefits of flexibility come at the price of stability; a price paid in the allowed adversary's corruption budget.
>
> ### Q4: Sensitivity to the weighting function
>
> In theory, any weighting function satisfying RCGP's conditions (differentiable, bounded, with $\sup_ y |y| \cdot w^2 < \infty$) should work [1]. This includes Gaussian and Matérn kernels. We use IMQ because it has been a standard choice in robust Generalised Bayesian inference (Matsubara et al. [1]; Altamirano et al. [2]). We refer to our response to Reviewer MXef for further discussion. The intuition for choosing a weighting function: the more confident you are about your problem setup, the more aggressive the weight decay can be; if uncertain, a more permissive scheme is safer.
>
> ### Q5: Practical scenarios for infinite-magnitude corruptions
>
> Our corruptions need not be literally infinite — just large enough that assuming a finite bound $C$ on $\sum_ t |c_ t|$ becomes impractical. None of our experiments used infinite values. Practical examples include: crashed simulations mapped to arbitrary penalty values (our CIFAR and Lunar Lander experiments), or catastrophic instabilities in plasma control for nuclear fusion, where disruption events produce sensor readings orders of magnitude outside the normal range.
>
> ## Weaknesses
>
> ### Strong assumptions
>
> We acknowledge the assumptions are strong, but note that the existing literature assumes a finite **cumulative** corruption budget $\sum_ t |c_ t| \leq C$ with $C$ known — which is even more restrictive, as a handful of corruptions exhaust it entirely. Our model only bounds the frequency, making the adversary strictly more flexible. While our theory assumes knowledge of $T_ c$, it is irrelevant in practice (see Q1). We also ran plateau width ablations (see response to Reviewer MXef) showing that a conservative (tighter) plateau is safer to use: it converges, just with less efficiency.
>
> ### Conservative bounds
>
> Agreed. The bounds derived are a direct result of the proof technique used. Tighter bounds could potentially be achieved if one can manage to bound the RCGP bias term $m_ w$ without the need of the plateau condition.
>
>
> ### Exposition of A2-RCGP-UCB
>
> We will revise the manuscript to improve clarity of the dual-model structure.
>
> [1] Matsubara, Takuo, Jeremias Knoblauch, François-Xavier Briol, and Chris J. Oates. "Robust generalised Bayesian inference for intractable likelihoods." JRSSB 84(3):997–1022 (2022).
>
> [2] Altamirano, Matias, François-Xavier Briol, and Jeremias Knoblauch. "Robust and conjugate Gaussian process regression." arXiv preprint arXiv:2311.00463 (2023).

---

> > ### Author Rebuttal · Reviewer_fhHM · 2026-04-02
> >
> > Thank you for the detailed and thoughtful rebuttal. Several of my concerns have been clarified satisfactorily. In particular, the discussion on the practical relevance of frequency-constrained corruptions, the computational complexity comparison with GP-UCB, and the explanation of the adaptive variant’s theory–practice gap are helpful and strengthen the paper. The additional intuition and empirical evidence regarding the insensitivity to the corruption-frequency parameter are also appreciated. However, some of my core concerns are only partially addressed:
> >
> > 1. Dependence on the corruption frequency parameter T_c: While the rebuttal argues that performance is not sensitive to T_c in practice and that it can be estimated or conservatively set, the theoretical guarantees and algorithm design still fundamentally rely on knowing or bounding this quantity. This gap between theory and deployability remains important, especially for real-world applications where such information is typically unavailable.
> >
> > 2. Sensitivity to the weighting function: The response provides useful intuition and references supporting the choice of weighting functions, but does not include empirical or systematic analysis of how different choices affect performance. Given that robustness critically depends on this design, a more concrete evaluation or discussion would strengthen the claims.
> >
> > 3. Strength of assumptions and conservativeness of bounds: The authors acknowledge that some assumptions (e.g., plateau conditions, bounds on the function) and regret bounds are conservative and partly artifacts of the proof technique. While this is understandable for a theoretical paper, it still limits the interpretability of the guarantees in realistic settings.
> >
> > Overall, the rebuttal improves clarity and addresses several questions, but key aspects related to robustness assumptions and practical applicability would benefit from further clarification or future investigation.

---

> > > ### Author Response · Authors · 2026-04-02
> > >
> > > We thank the reviewer for their feedback. We address the remaining concerns below.
> > >
> > > ## Dependence on T_ c
> > >
> > > We emphasise that our setting is **not worse than the existing literature**. The standard corruption model [1, 2] assumes knowledge of the **cumulative corruption budget** $C = \sum |c_ t|$, requiring bounds on both the frequency *and* magnitude of corruptions. Our model only requires bounding the **count**, a strictly weaker requirement.
> > >
> > > Moreover, we demonstrated empirically that our algorithms are **robust to misspecification** of $T_ c$: both adaptive estimation (counting observations outside the plateau) and simply setting $T_ c = 0$ yielded strong performance. This insensitivity suggests the explicit dependence on $T_ c$ is an artifact of the proof technique rather than a fundamental limitation, supporting practical deployability when $T_ c$ is unknown.
> > >
> > > ## Sensitivity to the weighting function
> > >
> > > We conducted an ablation experiment on the Forrester benchmark comparing four plateau weighting functions with different decay rates: P-IMQ (polynomial, gentle tail), P-Cauchy (polynomial, moderate), P-Matérn 3/2 (exponential), and P-RBF (exponential, most aggressive). All satisfy the theoretical requirements (positivity, differentiability, Huber robustness). To isolate the effect of tail decay, all four functions use the same plateau width $L$ and shape parameter $c$. For an ablation on the plateau width itself, we refer the reviewer to the experiments provided in response to reviewer MXef. Results are averaged over 10 seeds.
> > >
> > > **Uncorrupted setting (T_ c = 0):**
> > >
> > > | Algorithm | Weight | Cumul. Regret |
> > > |---|---|---|
> > > | A2-RCGP | P-IMQ |  **165.5 ± 12.8** |
> > > | A2-RCGP | P-RBF | 176.6 ± 11.0 |
> > > | A2-RCGP | P-Cauchy | 185.1 ± 12.5 |
> > > | A2-RCGP | P-Matérn 3/2 | 189.4 ± 11.6 |
> > > | FC-RCGP | P-IMQ | **232.7 ± 38.3** |
> > > | FC-RCGP | P-Cauchy | 301.0 ± 15.9 |
> > > | FC-RCGP | P-RBF | 321.2 ± 19.8 |
> > > | FC-RCGP | P-Matérn 3/2 | 333.0 ± 14.3 |
> > >
> > > **Corrupted setting (T_ c = T^{1/3}):**
> > >
> > > | Algorithm | Weight | Cumul. Regret |
> > > |---|---|---|
> > > | FC-RCGP | P-RBF | **271.1 ± 63.2** |
> > > | FC-RCGP | P-IMQ | 393.5 ± 40.5 |
> > > | FC-RCGP | P-Cauchy | 409.2 ± 40.6 |
> > > | FC-RCGP | P-Matérn 3/2 | 414.2 ± 79.4 |
> > > | A2-RCGP | P-RBF | **495.7 ± 38.5** |
> > > | A2-RCGP | P-Matérn 3/2 | 522.7 ± 84.9 |
> > > | A2-RCGP | P-Cauchy | 525.0 ± 43.1 |
> > > | A2-RCGP | P-IMQ | 525.5 ± 40.3 |
> > >
> > > The results reveal a clear trade-off. Without corruption, gentle tails (P-IMQ) preserve statistical efficiency by retaining information from observations outside the plateau (their contribution is downweighted just enough to ensure Huber robustness but is preserved otherwise). Under corruption, the ranking reverses: aggressive tails (P-RBF) win by decisively eliminating outlier influence.
> > >
> > > **All 16 configurations achieved zero simple regret**, confirming that every weighting function finds the optimum regardless of tail choice. The differences are purely in convergence speed. A practitioner can thus select the tail profile based on expected corruption severity without risking failure.
> > >
> > >
> > > ## Strength of assumptions
> > >
> > > We acknowledge the conservativeness of some assumptions and bounds. However, the assumptions are **comparable to the existing literature**: GP-UCB (Srinivas et al., 2010) assumes bounded RKHS norm and sub-Gaussian noise; corruption-tolerant algorithms [1, 2] additionally assume a known cumulative budget. Our assumptions are of a similar nature, while facing a strictly **stronger adversary** with possibly infinite-magnitude corruptions.
> > >
> > > We agree that alternative proof techniques or acquisition functions could yield tighter bounds. The gap between our proven rates ($T_ c = O(T^{1/4})$ for FC-RCGP, $O(T^{1/7})$ for A2-RCGP) and the empirically tolerated rate ($T_ c = O(T^{1/3})$) confirms the bounds are not tight, and we view closing this gap as a promising direction for future work.
> > >
> > > [1] Bogunovic, Ilija, Andreas Krause, and Jonathan Scarlett. "Corruption-tolerant gaussian process bandit optimization." International Conference on Artificial Intelligence and Statistics. PMLR, 2020.
> > >
> > > [2] Bogunovic, Ilija, et al. "A robust phased elimination algorithm for corruption-tolerant gaussian process bandits." Advances in Neural Information Processing Systems 35 (2022): 23951-23964

---

### Official Review · Reviewer_wdvh · 2026-02-24

**Soundness:** 3
**Presentation:** 3
**Significance:** 3
**Originality:** 3
**Overall Recommendation:** 5
**Confidence:** 3

**Summary:**

The authors propose two UCB-based algorithms for Robust Bayesian Optimization that rely on Robust Conjugate Gaussian Processes. Unlike most existing robust BO literature (which assumes a bounded cumulative perturbation budget, leaving algorithms vulnerable to a single massive outlier), this work considers an adversary constrained only by the frequency of attacks. To address this frequency-constrained setting, the authors derive formal theoretical guarantees showing that their algorithms achieve sublinear regret under limited corruption budgets. An interesting theoretical property of these methods is "zero-cost robustness": in the absence of corruptions, the algorithms are guaranteed to recover the standard GP-UCB regret bounds. Finally, empirical evaluations demonstrate that the algorithms perform better in practice than their theoretical bounds suggest.

**Compliance With Llm Reviewing Policy:**

Affirmed.

**Final Justification:**

After the rebuttal, I maintain my intial positive assessment. I believe thi paper represent a novel and sound contribution to the Robust BO literature.

**Key Questions For Authors:**

The empirical evaluation currently relies on somewhat naive, 'greedy' adversaries (e.g., crashing runs as fast as the budget allows or corrupting based on a fixed distance). What would happen if the algorithms faced a more powerful, strategic 'white-box' adversary that understands how the RCGP processes points? Specifically, since the P-IMQ weight function uses a threshold ($L_T$), couldn't a smart adversary 'cheat' the system by deliberately injecting corruptions just inside the plateau boundary ($|c_t| \approx L_T$)? This seems like it could systematically drag the posterior mean away from the true function without ever triggering the outlier down-weighting mechanism. How resilient are the proposed methods to this kind of targeted data poisoning?

**Limitations:**

My only suggestion here would be that the authors emphasize earlier in the paper that their theoretical guarantees and algorithms rely on constants that are fundamentally unknown in practice.

**Strengths And Weaknesses:**

1. **Soundness.** The paper is mathematically sound, and the proofs appear correct. The theoretical assumptions are generally reasonable. However, to strengthen the work, I recommend the authors expand on the practical realism of certain theoretical requirements in the main text. For instance, the regret guarantees rely on prior knowledge of the total corruption budget $T_c$. Since this is fundamentally unknown in a true black-box setting, the authors rely on dynamic estimation heuristics in their experiments. While they do mention this estimation strategy in Appendix I, this practical workaround should be introduced much earlier in the main body of the paper. Otherwise, readers are left wondering how this critical parameter is handled in practice until they reach the experimental section. Finally, adding a brief empirical analysis showing how sensitive the algorithms are to misestimating this budget would further improve the paper's practical impact.

2. **Presentation.** The paper is well-written and clearly structured. Although the dense notation can be challenging to track at times, the inclusion of a dedicated notation appendix is helpful.

3. **Significance.** The paper addresses any important, unexplored, problem in Bayesian Optimization: robustness against catastrophic outliers. The theoretical shift from a cumulative corruption budget to a frequency-constrained budget is a novel and significant contribution to the BO literature. However, the practical significance of the proposed framework is somewhat diluted by the core premise of "unbounded" perturbations. In almost all realistic settings, physical constraints will naturally bound the maximum size of an outlier. By optimizing for infinite outliers, the proposed methods might be over-engineered for a threat model that rarely exists in practice. Consequently, while the theoretical advance is strong, the paper’s practical impact would be much clearer if the authors could better justify why this specific "infinite magnitude" adversary could be a more relevant concern than traditional bounded outliers, or more common real-world failures like high-frequency, small-magnitude errors (e.g., systematic sensor bias).

4. **Originality.** The paper is highly original. To the best of my knowledge, it is the first to formalize and analyze a novel Robust Bayesian Optimization setting where adversarial corruptions can have unbounded magnitudes.

---

> ### Author Rebuttal · Authors · 2026-03-30
>
> We thank the reviewer for their constructive feedback. We address the questions first, then comment on the weaknesses.
>
> ## Questions
>
> ### Q1: Resilience against a white-box adversary injecting within-plateau corruptions
>
> Our regret bounds (Theorems 2 and 3) are **worst-case over all adversary strategies**, including the targeted within-plateau attack the reviewer describes. In theory, as long as the adversary is limited by the frequency at which it applies corruptions, the RCGP algorithms should be able to recover.
>
> The adversary faces two scenarios. If the corruption is large enough to fall outside the plateau, the P-IMQ down-weighting limits its influence on the posterior mean — unlike standard GP, which can be derailed for the entire BO episode by a single large corruption (as proven in Appendix H). If the corruption is of reasonable magnitude and stays within the plateau, the assumption on the adversary's frequency budget becomes the binding constraint: within-plateau corruptions behave like bounded-budget corruptions (amplitude $\leq L$, frequency $\leq T_ c$), which standard GP can recover from with enough time steps.
>
> Empirically, we validated robustness across corruption magnitudes on the Forrester function (100 steps). Here, "magnitude" refers to the observed value assigned to corrupted observations: e.g., at magnitude 30, a corrupted observation is replaced by the value 30 regardless of the true function value at that point.
>
> | Corruption magnitude | FC-RCGP | A2-RCGP |
> |----------------------|---------|---------|
> | 10 | 475 | 586 |
> | 15 | 411 | 336 |
> | 20 | 438 | 384 |
> | 30 | 512 | 495 |
>
> FC-RCGP remains stable across all corruption magnitudes (411–512). The adaptive variant (A2-RCGP) is more sensitive at lower magnitudes but achieves the best individual results at moderate corruption.
>
> It is also worth noting that bounded cumulative corruption approaches such as Bognovic et al. [1] should be better equipped this type of corruption if $T_ c=O(1)$ and if allowed to run for longer BO episodes. We compared against the approach [1] which achieved 1381 cumulative regret vs A2-RCGP's 384 on corrupted Forrester with corruption magnitude 20. We reckon however, that the 100 steps we used for the BO episode are not enough for [1] to recover from the applied corruption.
>
> ## Weaknesses
>
> ### $T_ c$ estimation and practical handling
>
> We agree this should appear earlier in the main text and will move the discussion from Appendix I to Section 4. Empirically, setting $T_ c = 0$ yields near-identical performance to adaptive estimation:
>
> | Method | Estimated $T_ c$ | $T_ c = 0$ |
> |--------|-----------------|-----------|
> | RCGP | 431 | 438 |
> | A2-RCGP | 384 | 384 |
>
> Practitioners can safely ignore setting $T_ c$ entirely. The robustness comes from the RCGP model itself, not the inflated confidence multiplier — the theoretical dependence on $T_ c$ is an artifact of the proof technique. This would be an advantage over standard bounded corruption models such as Bogunovic et al [1] where knowing the adversary's total budget is required.
>
> ### Practical relevance of unbounded corruptions
>
> We want to reframe this point. Our frequency-constrained corruption model is a stronger adversary than the bounded cumulative budget $\sum_ t |c_ t| \leq C$ standard in the literature. Under the existing model, the total corruption budget $C$ must be known and finite — meaning a handful of large corruptions can exhaust it entirely, and truly extreme outliers may violate the assumption outright. Our model removes the magnitude constraint and bounds only the number of corrupted observations, which is a less restrictive set of assumptions and more amenable to many real world settings albeit unperfect on its own.
>
> Crucially, corruptions need not be literally infinite to benefit from our framework. None of our experiments used NaN or infinite values — we relied on finite corruption levels throughout (e.g., values of 25 when the true maximum is ~15 in Forrester). These are bounded magnitudes that are nonetheless large enough to derail standard GP-UCB (as our GP baseline confirms).
>
> An interesting future direction: since our RCGP surrogate is agnostic to the acquisition function, one could combine it with acquisition functions from the bounded-budget literature.
>
> [1] Bogunovic, Ilija, et al. "A robust phased elimination algorithm for corruption-tolerant gaussian process bandits." Advances in Neural Information Processing Systems 35 (2022): 23951-23964.

---

> > ### Author Rebuttal · Reviewer_wdvh · 2026-04-01
> >
> > I thank the authors for addressing my questions.

---

### Official Review · Reviewer_MXef · 2026-03-10

**Soundness:** 3
**Presentation:** 3
**Significance:** 3
**Originality:** 4
**Overall Recommendation:** 4
**Confidence:** 4

**Summary:**

This paper modifies the weight function of the robust conjugate Gaussian process (RCGP)  called P-IMQ to avoid downweighting inlier observations. Furthermore, this paper integrates this extended RCGP (with P-IMQ function) into a classical GP-UCB algorithm and provides the corresponding regret bounds. Moreover, the authors show that their algorithm will converge to GP-UCB when there is no outlier and provide the conditions for their method to achieve sub-linear regret. Finally, they validate their claims through benchmark tasks, including Forrester 1D function and Lunar Lander 12D tasks.

**Compliance With Llm Reviewing Policy:**

Affirmed.

**Final Justification:**

In general, I am satisfied with the paper, other reviews, and the authors' response. I will maintain my good initial score.

**Key Questions For Authors:**

- Figure 2: Once observation $y$ lies outside the plateau, the P-IMQ function tends to penalize $y$ more heavily than IMQ. I believe we can adjust the curvature of this function. Is there any reason for choosing this form? Can authors give comments on how the steepness of the curvature affects the proposed methods?
- It is known that GP-based BO struggles on high-dimensional tasks ($d > 10$). Can authors give comments on how A2/FC-RCGP behaves in such a scenario, e.g., Lunar Lander experiment?

**Limitations:**

yes

**Strengths And Weaknesses:**

Soundness:
- Authors consider both fixed and dynamic length of plateau to develop their robust GP-UCB algorithms.
- Page 4, lines 191 and 213: The authors should provide the intuition or explanation of the choice of $C_w$ and $\Phi$.
- The authors did not compare their method with recent robust GP methods.
- The authors did not compare the experiments with the original RCGP. I think it is reasonable to include RCGP in the experiments and see how the proposed method improves.
- It will strengthen the paper if the authors can provide ablation studies on their method. For example, the impact of $B_f, T_c$ and $\kappa$.
- Providing the conditions to achieve sublinear regret helps the readers to understand the current limitation and perhaps improve the current work.

Presentation:
- Figure 2 provides a clear understanding of how the P-IMQ function behaves differently from its predecessor.
- It would be great if authors could also plot the corresponding P-IMQ function of each experiment.

Significance:
- This work is considered impactful due to the regret bounds provided under the presence of unbounded outliers. These results may initiate further work in theoretical BO works under various outlier scenarios.

Originality:
- This work considers a practical way to extend RCGP as a surrogate for BO (P-IMQ function and exploration bonus term). The regret bounds are novel.

---

> ### Author Rebuttal · Authors · 2026-03-30
>
> We thank the reviewer for their constructive feedback. We address the questions first, then comment on the weaknesses.
>
> ## Questions
>
> ### Q1: P-IMQ curvature and comparison with IMQ (Figure 2)
>
> The caption of Figure 2 does not reflect the true parameters used: different values of $c$ were used for IMQ and P-IMQ, which makes the decay rates appear different. When using the same $c$, both have the same asymptotic decay rate, and the P-IMQ always assigns **higher weights** than IMQ outside the plateau (due to the lag in when down-weighting starts). We will fix the figure with the correct, matched parameters.
>
> Regarding alternative weight functions: any function satisfying RCGP's conditions from Altamirano et al. [1] would work — specifically: (i) $w$ is differentiable w.r.t. $y$, and (ii) $\sup_ {x,y}|y \cdot w^2(x,y)| < \infty$. This includes Gaussian and Matérn kernels. We use the IMQ kernel because it has been shown to be particularly well-suited to robustness against severe outliers in generalised Bayesian inference (Matsubara et al. [2]; Altamirano et al. [1]). Intuitively, more aggressive downweighting is appropriate when one is confident about the plateau construction; otherwise, a more permissive decay is safer.
>
> ### Q2: High-dimensional behavior (Lunar Lander)
>
> None of the algorithms nor baselines converged in 300 iterations on this 36D problem. Although RCGP variants had lower regrets in our experiments, we expect it to suffer from the same pathologies as standard GP-BO in high dimensions (e.g., edge-seeking behavior). The only foreseeable added risk is slightly reduced data efficiency from down-weighting, which the P-IMQ plateau helps mitigate by treating within-plateau observations identically to a standard GP.
>
> ## Weaknesses
> **Note: Ablation results will be added to an appendix.**
>
> ### Intuition for $C_ w$ and $\Psi$
>
> $C_ w$ is a derived constant (Lemma D.3) capturing the maximum influence a single corrupted point can exert on the posterior mean. In theory, it is not a hyperparameter we choose and is solely decided by the weighting function used.
>
> $\Psi(T_ c)$ is the observability penalty (Lemma D.6) from upper-bounding the unobservable $\sigma_ {\text{uc}}$ with the observable $\sigma^R$. $\Psi(T_ c)$ is not a function we can choose but rather derived theoretically. In practice, we can use $T_ c=0$ for which $\Psi(T_ c)=1$ which greatly simplifies the expressions we have.
>
> ### RCGP comparison and plateau width ablation
>
> We swept the plateau width $L$ on the Forrester function with unstandarised outcomes, comparing original RCGP ($L=0$) against P-IMQ variants. Cumulative regret results:
>
> **Corrupted setting:**
>
> | L | RCGP | A2-RCGP |
> |---|------|---------|
> | 0 (original IMQ) | 305 | 441 |
> | 2 | **182** | **227** |
> | 4 | 394 | 232 |
> | 8 | 269 | 265 |
> | 16 | 397 | 413 |
>
> We then run on the **clean setting:**
>
> | L | RCGP | A2-RCGP |
> |---|------|---------|
> | 0 (original IMQ) | 251 | 209 |
> | 2 | **192** | 220 |
>
> $L = 2$ is best in both settings. The original RCGP ($L = 0$) converges, albeit slower. Large $L$ (8, 16) risks missing outliers. We recommend **conservative, tighter plateaus**: a too-narrow plateau still converges (RCGP retains robustness through its weighting) with some efficiency loss, while a too-wide plateau risks letting corruptions pass. The algorithms depend on $B_ f$ and $\kappa$ solely through $L$; our experimtns showed $\sqrt{B_ f \kappa} = 1$ worked reliably with standarisation.
>
> ### Ablation: sensitivity to $T_ c$
>
> | Method | Estimated $T_ c$ | $T_ c = 0$ |
> |--------|-----------------|-----------|
> | RCGP | 431 | 438 |
> | A2-RCGP | 384 | 384 |
>
> The algorithms are insensitive to $T_ c$. Setting $T_ c = 0$ works just as well — robustness comes from the RCGP model, not the inflated confidence multiplier. Note: $T_ c$ here refers to the hyperparameter we set, not the adversary budget (which remained $T^{1/3}$).
>
>
> ### P-IMQ visualization
>
> We will include P-IMQ plots. With outcome standardisation, the P-IMQ looks similar across all experiments.
>
> ### Additional baselines
>
> We added RGP-PE (Bogunovic et al. [3]): 1381 vs A2-RCGP's 384. This is somewhat unfair to RGP-PE: it targets bounded cumulative corruption and needs many more iterations to exhaust the adversary's budget. Our baselines are better suited to the unbounded corruption model we introduce.
>
>
> [1] Altamirano, Matias, François-Xavier Briol, and Jeremias Knoblauch. "Robust and conjugate Gaussian process regression." arXiv preprint arXiv:2311.00463 (2023).
>
> [2] Matsubara, Takuo, Jeremias Knoblauch, François-Xavier Briol, and Chris J. Oates. "Robust generalised Bayesian inference for intractable likelihoods." JRSSB 84(3):997–1022 (2022).
>
> [3] Bogunovic, Ilija, et al. "A robust phased elimination algorithm for corruption-tolerant gaussian process bandits." Advances in Neural Information Processing Systems 35 (2022): 23951-23964.

---

> > ### Author Rebuttal · Reviewer_MXef · 2026-04-02
> >
> > I thank the authors for answering all of my questions.

---

### Official Review · Reviewer_Xx5p · 2026-03-12

**Soundness:** 3
**Presentation:** 3
**Significance:** 2
**Originality:** 3
**Overall Recommendation:** 5
**Confidence:** 4

**Summary:**

The paper studies Bayesian optimization in the presence of an adversary who can corrupt observations with arbitrarily large magnitude, but is limited in the frequency of corruptions it can introduce up to round \(T\). This is a novel framework in BO, as previous work has mostly considered bounded-magnitude corruptions. The authors propose two algorithms, both utilizing Robust Conjugate Gaussian Process (RCGP) surrogates with a weight function that does not penalize observations that lie within an algorithm-dependent interval, but penalizes observations that lie outside it. In this way, they aim to continue learning from honest observations while attaining robustness against extreme outliers.

The choice of the plateau region differs between the proposed algorithms. **FC-RCGP-UCB** uses a fixed plateau region centered around the prior mean, whereas **A2-RCGP-UCB** uses a dynamic plateau region centered around the posterior mean. The authors show that the two algorithms achieve sublinear regret when the frequency of the adversary is budgeted by \(O(T^{1/4})\) and \(O(T^{1/7})\), respectively. They also conduct numerical experiments which show the success of the proposed methods over a non-robust baseline and a more modest improvement over a robust baseline.

**Compliance With Llm Reviewing Policy:**

Affirmed.

**Final Justification:**

I belive the problem formulation is strong and the analysis is coherent. The algorithms and their analysis have moderate novelty. The analysis seemed sound to me. After rebuttal, my concerns have been addressed and I have increased the score.

**Key Questions For Authors:**

1) How do the $\beta$ and $L$ parameters compare between the two algorithms? In particular, are these parameters larger in one algorithm than the other, and what intuition should we have for this difference?

2) In lines 253–255, the paper mentions the algorithm’s dependence on $\sigma_{uc}$ and the need to estimate it. However, as I understand it, both algorithms appear to require only the observable RCGP variance. I therefore did not understand why $\sigma_{uc}$ is required. Could the authors clarify the role of this quantity?

3) How do the proposed algorithms behave when the adversary introduces small corruptions that remain within the plateau region? In this case, the observations would not be downweighted and the algorithm would behave similarly to GP-UCB. Could the authors comment on the robustness of the method in this regime?

4) The experiments section could benefit from including additional robust baselines. In particular, it would be helpful to compare against methods designed for bounded-magnitude corruptions. If the proposed algorithms outperform such methods in the unbounded-corruption setting, this would more convincingly demonstrate the value of the proposed framework.

**Limitations:**

Yes

**Strengths And Weaknesses:**

**Strengths**
- Good problem formulation. The setting of bounded corruption frequency with unlimited corruption magnitude appears to be new in the BO literature and is also practically meaningful, as extreme outliers occur infrequently in many real-world domains.
- The paper is well written in most parts. The problem setup, algorithms, and theoretical results are generally presented clearly, with the exception of the A2-RCGP-UCB algorithm, whose description is harder to follow.
- The theoretical results are novel, as the problem setting has not been studied in the BO literature before. The bounds provided are meaningful in that they identify regimes of corruption frequency where the proposed algorithms guarantee good performance.
- The proposed methods also perform comparably to non-robust baselines in the absence of corruptions.


**Weaknesses**
- A lower bound is lacking; hence it is unclear whether the provided upper bounds are tight. The authors also acknowledge this point, noting that in the experiments the algorithms appear to tolerate more corruption than the theoretical bounds suggest.
- The corruption-frequency regimes guaranteed by the theory are quite restrictive, with sublinear bounds of $O(T^{1/4})$ and $O(T^{1/7})$. This implies that the corruption rate must decrease over time. In many real-world applications, however, extreme outlier events may occur at a roughly constant rate (e.g., following a Poisson process). It would therefore be interesting to analyze the behavior of the algorithms when the corruption budget scales linearly with $T$, for example $T_c = T/10$.
- The algorithms themselves are relatively straightforward, which is not inherently a weakness. However, the regret analysis largely builds on existing tools from the GP-UCB literature. From my perspective, the main technical novelty lies in the introduction of the plateau weight function and its role in the analysis.
- While the proposed weighting scheme provides robustness against extreme outliers, it offers no protection against corruptions that remain within the plateau region. In such cases, the algorithm behaves identically to GP-UCB. As a result, an adversary that injects smaller but frequent corruptions within the plateau region may still significantly influence the posterior. The strong dependence of the regret guarantees on the corruption-frequency budget may partially stem from this design choice.
- The authors note that the theoretical bounds may be loose because the algorithms appear to tolerate a higher corruption frequency in the experiments. However, this may also depend on the adversarial attack strategy used in the experiments. Since the algorithm is inherently robust to extreme outliers, a more challenging attack might consist of corruptions that remain within the plateau region.
- The A2-RCGP-UCB algorithm could be explained in more detail. In particular, the differences between the two proposed algorithms and the roles of their respective variables should be highlighted more clearly.

---

> ### Author Rebuttal · Authors · 2026-03-30
>
> We thank the reviewer for their constructive feedback. We address the questions first, then the weaknesses.
>
> ## Questions
>
> ### Q1: How do β and L compare between the two algorithms?
>
> Intuitively, $L$ defines the region where observations are trusted as uncorrupted — residuals within $L$ are treated identically to a standard GP. The multiplier $\sqrt{\beta_ t}$ constructs inflated confidence intervals. The inflation is there to account for the potential corruptions.
>
> **FC-RCGP-UCB** centers on zero ($g = 0$), so $L_ T = \sqrt{B_ f \kappa} + N_ T(\delta/3)$ must cover the full function range plus noise. Its multiplier is $\sqrt{\beta_ t} = \sqrt{\beta'_ t} + C_ {w,T}\sqrt{T_ c}$ with $C_ {w,T} = \tilde{O}(\sqrt{\beta'_ T})$.
>
> **A2-RCGP-UCB**: $M_ A$ uses identical parameters. $M_ R$ centers on the Anchor's posterior mean $g_ R(x) = \mu_ {A,t-1}(x)$, so it models *residuals* from $M_ A$ rather than raw observations. Since the Anchor captures much of $f$'s signal, the residuals $y_ t - \mu_ {A,t-1}(x_ t) \approx f(x_ t) - \mu_ {A,t-1}(x_ t) + \epsilon_ t$ are dominated by the Anchor's remaining uncertainty. This is why the plateau shrinks: $L_ {R,t}(x) = \sqrt{\beta_ {A,t}} \Psi((t-1)_ c) \sigma^R_ {A,t-1}(x) + N_ T(\delta/3)$ tightens as the Anchor learns and $\sigma^R_ {A,t-1}(x)$ decreases.
>
> The multiplier $\sqrt{\beta_ {R,t}}$ is however larger: $C_ {w,R,T} = \tilde{O}(\Psi(T_ c)\sqrt{\beta'_ T}(1+\sqrt{T_ c}))$, because $g_ R = \mu_ {A,t-1}$ is itself influenced by corruptions — the Anchor's deviation propagates into $M_ R$'s plateau, requiring wider confidence intervals. A2-RCGP-UCB trades a tighter plateau (better local sensitivity) for a wider confidence interval (covering the Anchor's estimation error).
>
> Setting $T_ c = 0$ yields near-identical Forrester results (FC-RCGP: 438 vs 431; A2-RCGP: 384 vs 384), so in practice these expressions simplify.
>
> ### Q2: Role of σ_ uc
>
> This was an exposition oversight. The paragraph describes an earlier version relying on $\sigma_ {\text{uc}}$. The submitted algorithms use $\Psi(T_ c)$ (Lemma 6) to relate $\sigma_ {\text{uc}}$ to the observable $\sigma^R$, resolving observability at the cost of more conservative rates. We will update that section.
>
> ### Q3: Robustness against within-plateau corruptions
>
> Within the plateau, RCGP reduces to a standard GP update. Robustness then comes from the **frequency constraint**: within-plateau corruptions are bounded in amplitude by $L$ and frequency by a **sublinear** $T_ c$, for a cumulative budget $\leq L \cdot T_ c$. A robust standard GP method is able to recover over time when $T_c$ is heavily constrained (the literature assumes $T_ c = O(1)$). **Our regret bounds are worst-case over all adversary strategies**, so the theoretical guarantees already cover this scenario.
>
> That said, bounded-corruption models such as Bogunovic et al. [1] would handle within-plateau adversaries more effectively **if** $T_ c = O(1)$. Our framework is still less restrictive on the allowed adversary.
>
> Empirically, we validated across corruption magnitudess on Forrester (100 steps). FC-RCGP remained stable (regret 411–512). A2-RCGP achieves the best results at moderate corruption (336 at level 15) but was sensitive at lower corruption magnitudes. DiagnosticsGP wins at the highest level, as expected: it fully excludes outliers outside its threshold.
>
> ### Q4: Additional robust baselines
>
> We added RGP-PE (Bogunovic et al., [1]): 1381 vs A2-RCGP's 384 on corrupted Forrester. A2-RCGP is the clear winner. RGP-PE targets bounded cumulative corruption and needs more iterations to exhaust the adversary's budget. Student-t UCB and DiagnosticsGP, already included, are better suited for the unbounded setting.
>
> ## Weaknesses
>
> ### Lower bounds
> The bounds are likely loose due to the proof technique. Tighter bounds may be achievable as future work.
>
> ### Restrictive corruption-frequency regime
> Our adversary is stronger than bounded cumulative corruption: the latter implies only a handful of observations can have noticeable magnitude. We believe $T_ c = \Theta(T)$ with sizeable magnitude is fundamentally incompatible with sublinear regret, as each large corruption requires one or more nearby clean observations to recover from. Our experiments demonstrate robustness at $T_ c = O(T^{1/3})$, already exceeding proven rates.
>
> ### Novelty
> The contribution lies in combining RCGP with UCB, which is non-trivial: without the plateau, the RCGP bias term $m_ w$ must be carefully controlled to preserve UCB-style analysis. The plateau makes this achievable while retaining zero-cost robustness.
>
> ### A2-RCGP-UCB exposition
> We will revise the A2-RCGP-UCB presentation to better highlight the two-model structure.
>
> [1] Bogunovic, Ilija, et al. "A robust phased elimination algorithm for corruption-tolerant gaussian process bandits." Advances in Neural Information Processing Systems 35 (2022): 23951-23964.

---

> > ### Author Rebuttal · Reviewer_Xx5p · 2026-04-02
> >
> > I thank the authors for their responses. My main concerns are addressed, so I will increase my score to accept, as I think the problem formulation and analysis advance the subfield.

---

### Decision · Program_Chairs · 2026-04-30

**Decision:**

Accept (regular)

**Comment:**

This paper studies Bayesian optimization under unbounded corruptions with a limit on the corruption frequency.

All reviewers noted that their concerns have been addressed by the author rebuttal, except Reviewer fhHM, who raised further questions. As a response to Reviewer fhHM, the authors posted a reply rebuttal comment further clarifying concerns on the dependence of the results on the corruption frequency parameter, sensitivity to the weighting function, and strengths of the assumptions. These concerns have been adequately addressed.

The main strength of the paper is studying the novel setting of unbounded corruptions for the first time in BO in a rigorous and comprehensive way.

Some important criticisms were:

(i) Lack of lower bounds under corruptions: The authors acknowledge this and leave it as future work.

(ii) Corruption frequencies being sublinear in $T$: The authors argue that their adversary is stronger than an adversary with bounded cumulative corruption. However, it seems that the maximum rate of corruption that can be handled with sublinear regret remains an open problem.

(iii) How robust algorithms are against within-plateau corruptions and against other robust baselines, sensitivity to parameters, and ablation studies: New simulations are conducted by the authors in the rebuttal phase.

Most of the concerns are addressed, and the remaining ones do not diminish the significance of the current work. The authors should openly discuss the limitations of their approach and future research directions in the final version of the paper. The clarifications and new experiments that were made during the rebuttal phase were critical in reaching consensus among the reviewers and the final decision. Therefore, these changes should be carefully integrated into the final version of the paper.